# CHEM-R: LEARNING TO REASON AS A CHEMIST

## ABSTRACT

Although large language models (LLMs) have significant potential to advance chemical discovery, current LLMs lack core chemical knowledge, produce unreliable reasoning trajectories, and exhibit suboptimal performance across diverse chemical tasks. To address these challenges, we propose **Chem-R**, a generalizable **Chem**ical **R**easoning model designed to emulate the deliberative processes of chemists. Chem-R is trained through a three-phase framework that progressively builds advanced reasoning capabilities, including: 1) Chemical Foundation Training, which establishes core chemical knowledge. 2) Chemical Reasoning Protocol Distillation, incorporating structured, expert-like reasoning traces to guide systematic and reliable problem solving. 3) Multi-task Group Relative Policy Optimization that optimizes the model for balanced performance across diverse molecular- and reaction-level tasks. This structured pipeline enables Chem-R to achieve state-of-the-art performance on comprehensive benchmarks, surpassing leading large language models, including Gemini-2.5-Pro and DeepSeek-R1, by up to 32% on molecular tasks and 48% on reaction tasks. Meanwhile, Chem-R also consistently outperforms the existing chemical foundation models across both molecular and reaction level tasks. These results highlight Chem-R's robust generalization, interpretability, and potential as a foundation for next-generation AI-driven chemical discovery.

## 1 INTRODUCTION

Large Language Models (LLMs) have recently emerged as a transformative force in scientific discovery (Bai et al., 2025; Ma et al., 2024; Shojaee et al., 2024; Hatakeyama-Sato et al., 2023; Xia et al., 2025). Within the field of chemistry, LLMs demonstrate exceptional potential by learning expressive representations and knowledge of molecular structures and chemical reactions directly from large-scale datasets. This capability enables them to support a wide array of tasks, including molecular property prediction, reaction outcome estimation, retrosynthetic route planning, and reagent selection (Zhang et al., 2024b; Tan et al., 2025; Zhao et al., 2024; Jiang et al., 2025). However, the lack of chemical structured and reliable reasoning processes in current LLMs leads to suboptimal performance and limited interpretability on complex chemical problems.

Specifically, current LLMs encounter three fundamental challenges in performing chemical reasoning. **Challenge 1**: Current LLMs often lack the essential "chemical fundamentals", leading to frequent mistakes in molecular representations and reaction rules, which underminesreliability at the initial reasoning stage (Zhong et al., 2024; Liu et al., 2025). As illustrated in Figure 1 (Challenge 1), several Chain-of-Thought (CoT) on different tasks generated by DeepSeek-R1 (Guo et al., 2025) demonstrates that the model may fail to recognize basic SMILES (Weininger, 1988) and IUPAC (Kuhn et al., 2004), which undermines the reliability of any subsequent reasoning. **Challenge 2**: The model's reasoning process is fundamentally flawed because it is unsystematic, failing to adhere to the coherent, step-by-step workflow of an expert (Ouyang et al., 2023; Bran et al., 2025). This lack of a structured approach makes the reasoning unreliable and prone to factual errors. This lack of structure results in a confusing and untrustworthy reasoning process, as exemplified in Figure 1 (Challenge 2) where the model misidentifies fundamental functional groups and generates a flawed, unstructured line of reasoning. **Challenge 3**: As shown in Figure 1 (Challenge 3), even when models are guided by explicit reasoning patterns, their performance across diverse molecular and reaction level tasks remains highly imbalanced, with strong tasks dominating and weaker tasks

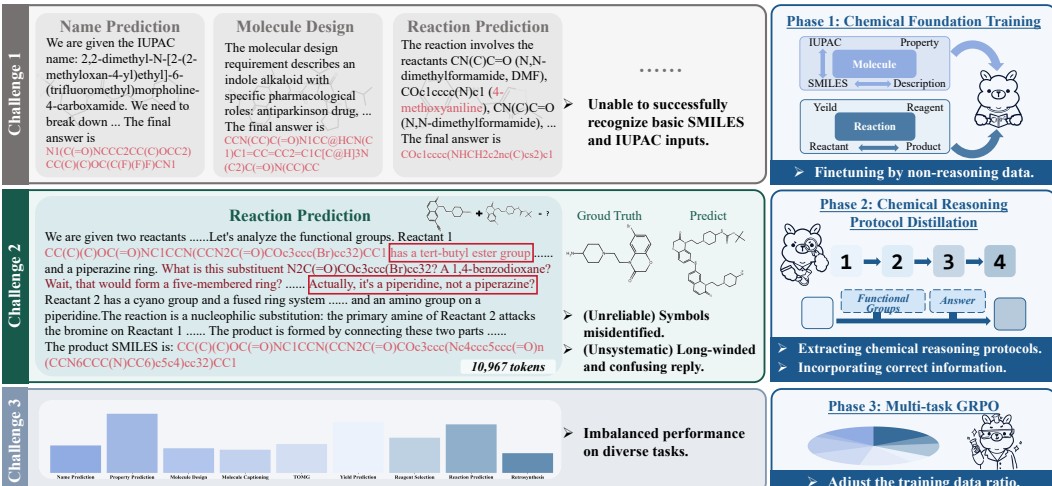

Figure 1: **Challenges and the proposed Chem-R solution.** The left panel highlights three key deficiencies observed in current reasoning models. To overcome these limitations, we introduce a three-phase training framework, illustrated on the right. This strategy is designed to first build a solid chemical foundation (Phase 1), then instill correct, step-by-step reasoning pathways (Phase 2), and finally, optimize for balanced, multi-task proficiency (Phase 3).

underrepresented. Together, these issues highlight that effective chemical reasoning requires domain knowledge, reliable and structured thought, and balanced generalization across heterogeneous tasks.

To address these challenges, we propose Chem-R, a unified framework comprising three phases that enables structured reasoning in molecular- and reaction-level tasks. As shown in Fig. 1, Chem-R follows a three-phase training paradigm, where each phase systematically mitigates one of the aforementioned bottlenecks. **Phase 1**: Chemical Foundation Training equips the model with robust chemical fundamentals by fine-tuning on large-scale non-reasoning corpora, covering both molecular representations (*e.g.*, SMILES, IUPAC) and reaction-level patterns, thereby reducing elementary errors. **Phase 2**: Chemical Reasoning Protocol (CRP) Distillation leverages structured protocols to guide a general-purpose teacher model toward expert-level chemical reasoning, subsequently distilling these strategies into a student model. In this process, expert-like protocols are converted into reusable, modular workflows that facilitate coherent and interpretable problem-solving. As illustrated in bottom row of Fig. 1, the reasoning process can be enhanced by extracting structured Chemical Reasoning Protocols, incorporating correction information, and providing targeted hints to mitigate common errors. **Phase 3**: Multi-task Group Relative Policy Optimization (Multi-task GRPO) further enhances the learned reasoning paradigm across heterogeneous tasks, employing a curriculum-like weighting scheme to prevent strong-task dominance and improve performance balance. Together, these three phases form a principled pipeline that not only reduces low-level mistakes, but also enables the model to generate chemically sound, structured, and explainable reasoning across both molecular and reaction domains.

Our main contributions are summarized as follows:

(1) We propose Chem-R, a unified three-phase framework that enables structured and generalizable chemical reasoning across both molecular and reaction level tasks. **Phase 1 (Chemical Foundation Training)** equips the model with robust chemical fundamentals by pre-training on large-scale non-reasoning corpora. **Phase 2 (CRP Distillation)** introduces Chemical Reasoning Protocols (CRP) distilled from a teacher model, providing modular and interpretable workflows that guide problem solving. **Phase 3 (Multi-task GRPO)** applies GRPO with a curriculum-like weighting scheme to enhance and balance performance across heterogeneous tasks.

(2) We comprehensively evaluate the model on four widely used benchmarks, including Chem-LLMBench (Guo et al., 2023), ChEBI-20 (Edwards et al., 2022), TOMG-Bench (Li et al., 2024a), and USPTO (Schneider et al., 2016). Our evaluation spans two major families of tasks, namely molecular- and reaction-level tasks, covering nine macro-tasks and 25 sub-tasks in total. Across this diverse suite, Chem-R consistently achieves state-of-the-art performance. For instance, compared with ChemDFM-v1.0-13B (Zhao et al., 2025c), Gemini-2.5-Pro (Comanici et al., 2025), and DeepSeek-R1 (Guo et al., 2025), Chem-R improves by 33%, 32%, and 44% on Name Prediction (Exact Match), and by 53%, 50%, and 52% on Yield Prediction (Accuracy), respectively. These sub-

stantial gains highlight Chem-R's ability to deliver both reliable accuracy and robust generalization across heterogeneous molecular and reaction tasks.

## 2 RELATED WORK

### 2.1 REASONING FOR LLMS

Generating a CoT (Wei et al., 2022; Kojima et al., 2022) significantly improves the ability of LLMs to perform complex reasoning. To elicit high-quality reasoning chains, various strategies have been proposed, including rejection sampling (Liu et al., 2023a; Tong et al., 2024), reward modeling (Ouyang et al., 2022; Zhang et al., 2025b), and preference learning (Rafailov et al., 2023; Pang et al., 2024). More recently, DeepSeek-R1 (Guo et al., 2025) has shown that complex reasoning behaviors (Gandhi et al., 2025) can emerge from simple rule-based reinforcement learning, particularly when initialized with a cold start phase using CoT data.

However, a common limitation of these approaches is their reliance on outcome-based supervision, which can produce unstructured, inconsistent and flawed reasoning chains (Arcuschin et al., 2025; Chen et al., 2025), a critical risk in scientific applications. To address this, process-level supervision provides fine-grained feedback on each intermediate step (Lightman et al., 2024; Wang et al., 2024; Zhang et al., 2024a; 2025a). Another strategy involves multi-model systems where verifier models scrutinize the reasoning process of a primary generator model (Du et al., 2023; Kirchner et al., 2024; Baker et al., 2025).

### 2.2 LLMS FOR CHEMISTRY

The application of LLMs is driving a paradigm shift in chemistry. Early work demonstrated that generalist models possess latent chemical knowledge (Hatakeyama-Sato et al., 2023; Sallam et al., 2024), paving the way for specialized models fine-tuned for chemistry-specific tasks. These include models like ChemLLM (Zhang et al., 2024b), ChemMLLM (Tan et al., 2025), Chem3DLLM (Jiang et al., 2025), ChemDFM (Zhao et al., 2025c) and others (Liu et al., 2023c; Zhang et al., 2025c; Li et al., 2025c), which handle tasks ranging from molecular captioning to reaction analysis.

More recent advancements have focused on complex reasoning and cross-domain integration. Reasoning models such as ether0 (Narayanan et al., 2025) and ChemDFM-R (Zhao et al., 2025b), trained via reinforcement learning, exhibit strong performance across diverse chemical tasks and provide transparent, interpretable outputs. In parallel, scientific foundation models like NatureLM (Xia et al., 2025) and Intern-S1 (Bai et al., 2025) have been trained on large-scale data from various scientific fields. These models can handle a diverse range of inputs spanning biology, chemistry, and materials science. Despite these advances, the progress of foundation models in chemistry lags significantly behind that in high-resource domains like mathematics and code, largely due to the relative scarcity of specialized scientific data for training (Bai et al., 2025). To overcome this data bottleneck, we introduce a specialized pipeline designed to synthesize high-quality CoT data on par with expert-level annotations for model training.

## 3 METHOD

**Overview.** We propose Chem-R, a three-phase framework explicitly designed to endow LLMs with structured and generalizable chemical reasoning capabilities. **Phase 1** (Section 3.1) establishes a chemistry-aware foundation by supervised fine-tuning on large-scale non-reasoning corpora, grounding the model in valid molecular and reaction representations. **Phase 2** (Section 3.2) introduces Chemical Reasoning Protocol (CRP) Distillation, which transfers structured and reusable reasoning workflows from a teacher model into a compact student model. **Phase 3** (Section 3.3) employs Multi-task Group Relative Policy Optimization (Multi-task GRPO) to further enhance and balance performance across diverse molecular- and reaction-level tasks. Together, these phases form a principled pipeline that transforms ad-hoc CoT traces into chemically sound, interpretable, and broadly generalizable reasoning.

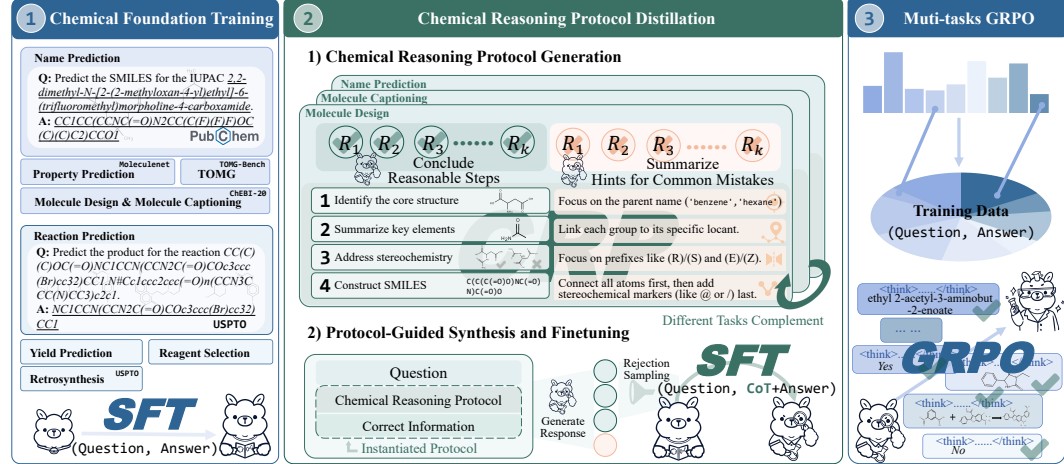

Figure 2: **The overall pipeline of Chem-R.** The model is trained in three phases. **1) Chemical Foundation Training:** Instills basic chemical knowledge using question-answer pairs. **2) Chemical Reasoning Protocol Distillation:** Teaches structured reasoning by fine-tuning on protocol-guided CoT. **3) Multi-task GRPO:** Refines reasoning skills across all tasks using reinforcement learning.

## 3.1 PHASE 1: CHEMICAL FOUNDATION TRAINING

Establishing a reliable chemical LLM necessitates the integration of domain-specific knowledge in molecular representations and reaction notation. General-purpose corpora (*e.g.*, Wikipedia, text-books) are inadequate in this regard, as they rarely capture the syntactic rules of SMILES strings or the systematic regularities of IUPAC nomenclature, let alone the canonical mapping between different molecular descriptors (Luo et al., 2022; Taylor et al., 2022; Irwin et al., 2022). To this end, Phase 1 establishes a chemistry-aware foundation by supervised fine-tuning (SFT) on large-scale non-reasoning corpora $\mathcal{D}_{\text{chem}}$, thereby grounding the model in chemically valid input–output behaviors.

Formally, $\mathcal{D}_{\text{chem}}$ is a paired chemistry corpus, $\mathcal{D}_{\text{chem}} = \{(x_i, y_i)\}_{i=1}^{N}$, where $x_i$ represents a structured chemical input (*e.g.*, a SMILES string, an IUPAC name, or a reaction query), and $y_i$ is the corresponding labels (*e.g.*, a canonical IUPAC name, a valid SMILES string, or the main product of a chemical reaction). $\mathcal{D}_{\text{chem}}$ encompasses both molecular- and reaction-level supervision.

At the **molecule level**, the corpus aligns alternative descriptors of the same compound. This enables the model to master not only the bidirectional translation between SMILES and IUPAC forms, but also the mapping from a molecular structure to its textual description. Such examples teach the model chemically consistent string generation and reduce elementary notational errors.

At the **reaction level**, the corpus encodes prototypical transformations, mapping reactants to their products or reagents and specifying the functional roles of reagents and conditions. Although such instances require only static mapping rather than explicit reasoning, they provide essential priors that prevent chemically implausible outcomes.

In this phase, the model is trained via supervised fine-tuning (SFT) to internalize the syntax and semantics of $\mathcal{D}_{\text{chem}}$, as illustrated in Figure 2 (left). This chemistry-aware initialization substantially reduces representational errors and serves as the basis for structured reasoning in subsequent phases. Examples of data used in the first phase can be found in Appendix B.

## 3.2 PHASE 2: CHEMICAL REASONING PROTOCOL DISTILLATION

While Phase 1 equips the model with foundational chemical knowledge by training on correct question-answer pairs, it does not yet instill the ability to perform reliable and structured reasoning. A conventional approach to bridge this gap is to distill CoT data from a more powerful teacher model and then train the student model. However, this direct distillation often perpetuates the exact problems we seek to eliminate; as shown in Figure 1, even advanced models can produce chaotic and unreliable reasoning trajectories. To address this challenge and ensure the distilled reasoning is of high quality, we introduce Phase 2 (as shown in middle pannel of Figure 2): CRP Distillation. This phase consists of two stages: (I) generating an expert-like reasoning protocol, and (II) using

this protocol to guide the synthesis of high-quality CoT data for finetuning. The overall objective is to teach Chem-R a systematic and reliable reasoning methodology.

**I. Chemical Reasoning Protocol Generation.** It creates the expert's blueprint for chemical reasoning. For a given task, we use a simple prompt to have the teacher model generate multiple, varied responses. From this collection of responses, we then sample $k$ positive examples (reasoning paths leading to the correct answer, as $R_i$ in Figure 2) and $k$ negative examples (those leading to incorrect answers). The teacher model systematically analyzes these positive examples to conclude a generalizable, step-by-step reasoning template. Concurrently, the teacher model also examines failed reasoning attempts to identify common mistakes, summarizing them as cautionary guidance attached to each step of the protocol. This process results in a strong thinking guide for each task. Furthermore, we create a more holistic and robust final reasoning guide by merging the cautionary guidance from analogous steps across different tasks, enriching the protocol for any given step with insights from as many relevant contexts as possible.

**II. Protocol-Guided Synthesis and Finetuning.** With the expert protocol established, the second stage focuses on data synthesis and student model training. For each question, we guide the teacher model by providing it with an *Instantiated Protocol*, a combination of the task's governing CRP and the reliable correct information (*i.e.*, functional groups and final answer). This prompts the model to produce a detailed CoT that strictly adheres to the protocol's structured steps. To ensure the absolute quality and logical fidelity of this synthetic data, we implement a Rejected Sampling mechanism. Specifically, the answer portion of a generated CoT (*e.g.*, tokens included in `<answer>` tag) is removed, and the model must regenerate the answer based solely on the preceding reasoning. Only those CoT paths where the regenerated answer matches the original correct answer are retained, guaranteeing that the reasoning logically and consistently leads to the correct solution. Finally, this curated dataset of pristine (`Question, CoT + Answer`) pairs is used to fine-tune Chem-R via SFT, effectively teaching it to internalize and replicate a reliable and interpretable reasoning process.

### 3.3 PHASE 3: MULTI-TASK GRPO

While Phase 2 equips the model with structured reasoning protocols, ensuring their robust execution across heterogeneous tasks remains non-trivial. In particular, naive multi-task training tends to favor easier or high-resource tasks, causing weaker tasks to be underrepresented and resulting in imbalanced performance. To overcome this issue, we introduce a *Multi-task GRPO* scheme, which enhances protocol-guided reasoning while explicitly enforcing balance across tasks.

Let $\mathcal{T}$ denote the task mixture. For each task $t \in \mathcal{T}$, we estimate its validation accuracy $s_t$ after Phase 2, and assign a sampling probability $p_t$ that up-weights weaker tasks:

$$p_t = \frac{(1 - s_t)^\alpha}{\sum_{t' \in \mathcal{T}} (1 - s_{t'})^\alpha}, \tag{1}$$

where $\alpha \geq 0$ controls the strength of reweighting. This adaptive curriculum ensures that difficult or underperforming tasks contribute proportionally more updates, thereby mitigating strong-task dominance and fostering balanced improvement. For each sampled question $q$, we roll out $G$ responses $\{o_i\}_{i=1}^G$ using the current policy $\pi_{\theta_{\text{old}}}$. Each token $o_{i,t}$ within a trajectory is optimized under a clipped-ratio surrogate with KL regularization:

$$
\begin{aligned}
J_{\text{GRPO}}(\theta) = \mathbb{E}_{q, \{o_i\}_{i=1}^G} \Big[ &\frac{1}{G} \sum_{i=1}^G \sum_{t=1}^{|o_i|} \min\big(\frac{\pi_\theta(o_{i,t}|q)}{\pi_{\theta_{\text{old}}}(o_{i,t}|q)} A_i, \\
&\text{clip}\big(\frac{\pi_\theta(o_{i,t}|q)}{\pi_{\theta_{\text{old}}}(o_{i,t}|q)}, 1-\epsilon, 1+\epsilon\big) A_i\big) - \beta D_{\text{KL}}(\pi_\theta \| \pi_{\text{ref}}) \Big].
\end{aligned}
\tag{2}
$$

Here, $o_{i,t}$ is the $t$-th token of the $i$-th response $o_i$, which has length $|o_i|$, $\epsilon$ is a hyperparameter that defines the clipping range, $A_i$ is the normalized group advantage, and $D_{\text{KL}}(\pi_\theta \| \pi_{\text{ref}})$ is a KL divergence regularizer, weighted by $\beta$, that penalizes deviation from a reference policy $\pi_{\text{ref}}$ (initialized from Phase 2). As for our reward design, we do not use any format-based rewards. The accuracy-based rewards are task-specific, with detailed calculations provided in Section 4.1.

## 4 EXPERIMENTS

### 4.1 EXPERIMENTAL SETUP

**Benchmarks.** We collect four widely used chemical benchmarks, *ChemLLMBench* (Guo et al., 2023), *ChEBI-20* (Edwards et al., 2022), *TOMG-Bench* (Li et al., 2024a), and *USPTO* (Schneider et al., 2016), with their official splits to ensure a fair comparison. Based on these, we organize the evaluation into two families: *molecular tasks* and *reaction tasks*, covering 9 macro-tasks with 25 sub-tasks in total. **Molecular tasks** include (1) *name prediction* (IUPAC↔SMILES); (2) *property prediction* on BBBP, HIV, Tox21, ClinTox, and BACE (Wu et al., 2018); (3) *molecule design* from text to SMILES; (4) *molecule captioning* from SMILES to text; (5) *text-based open molecule generation* includes molecule editing (with functional group addition, replacement, or removal), molecule optimization (guided toward target LogP, MR, and QED) and customized molecule generation (by atom count, bond count, and functional-group count). **Reaction tasks** include (6) *yield prediction* for Buchwald–Hartwig and Suzuki reactions; (7) *reagent selection* for reactant, solvent, and ligand in multiple-choice form (8) *reaction prediction* and (9) *retrosynthesis*. More detailed descriptions of the tasks can be found in Appendix B.

**Evaluation Metrics.** *We adopt task-specific evaluation metrics aligned with prior work.* For *name prediction* (1), we report exact match between predicted and reference strings. For *property prediction* (2) and *yield prediction* (6), which are binary classification tasks, we use average Accuracy across datasets. For *molecule design* (3), we measure exact match on the generated SMILES. For *molecule captioning* (4), we compute BLEU-4 to evaluate text generation quality. For *text-based open molecule generation* (5), which covers editing, optimization, and customized generation, we report weighted accuracy over constraints such as functional groups, atom counts, and property targets. For *reagent selection* (7), we evaluate by multiple-choice accuracy. For *reaction prediction* (8) and *retrosynthesis* (9), we use exact match on canonical SMILES, with unordered set matching for multi-product or multi-reactant cases separated by ".". *All SMILES and IUPAC comparisons are performed after canonicalization to ensure consistency.* For a more comprehensive analysis, supplementary metrics for these tasks are also reported in the Appendix D.

**Baselines.** We group baselines into five families, with the first four reported in the main tables and the fifth provided in Appendix D. The first group consists of *general foundation models*. These include Llama-3.1-8B-Instruct, Llama-3.3-70B (Dubey et al., 2024), and GPT-4o (Hurst et al., 2024). This group establishes the capability of non–chemistry-adapted systems. The second group is *general reasoning models*. Examples are Gemini-2.5-Pro (Comanici et al., 2025), DeepSeek-R1 (Guo et al., 2025), and QWQ-32B (Yang et al., 2025). These models test whether generic reasoning gains transfer to chemistry. The third group contains *chemical foundation models*. These are ChemLLM-DPO-20B (Zhang et al., 2024b), ChemDFM-v1.0-13B, and ChemDFM-v1.5-8B (Zhao et al., 2025c). This set of models emphasizes chemistry knowledge coverage without explicit multi-step reasoning optimization. The fourth group includes *chemical reasoning models*, such as ether0-24B (Narayanan et al., 2025) and our Chem-R, which target process-level reasoning and interpretability. To account for task-specific nuances, we additionally compare against a fifth group, *task-specialized models*, in the appendix. We evaluate these under each benchmark's standard protocol with unified normalization and scoring scripts for fairness and reproducibility.

**Implementation Details.** We select Llama-3.1-8B-Instruct as our base model, and Llama-3.3-70B-Instruct as our teacher model. The detailed data configurations, hyperparameter settings and specific prompts are provided in Appendix C.

### 4.2 MAIN RESULTS

As shown in Table 1, our 8B model sets a new state-of-the-art across a diverse range of chemical benchmarks. It surpasses not only general-purpose models like Gemini-2.5-Pro and other chemical foundation models like ether0-24B, but also outperforms task-specific specialist models in Property Prediction, Molecule Design, and TOMG tasks (see Appendix D). While the non-reasoning chemical model ChemDFM-v1.5-8B achieves a higher score in the Molecule Design task with direct outputs, Chem-R provides interpretable, step-by-step reasoning chains, offering critical explainability

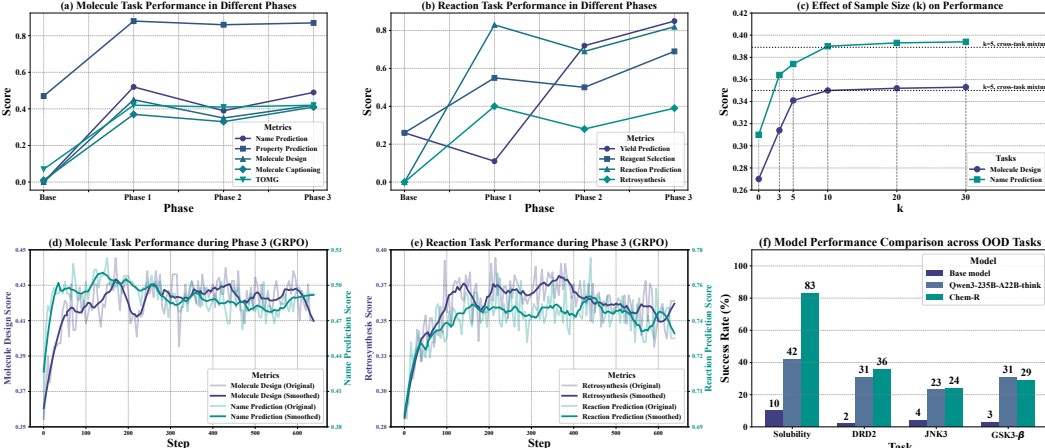

Figure 3: **Comprehensive evaluation of Chem-R.** (a) Molecule task performance in different phases. (b) Reaction task performance in different phases. (c) Effect of sample size ($k$) on performance. (d) Molecule task performance during phase 3. (e) Reaction task performance during phase 3. (f) Model performance comparison across OOD tasks in ChemCoTBench (Li et al., 2025a).

for scientific discovery. The model's most significant advances are in reaction-related tasks, where its performance represents a paradigm shift. Chem-R achieves a score of 0.85 in Yield Prediction (*more than doubling* the next-best score) and 0.39 in Retrosynthesis, *a nearly threefold* improvement over the strongest baseline of 0.15. These results validate that our methodology enables superior chemical reasoning within an efficient 8B parameter model. Additionally, the model's performance at various training phases is shown in Figures 3(a) and (b). For the performance curve during the multi-task GRPO phase, please refer to Figures 3(d) and (e).

Table 1: Performance of different models on chemistry-related tasks. The score for each major task is the average of its subtasks. Column headers use short names: **Name** = Name Prediction (evaluated by Exact Match), **Prop.** = Property Prediction (Accuracy), **Design** = Molecule Design (Accuracy), **Capt.** = Molecule Captioning (BLEU-4), **TOMG** = Tasks in TOMG-Bench (Weighted Accuracy), **Yield** = Yield Prediction (Accuracy), **Reag.** = Reagents Selection (Accuracy), **React.** = Reaction Prediction (Accuracy), **Retro** = Retrosynthesis (Exact Match). For each column: the best and second-best models are highlighted.

| Model | Molecule Tasks | | | | | Reaction Tasks | | | |
|---|---|---|---|---|---|---|---|---|---|
| | **Name** | **Prop.** | **Design** | **Capt.** | **TOMG** | **Yield** | **Reag.** | **React.** | **Retro** |
| *General Foundation Models* | | | | | | | | | |
| GPT-4o | 0.01 | 0.68 | 0.07 | 0.01 | **0.32** | 0.20 | 0.26 | 0.04 | 0.00 |
| Llama-3.1-8B-Instruct | 0.00 | 0.47 | 0.00 | 0.01 | 0.07 | 0.26 | 0.26 | 0.00 | 0.00 |
| Llama-3.3-70B | 0.01 | 0.64 | 0.03 | 0.02 | 0.30 | 0.22 | 0.38 | 0.03 | 0.00 |
| *General Reasoning Models* | | | | | | | | | |
| Gemini-2.5-Pro | **0.17** | 0.56 | 0.29 | 0.04 | – | 0.35 | 0.27 | 0.04 | **0.15** |
| DeepSeek-R1 | 0.05 | 0.63 | 0.22 | 0.04 | – | 0.33 | 0.13 | 0.34 | 0.13 |
| QWQ-32B | 0.01 | 0.71 | 0.03 | 0.04 | 0.30 | 0.29 | **0.39** | 0.01 | 0.00 |
| *Chemical Foundation Models* | | | | | | | | | |
| ChemLLM-DPO-20B | 0.00 | 0.49 | 0.00 | 0.03 | 0.10 | 0.21 | 0.08 | 0.02 | 0.00 |
| ChemDFM-v1.5-8B | 0.14 | 0.74 | **0.53** | 0.10 | 0.12 | **0.37** | 0.35 | 0.50 | 0.07 |
| ChemDFM-v1.0-13B | 0.16 | **0.78** | **0.42** | **0.27** | 0.27 | 0.32 | 0.37 | 0.31 | 0.04 |
| *Chemical Reasoning Models* | | | | | | | | | |
| ether0-24B | 0.15 | 0.64 | 0.30 | 0.03 | 0.03 | 0.03 | 0.21 | **0.65** | 0.04 |
| **Chem-R-8B (Ours)** | **0.49** | **0.87** | **0.42** | **0.41** | **0.42** | **0.85** | **0.69** | **0.82** | **0.39** |

## 4.3 ABLATION STUDY

**Phase-wise Contributions.** To understand the unique contribution of each training phase, we systematically removed individual phases while holding all other variables constant. The results, shown in Table 2 (A), confirm that all three phases are essential and work synergistically. First, *Phase 1*

*(Foundation Training)* establishes the model's fundamental understanding of chemistry. Removing this phase severely degrades performance, as seen by the Name Prediction score dropping from 0.49 to 0.14. While critical, this phase does not by itself enable the chain-of-thought reasoning necessary for explainability. The ability to reason emerges in *Phase 2 (CRP Distillation)*, which introduces the core reasoning framework. Without it, the model fails at performing complex reasoning; for example, performance on Reaction Prediction collapses to zero when both Phase 1 and 2 are removed. Lastly, *Phase 3 (Multi-task GRPO)* acts as a crucial refinement stage. Building on the skills from the previous phases, it delivers consistent improvements, boosting the Reagent Selection score from 0.50 to 0.69.

**Components of Instantiated Protocol in Phase 2.**   We analyze the two core components of our CRP Distillation in Phase 2: the task's governing CRP and *Correct Information*. As detailed in Table 2 (B) and cases in Appendix E, the *CRP* is crucial for improving accuracy by enforcing a logical structure on the reasoning. Its absence results in a consistent performance decline across tasks; for example, the Retrosynthesis score decreases from 0.28 to 0.20 (compared with Chem-R w/o Phase 3). Furthermore, incorporating the ground-truth *Correct Information* is essential for generating a high-quality, large-scale dataset of reasoning chains. Removing this component leads to a severe degradation in performance, with the Reaction Prediction score dropping from 0.69 to only 0.13. Therefore, the CRP provides the indispensable reasoning architecture, while the Correct Information ensures that architecture is used to teach truth, not sophisticated error.

**Single-task vs. Multi-task Training in Phase 2.**   As shown in Table 2 (C), we compare specialized *Single-task* models against a unified *Multi-task* model in Phase 2. While *Single-task* models (*i.e.*, 9 models in total) achieve high scores on their respective tasks, such as 0.75 in Reaction Prediction, confirming the quality of our distilled CoT data, the *Multi-task* model demonstrates clear positive transfer. It outperforms *Single-task* models on related tasks like Reagent Selection (0.50 vs. 0.46) and Retrosynthesis (0.28 vs. 0.26).

**Effect of Sample Size $k$ in Phase 2.**   We investigate the effect of the sample size, $k$, used to generate the CRP in Phase 2. As shown in Figure 3 (c), performance on both Name Prediction and Molecule Design improves rapidly as $k$ increases, but the gains begin to diminish significantly after $k$ reaches approximately 10. This indicates that simply collecting more samples for a single task yields limited returns. Crucially, we find that a small number of samples (*e.g.*, $k$=5) is sufficient when we enhance the protocol with our cross-task mixture strategy. This efficient approach allows us to achieve a high level of performance, as indicated by the dotted lines, without the need for extensive sampling. This is a critical advantage for complex tasks where successful reasoning paths are often too scarce to collect in large numbers. More detailed analysis about CRP is shown in Appendix C.5.

**Uniform vs. Weighted Sampling in Phase 3.**   A comparison between *Uniform* and our *Weighted* sampling in Phase 3 demonstrates the effectiveness of the latter. The results, presented in Table 2 (D), indicate that allocating more training focus to tasks the model finds more difficult yields significant performance gains within the same number of training steps. Notably, the score for the challenging Retrosynthesis task improved from 0.33 to 0.39, and the Reagent Selection score rose from 0.63 to 0.69.

**Generalization to Out-of-Distribution Tasks.**   To assess our model's generalization capabilities, we evaluate it on four out-of-distribution (OOD) molecule optimization tasks (Solubility, DRD2, JNK3, and GSK3) from ChemCoTBench (Li et al., 2025a). While standard benchmarks often focus on familiar properties, we tested performance on three more challenging protein activity-related properties. We intentionally select these tasks because they are not part of our training data and differ significantly from our training objectives, providing a robust test of generalization. As shown in Figure 3 (f), the baseline model (Llama-3.1-8B-Instruct) performs poorly, confirming its inability to generalize. In contrast, Chem-R achieves a massive leap in performance across all four tasks, for example, improving the success rate on Solubility from 10% to 83%.

Table 2: **Ablation study** across 9 chemistry tasks (25 sub-tasks). Columns follow the same short-hand as the main table. Gray cells denote Chem-R's performance.

| Ablation Variant | Molecule Tasks | | | | | Reaction Tasks | | | |
|---|---|---|---|---|---|---|---|---|---|
| | Name | Prop. | Design | Capt. | TOMG | Yield | Reag. | React. | Retro |
| *(A) Phase-wise Contributions* | | | | | | | | | |
| Chem-R w/o Phase 1 | 0.14 | 0.87 | 0.20 | 0.34 | 0.38 | 0.87 | 0.58 | 0.54 | 0.34 |
| Chem-R w/o Phase 2 | 0.53 | 0.88 | 0.43 | 0.41 | 0.41 | 0.80 | 0.59 | 0.82 | 0.39 |
| Chem-R w/o Phase 3 | 0.39 | 0.86 | 0.35 | 0.33 | 0.41 | 0.85 | 0.50 | 0.69 | 0.28 |
| Chem-R w/o Phase 1&2 | 0.00 | 0.67 | 0.03 | 0.17 | 0.30 | 0.80 | 0.51 | 0.00 | 0.00 |
| Chem-R w/o Phase 1&3 | 0.10 | 0.85 | 0.27 | 0.27 | 0.37 | 0.84 | 0.55 | 0.48 | 0.19 |
| Chem-R w/o Phase 2&3 | 0.52 | 0.88 | 0.43 | 0.40 | 0.42 | 0.11 | 0.55 | 0.83 | 0.40 |
| Chem-R w/o Phase 1&2&3 | 0.00 | 0.47 | 0.00 | 0.01 | 0.07 | 0.26 | 0.26 | 0.00 | 0.00 |
| *(B) Components of Instantiated Protocol in Phase 2* | | | | | | | | | |
| Phase 2 w/o CRP | 0.31 | 0.84 | 0.27 | 0.33 | 0.42 | 0.69 | 0.43 | 0.67 | 0.20 |
| Phase 2 w/o Correct Information | 0.04 | 0.79 | 0.07 | 0.02 | 0.27 | 0.67 | 0.38 | 0.13 | 0.05 |
| *(C) Single-task vs. Multi-task Training in Phase 2* | | | | | | | | | |
| Single-task | 0.41 | 0.87 | 0.40 | 0.32 | 0.41 | 0.84 | 0.46 | 0.75 | 0.26 |
| Multi-task | 0.39 | 0.86 | 0.35 | 0.33 | 0.41 | 0.85 | 0.50 | 0.69 | 0.28 |
| *(D) Uniform vs. Weighted Sampling in Phase 3* | | | | | | | | | |
| Uniform | 0.43 | 0.87 | 0.38 | 0.39 | 0.41 | 0.84 | 0.63 | 0.81 | 0.33 |
| **Chem-R-8B (Ours)** | **0.49** | **0.87** | **0.42** | **0.41** | **0.42** | **0.85** | **0.69** | **0.82** | **0.39** |

**Human Expert Evaluation.** To qualitatively assess CoTs, we conduct a human expert evaluation. We had chemistry PhDs rate 50 randomly sampled responses from Chem-R and three strong baselines (Gemini-2.5-Pro, DeepSeek-R1, ether0) on a 1-to-5 scale across six dimensions, with a detailed rubric available in the Appendix D. The averaged scores are visualized in a radar chart for comparison, as shown in Figure 4. Chem-R receives the highest scores across all six metrics, from Chemical Soundness and Logical Coherence to Expert-level Insight, validating the effectiveness of our structured reasoning protocol. This comprehensive feedback confirms that Chem-R not only achieves high accuracy but also generates reasoning chains that are significantly more reliable, interpretable, and aligned with expert thinking. This all-around superiority underscores its potential as a more trustworthy and dependable tool for chemical inquiry.

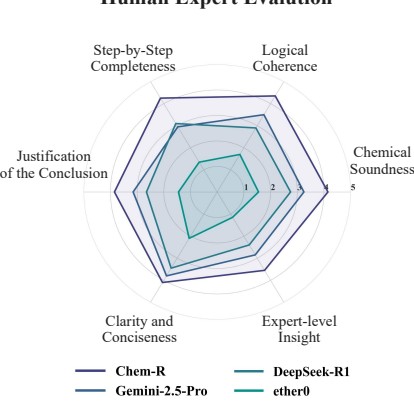

Figure 4: Radar chart of the human expert evaluation.

## 5 CONCLUSION

Chem-R establishes a new state-of-the-art in chemical reasoning by uniquely emulating the deliberative thought processes of expert chemists. Our novel three-phase training framework systematically builds foundational knowledge, distills structured reasoning protocols, and optimizes for balanced performance across a wide array of tasks. This approach enables Chem-R to significantly outperform leading models, including ChemDFM-v1.5-8B and Gemini-2.5-Pro, achieving unprecedented gains, particularly in complex reaction prediction and retrosynthesis tasks. Beyond superior accuracy, Chem-R's primary advantage lies in its ability to generate interpretable, logically coherent, and chemically sound reasoning chains, a quality validated by human expert evaluation. By producing reliable and explainable outputs, Chem-R provides a powerful and trustworthy foundation for accelerating the next generation of AI-driven chemical discovery.

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

# Appendix

## A OVERVIEW OF THE APPENDIX

**Section B (Task Description)** outlines the various tasks used to evaluate the model, which are divided into two main categories. "Molecular Tasks" focus on individual molecules, including challenges like predicting chemical names and properties, designing new molecules, and generating descriptions. "Reaction Tasks" involve chemical transformations, such as predicting reaction yields, selecting reagents, and determining the products or starting materials of a reaction.

**Section C (Implementation Details)** details the technical methodology of the study. It covers the specific hyper-parameters for model training, the data statistics showing how datasets were divided, CRP extraction prompt, and the structured distillation prompts used to guide the model's reasoning process. It also presents an analysis of the Chemical Reasoning Protocols. Additionally, it explains the criteria for the human evaluation performed by chemistry experts to assess the quality of the model's logic.

**Section D (Experiment Result)** presents the quantitative outcomes of the experiments. It contains a series of tables with performance metrics that demonstrate the model's effectiveness on each of the tasks described in Section B and on OOD tasks, providing a detailed, data-driven assessment of its capabilities in chemical reasoning. This section further includes supplementary analyses on the effectiveness of CRP as a prompting strategy, the impact of teacher model quality, and computational efficiency.

**Section E (More Case)** provides additional, specific examples of the model's outputs.

**Section F (Use of LLM)** clarifies the specific role and application of LLMs within this research.

## B TASK DESCRIPTION

In this section, we describe the tasks used in our experiments, which span both molecular tasks and reaction tasks. We selected these tasks from four widely recognized chemistry benchmarks—ChEBI-20, ChemLLMBench, USPTO, and TOMG-Bench—based on their practical relevance and the diversity they offer in evaluating molecular and reaction-level capabilities. These tasks cover a range of applications, from molecule generation and property prediction to reaction prediction and retrosynthesis, providing a comprehensive evaluation of the model's performance in chemical reasoning.

### B.1 MOLECULAR TASKS

**(1) Name Prediction.** In this task, the goal is to convert between *IUPAC names* (International Union of Pure and Applied Chemistry) and *SMILES* (Simplified Molecular Input Line Entry System) *strings*, which are two widely used methods for representing chemical structures. IUPAC names provide a formal, systematic way to describe molecules based on their structure, while SMILES is a textual representation that encodes molecular structure through a series of symbols and characters. Converting between these formats requires reasoning about the chemical structure itself—understanding the arrangement of atoms, bonds, functional groups, and molecular topology. The model must interpret the chemical details embedded in IUPAC names or SMILES strings and produce the corresponding representation, which involves complex reasoning about chemical conventions and rules.

> **Name Prediction** (SMILES→IUPAC)
>
> **Question:** Provide the IUPAC name for the following molecule. SMILES: `CCOC(=O)C(C(C)=O)=C(C)N`.
> **Answer:** `ethyl 2-acetyl-3-aminobut-2-enoate`

**(2) Property Prediction.** Property prediction involves classifying molecules into categories based on their biological activity or toxicity (i.e., *BBBP, HIV, Tox21, ClinTox, BACE*). For this, the model needs to reason about the underlying structure-property relationships. A molecule's structure influences its biological properties through factors like functional group interactions, molecular size, and polarity. The model must learn these complex relationships from data and reason about which aspects of the molecule's structure contribute to its biological effects. This makes property predic-

tion a key task for reasoning models, as they must generalize these chemical insights across diverse molecular structures and predict their effects.

---

**Property Prediction** (BACE)

**Question:** Predict whether the following molecule can inhibit BACE1 (Yes/No). SMILES: `Clc1cc(cnc1)-c1cc2c(CC(CC23N=C(OC3)N)(C)C)cc1`.
**Answer:** *No*

---

**(3) Molecule Design.** In this task, a model is given *a textual description of a molecule* and must generate its *corresponding SMILES representation*. The challenge here is that the model needs to map linguistic descriptions (which are often abstract) to concrete molecular structures. For instance, the description might mention the presence of a functional group, the molecule's size, or other key features, which the model must translate into a valid SMILES string. This task requires the model to reason about the relationship between the described features and how they translate into a molecular structure. It tests the model's ability to use abstract information to generate precise molecular representations, showcasing its reasoning in both language and chemistry.

---

**Molecule Design**

**Question:** Generate a molecule's SMILES string that fits the following description. *Description: The molecule is a member of the class of cyclopentanols carrying 1,2,4-triazol-1-ylmethyl and 4-chlorobenzylidene and geminal dimethyl substituents at positions 1, 2 and 5 respectively. It is a member of triazoles, a member of monochlorobenzenes, a member of cyclopentanols, a tertiary alcohol and an olefinic compound.*
**Answer:** `CC1(C)CC/C(=C\\c2ccc(Cl)cc2)C1(O)Cn1cncn1`

---

**(4) Molecule Captioning.** This task is the inverse problem of molecule design. Given a SMILES string, the model generates a natural language description of the molecule. Here, the model must reason about the structure encoded in the SMILES and generate a coherent description that accurately captures key molecular features, such as functional groups, chemical bonding, and overall molecular properties. The challenge lies in translating the complex, compact SMILES format into a human-readable description that captures both the structure and function of the molecule. It requires the model to interpret a structured representation and reason about how to explain it in a way that makes sense in natural language.

---

**Molecule Captioning**

**Question:** Provide a chemical description for the following molecule. SMILES: `CC1(C)CC/C(=C\\c2ccc(Cl)cc2)C1(O)Cn1cncn1`.
**Answer:** *The molecule is a member of the class of cyclopentanols carrying 1,2,4-triazol-1-ylmethyl and 4-chlorobenzylidene and geminal dimethyl substituents at positions 1, 2 and 5 respectively. It is a member of triazoles, a member of monochlorobenzenes, a member of cyclopentanols, a tertiary alcohol and an olefinic compound.*

---

**(5) Text-based Open Molecule Generation.** The TOMG-Bench benchmark focuses on text-based open molecule generation, evaluating a model's ability to generate, modify, and optimize molecular structures based on textual descriptions or specified criteria. Tasks in this benchmark include Customized Molecule Generation, Molecule Editing, and Molecule Optimization. In *Customized Molecule Generation*, the model is tasked with creating molecules that meet specific constraints, such as a predefined number of atoms, bonds, or functional groups, while maintaining chemical validity. *Molecule Editing* requires the model to modify an existing molecule by adding, replacing, or removing functional groups, with the challenge of reasoning about how these changes affect the overall structure and properties of the molecule. *Molecule Optimization* involves optimizing molecules to improve specific properties like LogP (partition coefficient), QED (drug-likeness), and MR (molecular refractivity), where the model must navigate trade-offs between conflicting goals, such as balancing hydrophobicity to improve LogP without compromising QED. Together, these

tasks test a model's ability to generate, edit, and optimize molecules, requiring reasoning about molecular structure, function, and the interdependencies between chemical properties.

---

**Text-based Open Molecule Generation** (MolOpt-LogP)

**Question:** Modify the molecule
`N#CCC1(n2cc(-c3ccnc4[nH]ccc34)cn2)C[NH+](C2CCN(C(=O)Nc3ccccc3Cl)CC2)C1` to have a lower LogP value.
**Answer:** `O=C(Nc1ccccc1Cl)N1CCC([NH+]2CC(CO)(n3cc(-c4ccnc5[nH]ccc45)cn3)C2)CC1`

---

## B.2   REACTION TASKS

**(6) Yield Prediction.**   Yield prediction involves determining whether a given chemical reaction will result in *a high or low yield* based on the reactants and reaction conditions in the *Buchwald-Hartwig and Suzuki-coupling reactions*. Here, reasoning is necessary because predicting the yield requires the model to understand both the intrinsic properties of the reactants and the external factors that can influence the efficiency of the reaction. It requires the model to simulate the chemical behavior of the system, predict potential losses, and estimate the likelihood of a successful reaction based on prior examples. This is a classic task of predicting outcomes under uncertainty, demanding robust reasoning capabilities to account for various complex variables.

---

**Yield Prediction** (Buchwald-Hartwig reaction)

**Question:** Predict if the following Buchwald-Hartwig reaction is high-yielding (Yes for >70% yield, No otherwise). Reaction:
`Brc1cccnc1.Cc1ccc(N)cc1.`
`O=S(=O)(O[Pd]1c2ccccc2-c2ccccc2N-1)C(F)(F)F.`
`COc1ccc(OC)c(P([C@]23C[C@H]4C[C@H](C[C@H](C4)C2)C3)[C@]23C[C@H]4C[C@H](C[C@H](C4)C2)C3)c1-c1c(C(C)C)cc(C(C)C)cc1C(C)C.`
`CCN=P(N=P(N(C)C)(N(C)C)N(C)C)(N(C)C)N(C)C.`
`CCOC(=O)c1cc(OC)no1>>Cc1ccc(Nc2cccnc2)cc1.`
**Answer:** *No*

---

**(7) Reagent Selection.**   This task involves selecting the appropriate *reagents (reactants, solvents, and ligands)* from a predefined list for a given reaction. Reasoning is critical here, as the model must understand the chemical context of the reaction and predict which reagents will interact most effectively to drive the desired transformation. Importantly, we choose this task over USPTO-Condition because, in Reagent Selection, *each option comes with an associated yield value, making the reasoning more concrete and verifiable*. Additionally, this task more closely mimics real-world chemical practices, where chemists have to select reagents from a limited set of available chemicals, often due to budget, availability, or experimental constraints. In contrast, USPTO-Condition involves broader, less constrained reaction conditions that may not align with practical laboratory limitations, and its accuracy is harder to verify because it lacks specific yield values and focuses on a wider range of conditions. Thus, Reagent Selection provides a more focused and realistic task, better suited to evaluating a model's ability to reason within the practical boundaries of chemical experimentation.

**Reagent Selection**

**Question:** From the provided list, select the optimal reactant to maximize the yield for the following reaction setup.
Reaction Setup:
reactant: `Ic1ccc2ncccc2c1`
catalyst: `CC(=O)[O-].CC(=O)[O-].[Pd+2]`
ligand: none
reagent: `O=P([O-])([O-])[O-].[K+].[K+].[K+]`
solvent: `C1CCOC1`
list of reactants for selection: `['B(O)O',`
`'Cc1ccc2c(cnn2C2CCCCO2)c1B1OC(C)(C)C(C)(C)O1',`
`'Cc1ccc2c(cnn2C2CCCCO2)c1[B-](F)(F)F.[K+]']`
**Answer:** `Cc1ccc2c(cnn2C2CCCCO2)c1B1OC(C)(C)C(C)(C)O1`

**(8) Reaction Prediction.** This task requires *predicting the products of a chemical reaction* based on the given reactants and reaction conditions. Reasoning is essential here because the model needs to understand the underlying chemistry, such as functional group reactivity, reaction mechanisms, and stereochemistry, in order to predict the correct products. Unlike simpler tasks that only involve pattern recognition, this task demands an ability to apply chemical principles (like how certain bonds break and form) and anticipate the reaction's outcome, which requires sophisticated reasoning beyond just memorization.

**Reaction Prediction**

**Question:** Predict the main product(s) for the following reaction. Reactants:
`CN(C)C=O.COc1cccc(N)c1.Cc1nc(CCl)cs1.Cl.O=C([O-])[O-].[K+].[K+]`
**Answer:** `COc1cccc(NCc2csc(C)n2)c1`

**(9) Retrosynthesis.** Retrosynthesis involves predicting the *starting materials (reactants)* required to synthesize a given target molecule. To perform reasoning, the model needs to deconstruct the target molecule into simpler components and reverse-engineer the chemical process. This requires understanding reaction pathways, identifying suitable reactions to break bonds, and selecting the appropriate reagents. It's a form of reverse reasoning, where the model must consider multiple potential pathways and choose the one that is most likely to lead to the desired product, based on its chemical structure and reactivity.

**Retrosynthesis**

**Question:** Predict the necessary reactant(s) for the following product. Product:
`CNC(=O)c1c(-c2ccc(F)cc2)oc2cc(N(C)S(C)(=O)=O)c(-c3cccc([N+]`
`(=O)[O-])c3)cc12`
**Answer:** `CNC(=O)c1c(-c2ccc(F)cc2)oc2cc(N(C)S(C)(=O)=O)c(Br)cc12.`
`O=[N+]([O-])c1cccc(B(O)O)c1`

# C IMPLEMENTATION DETAILS

## C.1 HYPER-PARAMETERS

In this section, we list the hyper-parameters used in different phases of training and inference. We used Llama-Factory to conduct SFT training (Phase 1 and Phase 2), and EasyR1 for GRPO training (Phase 3). Table 3 provides the values for the hyper-parameters in Phase 1, Phase 2, and Phase 3, as well as for inference. The table includes settings such as the number of GPUs, learning rates, batch sizes, and the number of epochs for each phase.

| Item | Value |
|---|---|
| *Phase 1* | |
| gpu_number (H100) | 2 |
| per_device_train_batch_size | 1 |
| gradient_accumulation_steps | 4 |
| learning_rate | 1.0e-5 |
| num_train_epochs | 5 |
| lr_scheduler_type | cosine |
| warmup_ratio | 0.1 |
| epoch | 3 |
| *Phase 2* | |
| gpu_number (H100) | 2 |
| per_device_train_batch_size | 1 |
| gradient_accumulation_steps | 4 |
| learning_rate | 1.0e-5 |
| num_train_epochs | 5 |
| lr_scheduler_type | cosine |
| warmup_ratio | 0.1 |
| epoch | 5 |

| Item | Value |
|---|---|
| *Phase 3* | |
| gpu_number (H100) | 8 |
| learning_rate | 1.0e-6 |
| weight_decay | 1.0e-2 |
| kl_coef | 1.0e-2 |
| n | 5 |
| rollout.temperature | 1.0 |
| global_batch_size | 128 |
| rollout_batch_size | 512 |
| micro_batch_size_per_device_for_update | 4 |
| epoch | 3 |
| step | 683 |
| *Inference* | |
| temperature | 0.6 |
| top_p | 0.9 |
| max_tokens | 4096 |

Table 3: Hyper-parameters for Different Phases of Training and Inference

## C.2 DATA STATISTICS

In this section, we outline the data splits used across different phases of training. The data partitioning follows the benchmark division strategies, ensuring that the test set is consistent with the evaluation criteria and standards. The amount of training data is strategically varied across the phases to match their distinct objectives:

- **Phase 1 (Chemical Foundation Training)**: The goal is to build a comprehensive foundation of chemical knowledge. Therefore, this phase utilizes a *large volume* of question-answer pairs (e.g., 920k for Name Prediction) to ensure broad exposure to facts and patterns.
- **Phase 2 (CRP Distillation)**: This phase focuses on teaching a structured reasoning method using high-quality synthetic CoT data. The strategy here is to provide a substantial and relatively balanced number of examples across different task categories, *generally targeting around 100k samples per major task*. For tasks identified as particularly difficult, such as Name Prediction, we ensure a higher volume of data to help the model master their complex reasoning protocols.
- **Phase 3 (Multi-task GRPO)**: The objective is to refine the model's reasoning skills. For this targeted alignment, the amount of training data for each task is not fixed but is calculated based on the model's performance after Phase 2. This strategy, detailed in Section 3.3, allows us to concentrate the training effort on tasks the model finds more difficult, optimizing the refinement process.

Importantly, we ensure that no molecules appearing in the test set are included in the training sets. For a detailed overview of the data splits and their distribution, please refer to Table 4. It is particularly noteworthy that the TOMG tasks were intentionally excluded from the GRPO phase (Phase 3) due to their long evaluation times and better performance.

| Tasks | Phase 1 | Phase 2 | Phase 3 | Train | Valid | Test |
|---|---|---|---|---|---|---|
| *Name Prediction* | | | | | | |
| SMILES2IUPAC | 920,734 | 100,000 | 6,978 | 828,661 | 500 | 100 |
| IUPAC2SMILES | 920,734 | 100,000 | 7,821 | 828,661 | 500 | 100 |
| *Property Prediction* | | | | | | |
| BACE | 1,413 | 20,000 | 3,489 | 1,413 | 50 | 100 |
| BBBP | 1,950 | 20,000 | 2,286 | 1,950 | 50 | 100 |
| ClinTox | 1,384 | 20,000 | 241 | 1,384 | 50 | 100 |
| HIV | 41,027 | 20,000 | 0 | 41,027 | 50 | 100 |
| Tox21 | 7914 | 20,000 | 2,166 | 7,914 | 50 | 100 |
| *Molecule Design* | | | | | | |
| ChEBI-20 | 26,407 | 100,000 | 7,821 | 26,407 | 3,300 | 3,300 |
| *Molecule Captioning* | | | | | | |
| ChEBI-20 | 26,407 | 100,000 | 7,008 | 26,407 | 3,300 | 3,300 |
| *Text-based Open Molecule Generation* | | | | | | |
| MolCustom-AtomNum | 133,334 | 33,333 | – | 133,334 | – | 5,000 |
| MolCustom-BondNum | 133,334 | 33,333 | – | 133,334 | – | 5,000 |
| MolCustom-FunctionalGroup | 133,334 | 33,333 | – | 133,334 | – | 5,000 |
| MolEdit-AddComponent | 133,333 | 33,333 | – | 133,333 | – | 5,000 |
| MolEdit-DelComponent | 133,333 | 33,333 | – | 133,333 | – | 5,000 |
| MolEdit-SubComponent | 133,333 | 33,333 | – | 133,333 | – | 5,000 |
| MolOpt-LogP | 133,333 | 33,333 | – | 133,333 | – | 5,000 |
| MolOpt-MR | 133,333 | 33,333 | – | 133,333 | – | 5,000 |
| MolOpt-QED | 133,333 | 33,333 | – | 133,333 | – | 5,000 |
| *Yield Prediction* | | | | | | |
| Buchwald-Hartwig | 3,855 | 40,515 | 1,925 | 3,855 | 50 | 100 |
| Suzuki-Miyaura | 5,660 | 58,485 | 1,685 | 5,660 | 50 | 100 |
| *Reagent Selection* | | | | | | |
| Reactant Selection | 1,436 | 44,763 | 5,174 | 1,436 | 100 | 100 |
| Solvent Selection | 1,340 | 41,770 | 6,497 | 1,340 | 100 | 100 |
| Ligand Selection | 380 | 13,467 | 6,517 | 380 | 100 | 100 |
| *Reaction Prediction* | | | | | | |
| USPTO-Mixed | 409,035 | 100,000 | 3,730 | 409,035 | 30,000 | 100 |
| *Retrosynthesis* | | | | | | |
| USPTO-50k | 40,029 | 100,000 | 8,663 | 40,029 | 5,004 | 100 |

Table 4: Tasks and data splits across different phases. *Note* that the quantities listed for Phases 1, 2, and 3 refer to the total data volume, inclusive of any repeated samples.

## C.3 CRP EXTRACTION PROMPT

Prior to generating the protocol, we establish a reference dataset by sampling the teacher model with a simple, generic task prompt. We evaluate the generated responses against ground truth to categorize them into two groups: **positive examples**, consisting of the input question and the correct reasoning trace, and **negative examples**, which comprise the question, the flawed reasoning, and the ground truth for correction. We then feed these curated examples into the CRP Extraction Prompt (presented below), enabling the teacher model to synthesize a structured Chemical Reasoning Protocol defined by reasonable logical steps and hints for common mistakes.

---

**CRP Extraction Prompt**

You are an expert Chemical Methodology Analyst and Pedagogue. Your mission is to analyze raw reasoning traces from specific chemical tasks and synthesize a robust, standardized *Chemical Reasoning Protocol (CRP)*. A CRP is a structured, algorithmic thinking blueprint designed to emulate the deliberate decision-making process of a professional chemist. It serves as a generalizable scaffold that breaks down complex chemical problems into a defined sequence of logical steps (derived from successful reasoning), while simultaneously integrating specific "Cautionary Guidance" (derived from failed reasoning) to preemptively block common hallucinations, syntactic errors, and logical fallacies.

**1. Task Context**
    **Task Name:** {{TASK_NAME}}
    **Description:** {{TASK_DESCRIPTION}}

**2. Input Data**
I will provide two sets of reasoning trajectories: (1) Positive Examples: 5 reasoning paths that logically lead to the correct answer. (2) Negative Examples: 5 reasoning paths that lead to incorrect answers or contain hallucinations.
    **Positive Examples:** {{POSITIVE_EXAMPLES}}
    **Negative Examples:** {{NEGATIVE_EXAMPLES}}

**3. Analytical Objectives**
Analyze these examples to construct a Generalizable Chemical Reasoning Protocol.
- **Synthesize Logical Steps:** Examine the **Positive Examples**. Abstract the common, underlying logical structure into 3-5 distinct steps (e.g., "Analyze Reactants", "Deduce Mechanism").
- **Extract Cautionary Hints:** Examine the **Negative Examples**. Identify the root causes of failure (e.g., ignoring stereochemistry, misidentifying functional groups). Convert these errors into *Cautionary Guidance* for the corresponding steps.
- **Generalization:** The protocol must be universal for this task type, avoiding references to specific molecules in the examples.

**4. Output Format**
Please output the protocol strictly following the Markdown structure below. Do not include any other introductory text.
1. **[Step 1 Title]**: [Detailed Description of what to do]
*\* [Cautionary hint derived from negative examples]*
*\* [Strategic hint derived from positive examples]*
2. **[Step 2 Title]**: [Detailed Description of what to do]
*\* [Cautionary hint...]*
*...*
*(Continue for all necessary steps)*

---

1242
1243
1244
1245
1246

C.4 DISTILLATION PROMPTS

1247
1248
1249
1250
1251
1252
1253
1254
1255
1256
1257
1258
1259
1260
1261
1262
1263
1264
1265
1266
1267
1268
1269
1270
1271
1272
1273
1274
1275
1276
1277
1278
1279
1280
1281
1282
1283
1284
1285
1286
1287
1288
1289
1290
1291
1292
1293
1294
1295

---

**Name Prediction** (SMILES→IUPAC)

You are an expert chemist demonstrating how to determine the IUPAC name for a given molecule.

You are given:
1. A SMILES string representing the molecule.
2. A list of functional groups present in the molecule (with representative SMILES patterns) to serve as hints.

Your task is to generate a step-by-step reasoning process that logically deduces the systematic IUPAC name from the SMILES string. You MUST NOT mention ground-truth, given reactants, provided answer, or any similar phrases in your `<think>` block. You must write the reasoning as if you are deducing the answer from scratch, even though you already know the destination.

Please structure your reasoning as follows:
1. **Deconstruct & Analyze**: Thoroughly parse the SMILES string to visualize the molecular graph, identifying all functional groups, cyclic systems, points of unsaturation, and any specified stereocenters.
* *Translate the linear SMILES notation into a clear mental or physical representation of the molecular structure.*
* *Systematically identify every functional group and ring, paying close attention to IUPAC priority.*
* *Note all stereochemical indicators (@, @@, /, \\) for inclusion in the final name.*
2. **Determine the Principal Chain or Parent Ring**: Apply IUPAC priority rules to identify the principal functional group, which in turn determines the parent structure (the longest chain or main ring system) and the suffix of the name.
* *The parent structure must contain the highest-priority functional group.*
* *If a choice exists, the parent is the structure with the most multiple bonds, then the longest carbon chain.*
3. **Number & Name the Parts**: Systematically number the atoms of the parent structure to assign the lowest possible locant to the principal functional group, then identify and name all substituent groups attached to the parent.
* *The numbering must grant the lowest locant to the principal functional group above all other considerations.*
* *If a choice remains, assign lowest locants to multiple bonds, then to the first point of difference in substituent locants.*
* *Correctly name each substituent (e.g., "methyl", "chloro") and note its locant.*
4. **Assemble & Order**: Construct the complete IUPAC name by arranging the named substituents alphabetically, prefixing them to the parent name, and incorporating all locants and stereochemical descriptors in their correct positions.
* *Alphabetize substituents by name, ignoring multiplying prefixes (di, tri) but not iso or neo.*
* *Assemble the final name in the standard order: Stereochemistry-Substituents-Parent-Suffix.*

---

**Input:**
SMILES: {instruction}
FunctionalGroups: {functional_groups_str}
Ground-truth IUPAC Name: {target}

---

**Output:**
`<think>`
1. **Deconstruct & Analyze**: ...
2. **Determine the Principal Chain or Parent Ring**: ...
3. **Number & Name the Parts**: ...
4. **Assemble & Order**: ...
`</think>`
`<answer>`{target}`</answer>`

**Name Prediction** (IUPAC→SMILES)

You are an expert chemist demonstrating how to determine the IUPAC name for a given molecule.

You are given:

1. An IUPAC name representing the molecule.

2. A list of functional groups present in the molecule (with representative SMILES patterns) to serve as hints.

Your task is to generate a step-by-step reasoning process that logically deduces the SMILES string from the IUPAC name. You MUST NOT mention ground-truth, given reactants, provided answer, or any similar phrases in your `<think>` block. You must write the reasoning as if you are deducing the answer from scratch, even though you already know the destination.

Please structure your reasoning as follows:

1. **Deconstruct IUPAC Name & Identify Components**: Break down the IUPAC name into its fundamental components, identifying the parent structure (chain or ring), principal functional group, all substituents, and their corresponding locants.

* *The parent name (e.g., "hexane", "cyclohexane") forms the SMILES backbone.*

* *Pay close attention to suffixes (-ol, -one, -oic acid) as they define the principal group.*

* *Isolate all stereochemical prefixes (R/S, E/Z, cis/trans) for later use.*

2. **Formulate Conceptual Structure**: Translate the deconstructed components into a clear, verbal description of the molecule's connectivity, specifying how each substituent attaches to the parent structure at its designated locant.

* *This step acts as a "blueprint" before writing any SMILES code.*

* *Mentally or explicitly number the parent chain to map out attachment points.*

* *Clarify the precise atom-to-atom connections for all parts.*

3. **Attempt SMILES Translation**: Individually convert each identified component—the parent structure, substituents, and functional groups—into its correct SMILES representation, treating them as separate fragments.

* *Use parentheses () for branches off the main chain.*

* *Represent double and triple bonds with '=' and '#', respectively.*

* *Keep track of numeric labels for defining ring closures.*

4. **Construct Final SMILES String**: Systematically assemble the individual SMILES fragments into a single, valid string, ensuring correct connectivity, branching, ring closures, and the inclusion of stereochemical markers.

* *Start with the fragment for the principal functional group or a logical endpoint of the main chain.*

* *Insert substituent fragments inside parentheses at the correct atom of the main chain.*

* *Apply stereochemical symbols (@, @@, /, \\) last to ensure correct placement relative to the final structure.*

**Input:**

IUPAC: {instruction}

FunctionalGroups: {functional_groups_str}

Ground-truth SMILES Name: {target}

**Output:**

`<think>`

1. **Deconstruct Name & Identify Components**: ...

2. **Formulate Conceptual Structure**: ...

3. **Translate Components to SMILES**: ...

4. **Assemble & Finalize String**: ...

`</think>`

`<answer>`{target}`</answer>`

**Property Prediction** (BACE1)

You are an expert chemist demonstrating how to predict the BACE1 inhibition potential of a molecule from its chemical structure.

You are given:

1. A SMILES string representing the molecule.

2. A list of functional groups present in the molecule (with representative SMILES patterns) to serve as hints.

Your task is to generate a step-by-step reasoning process that logically deduces whether the molecule can inhibit (Yes) or cannot inhibit (No) the Beta-site Amyloid Precursor Protein Cleaving Enzyme 1 (BACE1). You MUST NOT mention "ground-truth", "given reactants", "provided answer", or any similar phrases in your `<think>` block. You must write the reasoning as if you are deducing the answer from scratch, even though you already know the destination.

Please structure your reasoning as follows:

1. **Structural/Property Identification**: Analyze the SMILES to identify key pharmacophores, functional groups, and physicochemical properties relevant for BACE1 inhibition.

* *Identify key pharmacophoric features (H-bond donors/acceptors, aromatic rings).*

* *Assess general drug-like properties such as MW, flexibility, and logP.*

* *Check for structural motifs common to known inhibitors of this target class.*

2. **Property-Activity Correlation**: Correlate the identified features with general Structure-Activity Relationships (SAR) for BACE1 inhibition.

* *Assess how functional groups might form hydrogen bonds or electrostatic interactions in an active site.*

* *Evaluate if hydrophobic regions are suitably sized for binding within lipophilic pockets.*

* *Align the molecule's properties with general drug-likeness principles (e.g., Lipinski's Rules).*

3. **Holistic Assessment & Consideration**: Synthesize all positive and negative factors to form a holistic judgment of the molecule's inhibitory potential.

* *Weigh the combined evidence for and against binding, not just a simple count of features.*

* *Consider potential liabilities that could prevent binding, such as steric hindrance or unfavorable charges.*

4. **Final Prediction Verdict**: Conclude with a definitive "Yes" or "No" prediction based on the integrated assessment.

* *The final verdict must be a direct logical consequence of the previous assessment.*

* *State the primary molecular features that drive the final "Yes" or "No" decision.*

Wrap your entire reasoning process in `<think>` tags and output only the final answer (Yes or No) in `<answer>` tags.

**Input:**

SMILES: {instruction}

FunctionalGroups: {functional_groups_str}

Ground-truth Answer: {target}

**Output:**

`<think>`

1. **Structural/Property Identification**: ...

2. **Property-Activity Correlation**: ...

3. **Holistic Assessment & Consideration**: ...

4. **Final Prediction Verdict**: ...

`</think>`

`<answer>`{target}`</answer>`

**Property Prediction** (BBBP)

You are an expert chemist demonstrating how to predict the Blood-Brain Barrier (BBB) penetration potential of a molecule from its chemical structure.

You are given:
1. A SMILES string representing the molecule.
2. A list of functional groups present in the molecule (with representative SMILES patterns) to serve as hints.

Your task is to generate a step-by-step reasoning process to predict the binary label (Yes for penetration, No for non-penetration) for the given molecule's ability to cross the Blood-Brain Barrier. You MUST NOT mention "ground-truth", "given reactants", "provided answer", or any similar phrases in your `<think>` block. You must write the reasoning as if you are deducing the answer from scratch, even though you already know the destination.

Please structure your reasoning as follows:
1. **Structural/Property Identification**: Identify key physicochemical properties from the SMILES that govern membrane permeability.
* *Estimate properties like molecular weight (MW), lipophilicity (logP), and polar surface area (TPSA).*
* *Count hydrogen bond donors/acceptors and identify any permanent charges.*
2. **Property-Activity Correlation**: Correlate the identified properties with principles favoring or hindering BBB penetration.
* *Assess if size, polarity, and lipophilicity fall within a favorable range for passive diffusion.*
* *Recognize that high polarity or a permanent charge strongly hinders penetration.*
3. **Holistic Assessment & Consideration**: Synthesize all factors to form a balanced judgment on the molecule's overall profile for BBB penetration.
* *Weigh the combined impact of all properties; a single critical flaw can prevent penetration.*
* *Consider the possibility of active efflux mechanisms (e.g., P-gp substrate).*
4. **Final Prediction Verdict**: Conclude with a definitive "Yes" or "No" prediction based on the holistic assessment.
* *The final verdict must be a logical conclusion of the prior analysis.*
* *State the most influential properties driving the decision.*

Wrap your entire reasoning process in `<think>` tags and output only the final answer (Yes or No) in `<answer>` tags.

**Input:**
SMILES: {instruction}
FunctionalGroups: {functional_groups_str}
Ground-truth Answer: {target}

**Output:**
```
<think>
1. Structural/Property Identification: ...
2. Property-Activity Correlation: ...
3. Holistic Assessment & Consideration: ...
4. Final Prediction Verdict: ...
</think>
<answer>{target}</answer>
```

**Property Prediction** (ClinTox)

You are an expert chemist demonstrating how to predict the clinical trial toxicity of a molecule from its chemical structure.

You are given:
1. A SMILES string representing the molecule.
2. A list of functional groups present in the molecule (with representative SMILES patterns) to serve as hints.
3. The FDA approval status for clinical trials.

Your task is to generate a a step-by-step reasoning process to predict whether a molecule is Clinically-trial-Toxic (Yes) or Not Clinically-trial-toxic (No). The FDA-approved status is provided as an additional piece of evidence. You MUST NOT mention "ground-truth", "given reactants", "provided answer", or any similar phrases in your `<think>` block. You must write the reasoning as if you are deducing the answer from scratch, even though you already know the destination.

Please structure your reasoning as follows:
1. **Structural/Property Identification**: Identify structural alerts, reactive functional groups, and other properties linked to toxicity from the SMILES.
* *Identify known toxicophores and chemically reactive motifs.*
* *Assess if extreme physicochemical properties (e.g., high MW or logP) increase risk.*
2. **Property-Activity Correlation**: Correlate the identified structural risks with general toxicological principles.
* *Assess how reactive groups could potentially cause organ damage.*
* *Note that a structural alert suggests risk but is not definitive proof of toxicity.*
3. **Holistic Assessment & Consideration**: Synthesize the structural risk analysis with the provided FDA approval status to form a comprehensive judgment.
* *Use the FDA approval status as powerful evidence to adjust the initial structural risk assessment.*
* *An "Approved" status implies any identified structural risks were clinically acceptable.*
4. **Final Prediction Verdict**: Conclude with a definitive "Yes" or "No" prediction based on the integrated evidence.
* *The verdict must balance the structural analysis against the FDA approval status.*
* *State the primary factors that drive the final "Yes" or "No" decision.*

Wrap your entire reasoning process in `<think>` tags and output only the final answer (Yes or No) in `<answer>` tags.

**Input:**
SMILES: {instruction}
FunctionalGroups: {functional_groups_str}
FDA_APPROVED: {FDA_APPROVED}
Ground-truth Answer: {target}

**Output:**
```
<think>
1. Structural/Property Identification: ...
2. Property-Activity Correlation: ...
3. Holistic Assessment & Consideration: ...
4. Final Prediction Verdict: ...
</think>
<answer>{target}</answer>
```

**Property Prediction** (HIV)

You are an expert chemist demonstrating how to predict the HIV replication inhibition potential of a molecule from its chemical structure.

You are given:
1. A SMILES string representing the molecule.
2. A list of functional groups present in the molecule (with representative SMILES patterns) to serve as hints.
3. The result from an activity test, classified as CA (confirmed active), CM (confirmed moderately active), or CI (confirmed inactive).

Your task is to generate a step-by-step reasoning process to predict the binary label for the molecule's ability to inhibit HIV replication (Yes or No), considering its structure and the provided activity test result. You MUST NOT mention "ground-truth", "given reactants", "provided answer", or any similar phrases in your `<think>` block. You must write the reasoning as if you are deducing the answer from scratch, even though you already know the destination.

Please structure your reasoning as follows:
1. **Structural/Property Identification**: Analyze the SMILES for structural motifs and properties linked to antiretroviral activity.
* *Identify scaffolds similar to known HIV inhibitor classes (e.g., protease inhibitors).*
* *Assess functional groups for key interactions (e.g., H-bonding) with viral targets.*
* *Evaluate general properties like size and polarity relevant for cellular uptake.*
2. **Property-Activity Correlation**: Correlate the identified features with general mechanisms of HIV inhibition.
* *Assess the molecule's potential to mimic the natural substrates of viral enzymes.*
* *Consider if the molecule's geometry is suitable for binding to key viral targets.*
3. **Holistic Assessment & Consideration**: Synthesize the structural analysis with the provided activity data to form a comprehensive judgment.
* *Use the activity result (CA/CM/CI) as key evidence to confirm or override the structural analysis.*
* *A "CA" or "CM" result strongly supports a "Yes" prediction, even with subtle structural cues.*
* *A "CI" result strongly supports a "No" prediction, regardless of favorable features.*
4. **Final Prediction Verdict**: Conclude with a definitive "Yes" or "No" prediction based on the integrated evidence.
* *The verdict must clearly integrate the structural analysis with the activity data.*
* *State the primary factors (e.g., "Structurally promising and confirmed by 'CA' result") driving the decision.*
* *The final verdict must explicitly state how the structural analysis and the activity data were combined.*
* *Summarize the key factors (e.g., "Structurally promising and confirmed active by 'CA' result") driving the decision.*

Wrap your entire reasoning process in `<think>` tags and output only the final answer (Yes or No) in `<answer>` tags.

**Input:**
SMILES: {instruction}
FunctionalGroups: {functional_groups_str}
activity: {activity}
Ground-truth Answer: {target}

**Output:**
```
<think>
1. Structural/Property Identification: ...
2. Property-Activity Correlation: ...
3. Holistic Assessment & Consideration: ...
4. Final Prediction Verdict: ...
</think>
<answer>{target}</answer>
```

**Property Prediction** (Tox21)

You are an expert chemist demonstrating how to predict the toxicity of a molecule from its chemical structure.

You are given:
1. A SMILES string representing the molecule.
2. A list of functional groups present in the molecule (with representative SMILES patterns) to serve as hints.

Your task is to generate a step-by-step reasoning process to predict whether a molecule is toxic (Yes) or not toxic (No). You MUST NOT mention "ground-truth", "given reactants", "provided answer", or any similar phrases in your `<think>` block. You must write the reasoning as if you are deducing the answer from scratch, even though you already know the destination.

Please structure your reasoning as follows:
1. **Structural/Property Identification**: Analyze the SMILES string for known toxicophores, structural alerts, and chemically reactive functional groups.
* *Identify highly reactive motifs (e.g., Michael acceptors, epoxides) or strained rings.*
* *Search for groups that can be metabolically activated into toxic species (e.g., nitroaromatics).*
* *Assess if extreme physicochemical properties (e.g., high logP) suggest non-specific toxicity.*
2. **Property-Activity Correlation**: Correlate the identified structural features with common mechanisms of chemical toxicity.
* *Evaluate the potential for reactive groups to cause cellular damage via covalent modification of proteins or DNA.*
* *Consider the possibility of off-target pharmacology based on the molecule's overall profile.*
* *Recognize that a structural alert increases toxicity risk but is not definitive proof.*
3. **Holistic Assessment & Consideration**: Synthesize all structural risk factors to form a comprehensive judgment on the molecule's likely toxicity.
* *Weigh the combined impact of all identified risks; the presence of multiple alerts is a strong indicator of toxicity.*
* *Consider if the overall molecular context might mitigate the reactivity of a potential toxicophore (e.g., via steric hindrance).*
4. **Final Prediction Verdict**: Conclude with a definitive "Yes" (toxic) or "No" (not toxic) prediction based on the overall risk assessment.
* *The verdict must be a logical conclusion derived from the balance of identified structural risks.*
* *State the primary structural reasons that drive the final "Yes" or "No" decision.*

Wrap your entire reasoning process in `<think>` tags and output only the final answer (Yes or No) in `<answer>` tags.

**Input:**
SMILES: {instruction}
FunctionalGroups: {functional_groups_str}
Ground-truth Answer: {target}

**Output:**
```
<think>
```
1. **Structural/Property Identification**: ...
2. **Property-Activity Correlation**: ...
3. **Holistic Assessment & Consideration**: ...
4. **Final Prediction Verdict**: ...
```
</think>
<answer>{target}</answer>
```

**Molecule Design** (Text→SMILES)

You are an expert chemist demonstrating how to solve a molecule design problem.

You are given:
1. A molecular description.
2. A list of functional groups present in the molecule (with representative SMILES patterns).
Your task is to generate a step-by-step reasoning process that logically deduces the SMILES from the description and the functional groups. You MUST NOT mention "ground-truth", "given reactants", "provided answer", or any similar phrases in your `<think>` block. You must write the reasoning as if you are deducing the answer from scratch, even though you already know the destination.

Please structure your reasoning as follows:
1. **Identify the core structure**: Determine the central molecular scaffold or backbone by interpreting the root name in the chemical description.
* *Focus on identifying the parent compound, such as a main chain ('hexane') or a ring system ('benzene').*
* *This core structure will serve as the primary chain for the SMILES string.*
2. **Summarize key elements**: Systematically list all substituents and functional groups from the description and identify their attachment points (locants) on the core.
* *Create a clear connectivity map, linking each group to its specific locant on the parent structure.*
* *Use the provided functional group list to confirm the identity of complex groups.*
3. **Address stereochemistry**: Scan the description for any stereochemical descriptors and determine the specific configuration at each chiral center or double bond.
* *Identify specific prefixes such as (R)/(S) for chiral centers and (E)/(Z) for double bonds.*
* *If no stereochemistry is mentioned, explicitly note this fact.*
4. **Construct SMILES**: Build the final SMILES string by translating the conceptual blueprint into the correct syntax, combining the core, substituents, and stereochemistry.
* *Use parentheses () for branches and numeric labels for ring closures.*
* *Apply stereochemical markers (@, @@, /, \\) at their correct positions after establishing atom connectivity.*
Wrap your reasoning in `<think>` tags and output the final SMILES in `<answer>` tags.

**Input:**
Description: {instruction}
FunctionalGroups: {functional_groups_str}
Ground-truth SMILES: {target}

**Output:**
```
<think>
```
1. **Identify the core structure**: ...
2. **Summarize key elements**: ...
3. **Address stereochemistry**: ...
4. **Construct SMILES**: ...
```
</think>
<answer>{target}</answer>
```

**Molecule Captioning** (SMILES→Text)

You are an expert chemist demonstrating how to describe a molecule based on its structure.

You are given:
1. A SMILES string.
2. A list of functional groups present in the molecule (with representative SMILES patterns). Your task is to generate a step-by-step reasoning process that logically deduces a detailed chemical description from the SMILES. You MUST NOT mention "ground-truth", "given reactants", "provided answer", or any similar phrases in your `<think>` block. You must write the reasoning as if you are deducing the answer from scratch, even though you already know the destination.

Please structure your reasoning as follows:
1. **Analyze Molecular Structure**: Deconstruct the SMILES string by identifying the primary carbon backbone with its branching points, systematically locating all heteroatoms and specified functional groups, and characterizing any cyclic systems found via ring-closure numerals.
* *Account for all specified functional groups without omission.*
* *Note aromaticity from lowercase letters to define the core scaffold.*
2. **Summarize Key Elements**: Integrate the individual components into a cohesive overview to characterize the core scaffold, such as an aliphatic chain or aromatic system, and determine the principal chemical class or classes to which the molecule belongs based on the combination of functional groups.
* *Synthesize findings into a chemical class (e.g., "a substituted ester"), not a simple list of parts.*
* *Infer the molecule's dominant chemical character from its functional groups.*
* *Consider the relative positions of substituents on the core structure.*
3. **Address Stereochemistry**: Scrutinize the SMILES string for stereochemical indicators, specifically '@' and '@@' for tetrahedral chirality and '/' and '\' for geometric isomerism, to explicitly determine the R/S or E/Z configuration for each identified stereocenter or double bond.
* *Explicitly state if stereochemistry is absent or unspecified.*
* *Link each R/S or E/Z descriptor to its precise atom or bond.*
4. **Predict Molecular Description**: Construct a final, detailed chemical description by systematically assembling the preceding findings, starting with the core structure before adding the names and locations of all functional groups and incorporating stereochemical descriptors to form a complete narrative.
* *The final description must be a cohesive, flowing narrative.*
* *Ensure all identified structural and stereochemical features are included.*
* *Avoid a simple list of facts; the summary should be a comprehensive chemical story.*
Wrap your complete reasoning process in `<think>` tags and output the final description in `<answer>` tags.

**Input:**
SMILES: {instruction}
FunctionalGroups: {functional_groups_str}
Ground-truth description: {target}

**Output:**
```
<think>
```
1. **Analyze Molecular Structure**: ...
2. **Summarize Key Elements**: ...
3. **Address Stereochemistry**: ...
4. **Predict Molecular Description**: ...
```
</think>
<answer>{target}</answer>
```

**TOMG** (MolCustom)

You are an expert chemist demonstrating how to solve a molecule generation problem based on given constraints (Mol Custom).

You are given:

1. An instruction that specifies the constraints for the molecule to be generated.

Your task is to generate a step-by-step reasoning process that logically constructs a valid molecule that satisfies all the given constraints. You MUST NOT mention "ground-truth", "given reactants", "provided answer", or any similar phrases in your `<think>` block. You must write the reasoning as if you are deducing the answer from scratch, even though you already know the destination.

Please structure your reasoning as follows:

1. **Analyze Constraints**: Deconstruct the instruction to create a comprehensive checklist of all specified molecular requirements.

* *List all explicit constraints, such as required functional groups, atom counts, or ring systems.*

* *Identify any implicit chemical rules, like ensuring correct valency for all atoms.*

* *This checklist will serve as the guide and final validation tool.*

2. **Construct Core Backbone**: Begin molecular construction by creating the primary scaffold that satisfies the most significant structural constraint.

* *Start with the largest specified fragment, such as a named ring system or a carbon chain of a specific length.*

* *If no core is defined, choose a simple starting point that facilitates adding other required features.*

3. **Assemble and Refine Structure**: Iteratively add the remaining functional groups and substituents to the core backbone to satisfy all remaining constraints.

* *Attach the required groups at chemically plausible positions on the backbone.*

* *Adjust the structure as needed, ensuring all atoms adhere to standard valency rules during assembly.*

4. **Comprehensively Validate**: Perform a final, rigorous check of the generated molecule against the initial constraint checklist for accuracy and chemical validity.

* *Systematically verify that every single constraint from the initial list has been met.*

* *Confirm the final SMILES string is syntactically correct and represents a chemically stable molecule.*

The final predicted molecule must be expressed as a SMILES string. Wrap your entire reasoning process in `<think>` tags and output only the final answer in `<answer>` tags.

**Input:**

Instruction: {`instruction`}

Ground-truth Molecule: {`target`}

**Output:**

```
<think>
```
1. **Analyze Constraints**: ...
2. **Construct Core Backbone**: ...
3. **Assemble and Refine Structure**: ...
4. **Comprehensively Validate**: ...
```
</think>
<answer>{target}</answer>
```

**TOMG** (MolEdit)

You are an expert chemist demonstrating how to solve a molecule editing problem.

You are given:
1. A molecule editing problem.
2. A list of key functional groups present in the SMILES, which can serve as hints.
Your task is to generate a step-by-step reasoning process that logically deduces the final edited molecule's SMILES from the given starting molecule and editing instruction. You MUST NOT mention "ground-truth", "given reactants", "provided answer", or any similar phrases in your <think> block. You must write the reasoning as if you are deducing the answer from scratch, even though you already know the destination.

Please structure your reasoning as follows:
1. **Task Parsing**: Deconstruct the problem statement to identify the starting molecule, the specific editing operation, and the target substructure.
* *Clearly state the input molecule's SMILES and the intended action (e.g., 'add', 'remove', 'replace').*
* *Identify the precise chemical group or site targeted for modification as described in the instruction.*
2. **Functional Group Analysis & Site Localization**: Analyze the starting molecule's SMILES to pinpoint the exact location for the edit.
* *Identify all functional groups within the starting molecule to create a structural map.*
* *Use this map to locate the exact atom(s) corresponding to the target site from the instruction.*
3. **Edit Molecule**: Execute the specified modification on the SMILES string by performing the necessary bond-breaking and bond-forming operations.
* *Describe the transformation in terms of SMILES manipulation (e.g., "inserting a 'C' atom into the chain").*
* *Generate the new SMILES string that represents the post-edit molecular structure.*
4. **Structure Validation**: Verify that the resulting molecule accurately reflects the requested edit and is chemically plausible.
* *Confirm that the modification has been correctly applied to the intended site.*
* *Check the final SMILES for correct syntax and ensure all atoms have proper valency.*
The final predicted molecule must be expressed as a SMILES string. You must output your reasoning within `<think>`...`</think>` and the final answer within `<answer>`...`</answer>`.

**Input:**
Problem: {`problem`}
FunctionalGroups: {`functional_groups_str`}
Ground-truth Edited Molecule: {`target`}

**Output:**
`<think>`
1. **Task Parsing**: ...
2. **Functional Group Analysis & Site Localization**: ...
3. **Edit Molecule**: ...
4. **Structure Validation**: ...
`</think>`
`<answer>`{`target`}`</answer>`

**TOMG** (MolOptimization)

You are an expert chemist demonstrating how to solve a molecule optimization problem.

You are given:
1. A molecule optimization problem, which includes the starting molecule (SMILES) and the optimization goal.
2. A list of key functional groups present in the starting molecule, which can serve as hints. Your task is to generate a step-by-step reasoning process that logically deduces the final optimized molecule's SMILES. You MUST NOT mention "ground-truth", "given reactants", "provided answer", or any similar phrases in your `<think>` block. You must write the reasoning as if you are deducing the answer from scratch, even though you already know the destination.

Please structure your reasoning as follows:
1. **Clarify Optimization Goal**: Interpret the optimization instruction to define the target property and the underlying chemical principle for its improvement.
* *Clearly state the property to be optimized (e.g., solubility, binding affinity, metabolic stability).*
* *Connect the goal to a specific chemical strategy (e.g., "increase solubility by adding polar groups").*
2. **Molecular Analysis & Strategy Customization**: Analyze the starting molecule to identify a suitable site for modification that aligns with the optimization strategy.
* *Identify regions of the molecule where modification is chemically feasible and unlikely to disrupt core activity.*
* *Select a specific substituent or functional group to add/remove/replace based on the defined strategy.*
3. **Modify Molecule**: Execute the planned structural modification on the starting molecule's SMILES to generate the new, optimized structure.
* *Describe the specific edit performed on the SMILES string to implement the chemical change.*
* *Generate the new SMILES string representing the final, optimized molecule.*
4. **Validate Modification**: Confirm that the edit was performed correctly and that the resulting molecule plausibly achieves the optimization goal.
* *Verify that the final structure incorporates the intended change at the correct location.*
* *Briefly rationalize how the modification is expected to improve the target property (e.g., "the added hydroxyl group increases polarity").*

The final predicted molecule must be expressed as a SMILES string. You must output your reasoning within `<think>`...`</think>` and the final answer within `<answer>`...`</answer>`.

**Input:**
Problem: {`problem`}
FunctionalGroups: {`functional_groups_str`}
Ground-truth Optimized Molecule: {`target`}

**Output:**
`<think>`
1. **Clarify Optimization Goal**: ...
2. **Molecular Analysis & Strategy Customization**: ...
3. **Modify Molecule**: ...
4. **Validate Modification**: ...
`</think>`
`<answer>`{`target`}`</answer>`

**Reaction Prediction**

You are an expert chemist demonstrating how to solve a reaction prediction problem.

You are given:
1. A set of chemical reactants represented by SMILES string.
2. A list of key functional groups present in the reactants, which serve as hints for discon-nection.

Your task is to generate a step-by-step reasoning process that logically predicts the product SMILES from the given reactant SMILES and its functional groups. You MUST NOT mention "ground-truth", "given reactants", "provided answer", or any similar phrases in your `<think>` block. You must write the reasoning as if you are deducing the answer from scratch, even though you already know the destination.

Please structure your reasoning as follows:
1. **Reactant Analysis**: Examine the reactants' structures to identify key features relevant to reactivity.
* *Assess the overall structure, including the carbon skeleton and stereochemistry.*
* *Identify any notable structural motifs that could influence the reaction.*
2. **Functional Group Identification**: Locate and classify all functional groups to determine potential reaction sites.
* *Identify all key functional groups and classify them by reactivity (e.g., nucleophilic, elec-trophilic, acidic, basic).*
* *Consider potential interactions between these groups or with external reagents.*
3. **Reaction Type Analysis**: Determine the most probable reaction pathway based on the reactants' structural and functional group properties.
* *Deduce the most plausible chemical transformation by correlating the functional groups present.*
* *Propose a likely reaction type, such as a known named reaction or a fundamental class like substitution or addition.*
4. **Product Prediction**: Construct the final product by applying the proposed reaction mech-anism and generating the resultant SMILES string.
* *Detail the bond formations and breakages, considering regioselectivity and stereoselectiv-ity to determine the precise product structure.*
* *Generate the new SMILES string that represents the final predicted molecule(s).*
The final predicted product must be expressed as a SMILES string. If multiple prod-ucts are predicted, they MUST be separated by a period '.' instead of commas. You must output your reasoning within `<think>`...`</think>` and the final answer within `<answer>`...`</answer>`.

**Input:**
Reactants SMILES: {`reactants`}
FunctionalGroups: {`functional_groups_str`}
Ground-truth Product: {`target`}

**Output:**
`<think>`
1. **Reactant Analysis**: ...
2. **Functional Group Identification**: ...
3. **Reaction Type Analysis**: ...
4. **Product Prediction**: ...
`</think>`
`<answer>`{`target`}`</answer>`

**Retrosynthesis**

You are an expert chemist demonstrating how to solve a retrosynthesis problem.

You are given:

1. A chemical product represented by a SMILES string.

2. A list of key functional groups present in the product, which serve as hints for disconnection.

Your task is to generate a step-by-step reasoning process that logically deduces the reactant SMILES from the given product SMILES and its functional groups. You MUST NOT mention "ground-truth", "given reactants", "provided answer", or any similar phrases in your `<think>` block. You must write the reasoning as if you are deducing the answer from scratch, even though you already know the destination.

Please structure your reasoning as follows:

1. **Product Analysis**: Examine the product's structure to identify strategic bonds for disconnection.

*  Assess the product's carbon skeleton, stereochemistry, and any notable structural motifs.

*  Identify key bonds whose formation is synthetically plausible, often adjacent to functional groups.

2. **Functional Group Identification**: Analyze the functional groups to infer the last synthetic step.

*  Locate all key functional groups, recognizing them as the results of specific chemical reactions.

*  Use these groups to guide the selection of a primary disconnection strategy.

3. **Reaction Type Analysis**: Propose a plausible forward reaction that could have formed a key bond in the product.

*  Based on the target bond and adjacent functionalities, suggest a reliable forward synthesis reaction (e.g., Wittig, Grignard, Diels-Alder).

*  This proposed reaction dictates the nature of the precursor fragments (synthons).

4. **Reactant Prediction**: Perform the disconnection to derive the structures of the starting materials.

*  Break the target bond in the product to generate the corresponding synthons.

*  Convert these synthons into stable, charge-neutral reactant molecules and generate their SMILES strings.

The final predicted reactants must be expressed as SMILES strings. If multiple reactants are predicted, they MUST be separated by a period '.' instead of commas. You must output your reasoning within `<think>`...`</think>` and the final answer within `<answer>`...`</answer>`.

**Input:**

Product SMILES: {`products`}

FunctionalGroups: {`functional_groups_str`}

Ground-truth Reactants: {`target`}

**Output:**

`<think>`

1. **Product Analysis**: ...

2. **Functional Group Identification**: ...

3. **Reaction Type Analysis**: ...

4. **Reactant Prediction**: ...

`</think>`

`<answer>`{`target`}`</answer>`

## C.5  ANALYSIS OF CRPS

We provide a detailed analysis regarding the quantification, generalization, and robustness of the Chemical Reasoning Protocols (CRP) distilled in Phase 2.

**Protocol Count and Generalization.**  Our cross-task merging process resulted in **9 core protocols** built around fundamental "reasonable steps" such as "Analyze Molecular Structure" or "Identify Functional Groups." These steps are abstract and universal enough to cover virtually all chemical problem-solving workflows. The protocols generalize effectively for two reasons. First, the steps themselves remain consistent across tasks; what changes is the task-specific focus within each step. For example, "Analyze Molecular Structure" might focus on stereocenters for a property prediction task but on potential disconnection sites for a retrosynthesis task. Second, the "hints for common mistakes" derived from one task (e.g., misinterpreting a SMILES string for a reactant) are often relevant to others, creating a rich, shared knowledge base that enhances generalization.

**Robustness to Teacher Errors.**  Our protocol design is inherently robust to partially incorrect teacher traces. The goal of this phase is not to perfectly replicate every detail of a teacher's trace, but to extract a high-level, structured workflow. We address teacher errors through two mechanisms: (1) If the final result is wrong, we analyze the reasoning chain to identify the failure mode, and this analysis directly informs the "hints for common mistakes" attached to the relevant protocol step. (2) If the final result is correct but the reasoning is partially flawed, our process remains effective because we only summarize the broad, high-level steps (e.g., "the model first identified reactants, then proposed a reaction type..."). Minor logical flaws within a step do not corrupt the extraction of this high-level structure. Therefore, the protocol remains sound, and the risk of propagating subtle errors is minimal.

## C.6   HUMAN EVALUATION

To provide a nuanced assessment of reasoning quality, we conducted a human evaluation with the help of chemistry experts. The experts were tasked with evaluating the generated CoT from Chem-R and several leading baseline models. We designed a comprehensive rubric consisting of six distinct, orthogonal metrics to capture different facets of a high-quality explanation: *Chemical Soundness* (the factual correctness of the chemistry), *Logical Coherence* (the step-by-step logical flow), *Step-by-Step Completeness* (whether crucial steps are missing), *Justification of the Conclusion* (the faithfulness of the reasoning to the final answer), *Clarity and Conciseness* (the quality of the language), and Expert-level Insight (the depth and nuance of the reasoning). Each metric was scored on a 0-5 scale, allowing for a detailed comparison of the models' ability to produce human-like, expert-level thought processes.

**Evaluation Limitations.**   A potential limitation is that CoT from Chem-Rcan be stylistically distinct, often more structured, due to its protocol-based training. To mitigate any resulting bias, the evaluation was conducted under *strictly blind conditions*. Experts assessed fully anonymized responses without any knowledge of the generating model, ensuring their ratings were based exclusively on the intrinsic quality of the reasoning rather than on stylistic patterns.

| Score | Chemical Soundness | Logical Coherence | Step-by-Step Completeness | Justification of the Conclusion | Clarity and Conciseness | Expert-level Insight |
|---|---|---|---|---|---|---|
| 5 | **Excellent:** Flawless chemistry, all principles correctly applied. | **Excellent:** Perfectly logical and sequential flow. | **Excellent:** All critical steps are explicit and thorough. | **Excellent:** Correct conclusion is fully justified by the reasoning. | **Excellent:** Exceptionally clear, concise, and precise. | **Excellent:** Deep, nuanced understanding that mimics expert thinking. |
| 4 | **Good:** Minor, inconsequential chemical inaccuracy. | **Good:** Sound logic with a minor unclear transition. | **Good:** Omits a minor, obvious step. | **Good:** Correct conclusion, but the link to reasoning is slightly weak. | **Good:** Clear and easy to understand, but could be more polished. | **Good:** Strong competence, correctly identifies key factors. |
| 3 | **Acceptable:** Noticeable chemical error(s), but mostly correct. | **Acceptable:** Generally follows, but has some logical gaps. | **Acceptable:** Skips one or two important steps. | **Acceptable:** Reasoning is sound but leads to a minor error, OR the conclusion is not fully supported. | **Acceptable:** Generally understandable, but somewhat verbose or awkward. | **Acceptable:** Correct but superficial or "textbook" reasoning. |
| 2 | **Poor:** Significant misunderstanding of core concepts. | **Poor:** Disjointed and hard to follow. | **Poor:** Skips multiple crucial steps, making logical leaps. | **Poor:** Incorrect conclusion that is not supported by the reasoning. | **Poor:** Vague, confusing, and difficult to parse. | **Poor:** Misses key nuances and only addresses the obvious. |
| 1 | **Very Poor:** Built on fundamentally flawed chemical principles. | **Very Poor:** Largely illogical and chaotic. | **Very Poor:** A brief sketch missing almost all necessary logic. | **Very Poor:** Conclusion seems unrelated to the reasoning provided. | **Very Poor:** Nearly unintelligible. | **Very Poor:** Demonstrates a naive or simplistic understanding. |
| 0 | **No Value:** Completely incorrect chemistry. | **No Value:** No logical structure. | **No Value:** Provides only the final answer. | **No Value:** No connection between reasoning and answer. | **No Value:** Gibberish. | **No Value:** Shows a complete lack of understanding. |

Figure 5: Human evaluation rubric for Chain-of-Thought quality. Experts are to score the generated reasoning on a 0-5 scale (0=worst, 5=best) across the six metrics provided: Chemical Soundness, Logical Coherence, Completeness, Justification, Clarity, and Expert-level Insight.

Table 5: **Human evaluation of model-generated reasoning.** For each column the **best** and second-best models are highlighted.

| Metric | Chem-R | Gemini-2.5 Pro | DeepSeek-R1 | ether0 |
|---|---|---|---|---|
| Chemical Soundness | **4.75** | **3.95** | 3.45 | 2.15 |
| Logical Coherence | **4.85** | **4.25** | 3.80 | 2.35 |
| Step-by-Step Completeness | **4.20** | 3.85 | **3.90** | 1.95 |
| Justification of the Conclusion | **4.28** | **4.10** | 3.55 | 2.05 |
| Clarity and Conciseness | **4.65** | **4.55** | 3.70 | 2.55 |
| Expert-level Insight | **4.55** | **3.75** | 3.20 | 1.85 |

# D  EXPERIEMENT RESULT

In this section, we will present the specific results of each subtask in the experiment to provide a better demonstration of the model's performance.

## D.1  NAME PREDICTION

In our name prediction task, we evaluated the model's performance on two sub-tasks: SMILES to IUPAC name translation (SMILES2IUPAC) and IUPAC name to SMILES translation (IU-PAC2SMILES). For both sub-tasks, we use the **exact match accuracy** as the evaluation metric. All SMILES comparisons are performed after canonicalization to ensure consistency.

| Model | SMILES2IUPAC↑ | IUPAC2SMILES↑ | Average↑ |
|---|---|---|---|
| *Task-specific specialist models* | | | |
| STOUT (Rajan et al., 2021) | 0.55 | 0.70 | 0.63 |
| *LLM-based generalist models* | | | |
| GPT-4o | 0.00 | 0.02 | 0.01 |
| Gemini-2.5-Pro | 0.01 | 0.33 | 0.17 |
| DeepSeek-R1 | 0.01 | 0.09 | 0.05 |
| ChemDFM-v1.5-8B | 0.06 | 0.22 | 0.14 |
| ether0-24B | 0.00 | 0.29 | 0.15 |
| Chem-R-8B | 0.51 | 0.47 | 0.49 |

Table 6: Accuracy scores in name prediction tasks. The task-specific specialist models are sourced from (Guo et al., 2023).

## D.2 PROPERTY PREDICTION

In the molecule property prediction task, we evaluated the model's performance across a suite of benchmark datasets: BACE, BBBP, ClinTox, HIV, and Tox21. Each task is formulated as a binary classification problem. For evaluation, we uniformly use classification accuracy as the sole metric across all datasets. To further verify the competitiveness of our approach against specialized geometric deep learning models, we additionally evaluate a canonical Graph Neural Network (GNN) baseline, Chemprop (D-MPNN) (Heid et al., 2023), using classification accuracy.

Table 7: AUC-ROC scores of different models in molecular property prediction tasks. **Avg.**: average. The task-specific specialist models are sourced from (Zhao et al., 2025c).

| Model | BACE↑ | BBBP↑ | ClinTox↑ | HIV↑ | Tox21↑ | Avg.↑ |
|---|---|---|---|---|---|---|
| *Task-specific specialist models* | | | | | | |
| Uni-Mol (Zhou et al., 2023) | 0.86 | 0.73 | 0.92 | 0.81 | 0.80 | 0.82 |
| MolXPT (Liu et al., 2023b) | 0.88 | 0.80 | 0.95 | 0.78 | 0.77 | 0.84 |
| InstructMol (Cao et al., 2023) | 0.86 | 0.64 | - | 0.74 | - | - |
| *LLM-based generalist models* | | | | | | |
| ChemDFM-v1.5-8B | 0.78 | 0.75 | 0.69 | 0.88 | 0.74 | 0.77 |
| ether0-24B | 0.58 | 0.64 | 0.62 | 0.67 | 0.51 | 0.60 |
| Chem-R-8B | 0.78 | 0.81 | 0.95 | 1.00 | 0.71 | 0.85 |

Table 8: Accuracy scores of different models in molecular property prediction tasks. We include Chemprop (Heid et al., 2023) as a specialized GNN baseline for comparison.

| Model | BACE↑ | BBBP↑ | ClinTox↑ | HIV↑ | Tox21↑ | Avg.↑ |
|---|---|---|---|---|---|---|
| *Specialized GNN Baseline* | | | | | | |
| Chemprop (D-MPNN) | 0.66 | 0.72 | 0.84 | 0.85 | 0.83 | 0.78 |
| *General Foundation Models* | | | | | | |
| GPT-4o | 0.34 | 0.61 | 0.98 | 1.00 | 0.49 | 0.68 |
| Gemini-2.5-Pro | 0.36 | 0.78 | 0.30 | 1.00 | 0.36 | 0.56 |
| DeepSeek-R1 | 0.38 | 0.63 | 0.8 | 1.00 | 0.34 | 0.63 |
| *Chemical LLMs* | | | | | | |
| ChemDFM-v1.5-8B | 0.77 | 0.57 | 0.69 | 0.85 | 0.81 | 0.74 |
| ether0-24B | 0.43 | 0.48 | 0.74 | 0.82 | 0.72 | 0.64 |
| Chem-R-8B | **0.74** | **0.82** | **0.98** | **1.00** | 0.80 | **0.87** |

As shown in Table 8, Chem-R not only remains competitive but outperforms the strong Chemprop baseline on average accuracy (0.87 vs. 0.78). This demonstrates the power of our reasoning-based approach, which matches or exceeds specialized geometric deep learning models while offering superior interpretability.

## D.3 MOLECULE DESIGN

In the text-based molecule design task, we evaluate the model's ability to generate a correct SMILES string from a given textual description of a molecule, using the ChEBI-20 dataset. The evaluation is multi-faceted. First, we measure the fundamental **Validity** of the outputs, which is the percentage of generated SMILES that represent chemically valid molecules as verified by RDKit. To assess textual fidelity against the ground-truth SMILES, we employ several string-based metrics: **Exact Match (EM)** for identical strings, **Levenshtein Distance (Lev.)** to measure edit distance (lower is better), and the **BLEU score**, which quantifies n-gram overlap via the formula:

$$BLEU = BP \cdot \exp\left(\sum_{n=1}^{N} w_n \log p_n\right) \tag{3}$$

where $BP$ is the Brevity Penalty and $p_n$ is the modified n-gram precision. Finally, to evaluate structural correctness, we calculate the **Tanimoto coefficient** between the molecular fingerprints (MACCS, RDKit, and Morgan) of the generated and ground-truth molecules, where a higher similarity score indicates a greater structural resemblance. We choose MolT5-large (Edwards et al., 2022), MolReGPT (Li et al., 2024b), Mol-Instruction (Fang et al., 2023), MolReasoner (Zhao et al., 2025a) and Mol-R1 (Li et al., 2025b) as the task-specific specialist models.

| Model | BLEU↑ | EM↑ | Lev.↓ | MACCS↑ | RDK↑ | Morgan↑ | Validity↑ |
|---|---|---|---|---|---|---|---|
| *Task-specific specialist models* | | | | | | | |
| MolT5-large | 0.85 | 0.31 | 16.07 | 0.83 | 0.74 | 0.68 | 0.91 |
| MolReGPT | 0.86 | 0.28 | 17.14 | 0.90 | 0.80 | 0.74 | 0.90 |
| Mol-Instruction | 0.30 | 0.04 | 39.42 | 0.44 | 0.29 | 0.25 | 1.00 |
| MolReasoner | 0.78 | 0.08 | 26.93 | 0.68 | 0.44 | 0.36 | 0.97 |
| Mol-R1 | 0.64 | 0.23 | 32.94 | 0.82 | 0.68 | 0.61 | 0.85 |
| *LLM-based generalist models* | | | | | | | |
| GPT-4o | 0.45 | 0.07 | 48.38 | 0.79 | 0.58 | 0.50 | 0.77 |
| Gemini-2.5-Pro | 0.69 | 0.29 | 130.26 | 0.95 | 0.88 | 0.82 | 0.91 |
| DeepSeek-R1 | 0.51 | 0.22 | 169.36 | 0.92 | 0.82 | 0.75 | 0.78 |
| ChemDFM-v1.5-8B | 0.90 | 0.55 | 7.12 | 0.94 | 0.88 | 0.84 | 0.98 |
| ether0-24B | 0.39 | 0.27 | 860.99 | 0.82 | 0.70 | 0.64 | 0.73 |
| Chem-R-8B | 0.84 | 0.41 | 17.42 | 0.92 | 0.83 | 0.78 | 0.94 |

Table 9: Performance of different models on the text-based molecule generation task on the ChEBI-20 dataset. **BLEU**: Bilingual Evaluation Understudy, **EM**: Exact Match, **Lev.**: Levenshtein distance, **MACCS**: MACCS fingerprint similarity, **RDK**: RDK fingerprint similarity, **Morgan**: Morgan fingerprint similarity, **Validity**: Percentage of valid molecules.

### D.4 MOLECULE CAPTIONING

In the molecule captioning task, we evaluate the model's ability to generate an accurate and fluent natural language description from a given molecular structure (SMILES string), using the ChEBI-20 dataset. To comprehensively assess the quality of the generated text, we employ a suite of standard metrics. We use the **BLEU** score to measure n-gram precision, specifically reporting **BLEU-2** ($N = 2, w_1 = w_2 = 0.5$) and **BLEU-4** ($N = 4, w_n = 0.25$), based on the general formula:

$$BLEU = BP \cdot \exp\left(\sum_{n=1}^{N} w_n \log p_n\right) \tag{4}$$

where $BP$ is the Brevity Penalty and $p_n$ is the modified n-gram precision. For a recall-oriented evaluation, we utilize the **ROUGE** family of metrics. We report the F1-scores for **ROUGE-1 (R-1)** and **ROUGE-2 (R-2)**, which measure unigram and bigram overlap, and **ROUGE-L (R-L)**, which is based on the longest common subsequence (LCS). The ROUGE-L F-score is computed as:

$$\text{ROUGE-L}_{\text{f-score}} = \frac{(1 + \beta^2)R_{lcs}P_{lcs}}{R_{lcs} + \beta^2 P_{lcs}} \tag{5}$$

where $R_{lcs}$ and $P_{lcs}$ are the LCS-based recall and precision, and $\beta$ is set to 1 to weigh recall and precision equally for the F1-score. Lastly, we incorporate the **METEOR** score, which enhances evaluation by considering synonymy and stemming. It is based on the harmonic mean of unigram precision ($P$) and recall ($R$), $F_{mean}$, which weights recall more than precision:

$$F_{mean} = \frac{10PR}{R + 9P} \tag{6}$$

The final score is calculated by applying a fragmentation penalty ($Pen$) to this value:

$$\text{METEOR} = F_{mean} \cdot (1 - Pen) \tag{7}$$

where $Pen$ is a penalty for fragmentation based on the alignment of chunks between the generated and reference texts.

| Model | BLEU-2↑ | BLEU-4↑ | R-1↑ | R-2↑ | R-L↑ | METEOR↑ |
|---|---|---|---|---|---|---|
| *Task-specific specialist models* | | | | | | |
| MolT5-large | 0.59 | 0.51 | 0.65 | 0.51 | 0.59 | 0.61 |
| MolReGPT | 0.61 | 0.53 | 0.63 | 0.48 | 0.56 | 0.61 |
| MolReFlect | 0.68 | 0.61 | 0.70 | 0.57 | 0.64 | 0.68 |
| Mol-Instruction | 0.10 | 0.07 | 0.28 | 0.18 | 0.26 | 0.19 |
| MolReasoner | 0.44 | 0.32 | 0.55 | 0.37 | 0.48 | 0.48 |
| *LLM-based generalist models* | | | | | | |
| GPT-4o | 0.04 | 0.01 | 0.13 | 0.02 | 0.10 | 0.11 |
| Gemini-2.5-Pro | 0.09 | 0.04 | 0.20 | 0.06 | 0.13 | 0.31 |
| DeepSeek-R1 | 0.10 | 0.04 | 0.14 | 0.03 | 0.11 | 0.15 |
| ChemDFM-v1.5-8B | 0.33 | 0.28 | 0.46 | 0.36 | 0.43 | 0.40 |
| ether0-24B | 0.01 | 0.00 | 0.05 | 0.01 | 0.04 | 0.05 |
| Chem-R-8B | 0.48 | 0.41 | 0.61 | 0.44 | 0.53 | 0.58 |

Table 10: Performance of different models on the molecule description task on the ChEBI-20 dataset. **R-1**: ROUGE-1 (Recall-Oriented Understudy for Gisting Evaluation-1), **R-2**: ROUGE-2, **R-L**: ROUGE-L (ROUGE-L stands for longest common subsequence), MTEOR: METEOR (Metric for Evaluation of Translation with Explicit ORdering).

### D.5 TEXT-BASED OPEN MOLECULE GENERATION

In the text-based open molecule generation task, we evaluate the model's ability to perform complex chemical reasoning and creative design, using the TOMG-Bench benchmark (Li et al., 2024a). The evaluation is structured around three distinct tasks designed to probe different capabilities: **Molecule Editing (MolEdit)**, **Molecule Optimization (MolOpt)**, and **Customized Molecule Generation (MolCustom)**. And we choose MolT5 (Edwards et al., 2022), BioT5 (Pei et al., 2023) and Open-MolIns (Li et al., 2024a) (which is trained on the full set of training data of TOMG-Bench) for the task-specific specialist models.

For the **MolEdit** and **MolOpt** tasks, which involve modifying an existing molecule, the assessment is threefold. First, we measure the **Success Rate (SR)** to verify if the model's output correctly fulfills the textual instruction. Second, to ensure the modification is a rational and localized edit rather than a completely new structure, we calculate the Tanimoto **Similarity (Sim.)** between the Morgan fingerprints of the original and generated molecules.

For the **MolCustom** task, which requires generating a molecule from scratch (*de novo*), the metrics are adapted. The **Success Rate (SR)** evaluates if the generated molecule adheres to a set of specified structural constraints (e.g., atom counts, bond types). Instead of similarity, we measure **Novelty (Nov.)** to quantify the uniqueness of the generated molecule. The novelty $n$ for a generated molecule $m^g$ is calculated as:

$$n(m^g) = 1 - \frac{\sum_{m^k \in \text{Zinc}} \delta(m^g, m^k)}{|\text{Zinc}|} \quad (8)$$

where $\delta(m^g, m^k)$ is the Tanimoto similarity to a known molecule $m^k$ in the Zinc database.

To provide a single, comprehensive ranking of model performance, the benchmark introduces a **Weighted Success Rate (WSR)**. This metric combines the core success rate with a quality metric relevant to each task—Similarity for MolEdit/MolOpt and Novelty for MolCustom. The WSR for a given subtask $t$ is defined as:

$$WSR_t = \begin{cases} n_t \times SR_t, & t \in \{\text{MolCustom}\} \\ \delta_t \times SR_t, & t \in \{\text{MolEdit, MolOpt}\} \end{cases} \quad (9)$$

where $n_t$ is the novelty score and $\delta_t$ is the similarity score for that subtask. The final performance is then the average WSR across all nine subtasks:

$$WSR = \frac{1}{9} \sum_t WSR_t \quad (10)$$

Finally, for all tasks, we measure the fundamental **Validity (Val.)** to ensure every generated SMILES string represents a chemically sound molecule.

| MolEdit | AddComponent | | | DelComponent | | | SubComponent | | |
|---|---|---|---|---|---|---|---|---|---|
| | SR↑ | Sim.↑ | Val.↑ | SR↑ | Sim.↑ | Val.↑ | SR↑ | Sim.↑ | Val.↑ |
| *Task-specific specialist models* | | | | | | | | | |
| MolT5 | 0.28 | 0.11 | 0.93 | 0.22 | 0.12 | 0.92 | 0.17 | 0.09 | 0.94 |
| BioT5 | 0.35 | 0.16 | 1.00 | 0.17 | 0.16 | 1.00 | 0.07 | 0.16 | 1.00 |
| OpenMolIns | 0.78 | 0.68 | 0.95 | 0.86 | 0.62 | 0.91 | 0.61 | 0.74 | 0.96 |
| *LLM-based generalist models* | | | | | | | | | |
| GPT-4o | 0.62 | 0.68 | 0.74 | 0.70 | 0.60 | 0.85 | 0.80 | 0.72 | 0.94 |
| ChemDFM | 0.29 | 0.65 | 0.91 | 0.21 | 0.69 | 0.93 | 0.22 | 0.64 | 0.87 |
| Chem-R | 0.89 | 0.69 | 0.98 | 0.89 | 0.62 | 0.93 | 0.62 | 0.75 | 0.99 |
| **MolOpt** | **LogP** | | | **MR** | | | **QED** | | |
| | SR↑ | Sim.↑ | Val.↑ | SR↑ | Sim.↑ | Val.↑ | SR↑ | Sim.↑ | Val.↑ |
| *Task-specific specialist models* | | | | | | | | | |
| MolT5 | 0.42 | 0.10 | 0.82 | 0.45 | 0.11 | 0.87 | 0.47 | 0.12 | 0.92 |
| BioT5 | 0.52 | 0.15 | 1.00 | 0.51 | 0.16 | 1.00 | 0.51 | 0.16 | 1.00 |
| OpenMolIns | 0.88 | 0.67 | 0.93 | 0.70 | 0.67 | 0.94 | 0.86 | 0.67 | 0.93 |
| *LLM-based generalist models* | | | | | | | | | |
| GPT-4o | 0.72 | 0.66 | 0.88 | 0.69 | 0.64 | 0.84 | 0.40 | 0.62 | 0.86 |
| ChemDFM | 0.30 | 0.66 | 0.76 | 0.25 | 0.68 | 0.82 | 0.31 | 0.66 | 0.81 |
| Chem-R | 0.90 | 0.66 | 0.97 | 0.92 | 0.68 | 0.97 | 0.74 | 0.68 | 0.98 |
| **MolCustom** | **AtomNum** | | | **BondNum** | | | **FunctionalGroup** | | |
| | SR↑ | Nov.↑ | Val.↑ | SR↑ | Nov.↑ | Val.↑ | SR↑ | Nov.↑ | Val.↑ |
| *Task-specific specialist models* | | | | | | | | | |
| MolT5 | 0.02 | 0.71 | 0.84 | 0.01 | 0.56 | 0.89 | 0.04 | 0.61 | 0.94 |
| BioT5 | 0.01 | 0.84 | 1.00 | 0.01 | 0.67 | 1.00 | 0.05 | 0.68 | 1.00 |
| OpenMolIns | 0.12 | 0.68 | 0.85 | 0.12 | 0.67 | 0.90 | 0.35 | 0.64 | 0.95 |
| *LLM-based generalist models* | | | | | | | | | |
| GPT-4o | 0.20 | 0.67 | 0.59 | 0.07 | 0.63 | 0.86 | 0.23 | 0.65 | 0.86 |
| ChemDFM | 0.01 | 0.68 | 0.67 | 0.02 | 0.61 | 0.90 | 0.03 | 0.60 | 0.81 |
| Chem-R | 0.22 | 0.69 | 0.85 | 0.10 | 0.70 | 0.70 | 0.31 | 0.64 | 0.93 |

Table 11: Detailed results on TOMG-Bench for different models. The indicators are: **SR** = Success Rate, **Sim.** = Similarity, **Nov.** = Novelty, **Val.** = Validity. MolT5 refers to MolT5-large, BioT5 refers to BioT5-base, OpenMolIns refers to the performance of the Llama-3.1-8B model trained on the largest instruction fine-tuning dataset OpenMolIns-xlarge proposed by TOMG-Bench, and ChemDFM refers to ChemDFM-v1.5-8B. The task-specific specialist models are sourced from (Li et al., 2024a).

Due to the extensive size of the TOMG-Bench (45,000 test instances across 9 tasks), evaluating the full set with closed-source, API-based models like Gemini and DeepSeek incurs prohibitive costs. To provide a direct comparison, we performed an evaluation on a 5% stratified random sample of the TOMG test set. On this subset, Chem-R achieves the highest overall weighted accuracy (wAcc), demonstrating superior performance against leading API-based models, particularly in complex editing and optimization tasks. The detailed results are presented in Table 12.

Table 12: Performance comparison on a 5% stratified random sample of the TOMG-Bench test set. The metric here is accuracy.

| Models | MolCustom (Atom / Bond / Func) | MolEdit (Add / Del / Sub) | MolOpt (LogP / QED / MR) | wAcc |
|---|---|---|---|---|
| Gemini-2.5 | 0.420 / 0.250 / 0.550 | 0.780 / 0.820 / 0.680 | 0.850 / 0.680 / 0.880 | 0.4089 |
| DeepSeek-R1 | 0.360 / 0.210 / 0.480 | 0.820 / 0.790 / 0.650 | 0.820 / 0.650 / 0.840 | 0.3921 |
| **Chem-R** | **0.260 / 0.260 / 0.472** | **0.912 / 0.916 / 0.652** | **0.912 / 0.728 / 0.932** | **0.4523** |

### D.6 YIELD PREDICTION

In the reaction yield prediction task, we evaluate the model's performance on two reaction datasets: Buchwald-Hartwig and Suzuki-Miyaura. The task is framed as a binary classification problem to predict whether a reaction yield is high or low. We use **classification accuracy** as the sole evaluation metric, with the average accuracy across both datasets also reported (as shown in Table 13).

| Model | Buchwald-Hartwig↑ | Suzuki-coupling↑ | Average↑ |
|---|---|---|---|
| *Task-specific specialist models* | | | |
| UAGNN (Kwon et al., 2022) | 0.97 | 0.96 | 0.96 |
| *LLM-based generalist models* | | | |
| GPT-4o | 0.20 | 0.20 | 0.20 |
| Gemini-2.5-Pro | 0.23 | 0.47 | 0.35 |
| DeepSeek-R1 | 0.20 | 0.45 | 0.33 |
| ChemDFM-v1.5-8B | 0.35 | 0.38 | 0.37 |
| ether0-24B | 0.02 | 0.03 | 0.03 |
| Chem-R-8B | 0.87 | 0.85 | 0.86 |

Table 13: Accuracy scores of different models in yield prediction tasks. The task-specific specialist models is sourced from (Zhao et al., 2025c).

### D.7 REAGENT SELECTION

In the reagent selection task, we utilize the Suzuki High-Throughput Experimentation (HTE) dataset. This task is divided into three sub-tasks: predicting the correct reactant, solvent, and ligand for a given reaction. The evaluation metrics vary by sub-task. For reactant and solvent prediction, we report the **top-1 accuracy**. For ligand prediction, we report the **top-5 accuracy**, which considers a prediction correct if the ground-truth ligand is among the top five candidates proposed by the model. The 'Avg.' column in the table represents the average of these three accuracy scores.

| Model | Reagent Selection | | | | Reaction Prediction | Retrosynthesis |
|---|---|---|---|---|---|---|
| | Reactant | Solvent | Ligand | Avg. | | |
| *Task-specific specialist models* | | | | | | |
| Chemformer | – | – | – | – | 0.94 | 0.54 |
| Mol-Instruction | – | – | – | – | 0.05 | – |
| InstructMol | – | – | – | – | 0.54 | – |
| ochem_predict_nn | – | – | – | – | 0.72 | – |
| AiZynthFinder | – | – | – | – | – | 0.47 |
| *LLM-based generalist models* | | | | | | |
| GPT-4o | 0.48 | 0.11 | 0.19 | 0.26 | 0.04 | 0.00 |
| Gemini-2.5-Pro | 0.17 | 0.07 | 0.56 | 0.27 | 0.35 | 0.15 |
| DeepSeek-R1 | 0.15 | 0.17 | 0.08 | 0.13 | 0.34 | 0.13 |
| ChemDFM-v1.5-8B | 0.47 | 0.10 | 0.49 | 0.35 | 0.50 | 0.07 |
| ether0-24B | 0.11 | 0.35 | 0.18 | 0.21 | 0.65 | 0.04 |
| Chem-R-8B | 0.62 | 0.57 | 0.87 | 0.69 | 0.82 | 0.39 |

Table 14: Performance of task-specific specialist models and LLM-based generalist models on reagent selection, reaction prediction, and retrosynthesis tasks. The task-specific specialist models are sourced from (Zhao et al., 2025c). "–" means that the model was not designed for the task. We also include template-based (non-LLM based) baselines ochem_predict_nn (Coley et al., 2017) and AiZynthFinder (Genheden et al., 2020) for comparison.

### D.8 REACTION PREDICTION

In the reaction prediction task, we evaluate the model's ability to predict the major product of a chemical reaction, using the USPTO_Mixed dataset. Performance is measured using **exact match**

**accuracy**. All SMILES comparisons are performed after canonicalization to ensure consistency. We additionally benchmark against the specialized template-based model ochem_predict_nn (Coley et al., 2017). And we choose Chemformer (Irwin et al., 2022), Mol-Instruction (Fang et al., 2023) and InstructMol (Cao et al., 2023) for the task-specific specialist models. As shown in Table 14, Chem-R outperforms this baseline (0.82 vs. 0.72), demonstrating that our reasoning-based approach yields superior predictive accuracy compared to traditional template matching methods.

### D.9 RETROSYNTHESIS

In the retrosynthesis task, the goal is to predict the reactants that form a given product molecule. This is evaluated on the widely-used USPTO-50k dataset. Similar to reaction prediction, we employ **exact match accuracy** as the evaluation metric. All SMILES comparisons are performed after canonicalization to ensure consistency. We further compare Chem-R with the search-based tool AiZynthFinder (Genheden et al., 2020). And we choose Chemformer (Irwin et al., 2022) for the task-specific specialist models. As shown in Table 14, while the specialized search algorithm achieves higher accuracy (0.47 vs. 0.39), it operates as a black-box predictor. In contrast, Chem-R provides step-by-step reasoning traces, delivering a competitive level of accuracy while ensuring the interpretability and verifiability essential for scientific discovery.

### D.10 OUT OF DOMAIN

To evaluate the out-of-domain (OOD) generalization of our model, Chem-R, we benchmarked its performance on the challenging **Molecular Property Optimization** task from ChemCoTBench (Li et al., 2025a). We selected four representative targets: Solubility, DRD2, JNK3, and GSK-3$\beta$. The evaluation, presented in Table 15, compares Chem-R against its base model, Llama-3.1-8B-Instruct, and other powerful LLMs.

The results clearly demonstrate the effectiveness of our training. The base Llama-3.1-8B-Instruct model performs poorly, whereas Chem-R shows a dramatic improvement in both success rate (SR%) and mean property improvement ($\Delta$) across all tasks. This signifies that Chem-R has acquired robust chemical reasoning skills that generalize effectively. Furthermore, Chem-R proves to be highly competitive, significantly outperforming the much larger Llama-3.3-70B-Instruct model, which confirms the strong OOD capabilities of our approach.

| Models | Solubility | | DRD2 | | JNK3 | | GSK3-$\beta$ | |
|---|---|---|---|---|---|---|---|---|
| | $\Delta$ | SR% | $\Delta$ | SR% | $\Delta$ | SR% | $\Delta$ | SR% |
| Qwen3-235B-A22B-think | 0.27 | 42 | 0.03 | 31 | -0.01 | 23 | 0.01 | 31 |
| Qwen3-32B-think | 0.11 | 23 | 0.00 | 6 | -0.02 | 6 | -0.02 | 5 |
| Llama-3.1-8B-Instruct | 0.01 | 10 | -0.07 | 2 | -0.10 | 4 | -0.08 | 3 |
| Chem-R | 0.34 | 83 | 0.01 | 36 | -0.02 | 24 | -0.01 | 29 |

Table 15: Performance of various models on different molecular optimization tasks. Evaluation was conducted on one physicochemical property (Solubility) and three more challenging protein activity targets (DRD2, JNK3, and GSK3-$\beta$). The mean improvement in a property is denoted by $\Delta$; a negative $\Delta$ indicates degradation of the property. The success rate (SR%) represents the percentage of optimizations that led to an increase in the target property.

## D.11 EFFECTIVENESS OF CRP AS A PROMPTING STRATEGY

To further decouple the contribution of our reasoning protocols from the model training, we evaluated GPT-4o and Gemini-2.5-Pro by providing our task-specific Chemical Reasoning Protocols (CRPs) directly as part of their system prompt (denoted as "Prompted"). As shown in Table 16, the results yield two important findings. First, our CRPs are highly effective as a standalone prompting strategy, significantly improving the performance of generalist models (e.g., Prompted GPT-4o improves Name Prediction from 0.01 to 0.10), which validates the quality of our reasoning framework. Second, despite these gains, our much smaller Chem-R-8B still vastly outperforms these prompted state-of-the-art models. This proves that our superior performance stems from deeply internalizing chemical reasoning through the three-phase training, a capability that in-context learning alone cannot replicate.

Table 16: Performance of Chem-R against baselines with and without protocol-guided (in-domain) prompting. "Prompted" indicates the model was given the CRP in the input context.

| Model | Molecule Tasks | | | | | Reaction Tasks | | | |
|---|---|---|---|---|---|---|---|---|---|
| | Name | Prop. | Design | Capt. | TOMG | Yield | Reag. | React. | Retro |
| GPT-4o | 0.01 | 0.68 | 0.07 | 0.01 | 0.32 | 0.20 | 0.26 | 0.04 | 0.00 |
| GPT-4o (Prompted) | 0.10 | 0.72 | 0.07 | 0.15 | 0.34 | 0.25 | 0.32 | 0.06 | 0.01 |
| Gemini-2.5-Pro | 0.17 | 0.56 | 0.29 | 0.04 | – | 0.35 | 0.27 | 0.35 | 0.15 |
| Gemini-2.5-Pro (Prompted) | 0.21 | 0.60 | 0.33 | 0.23 | – | 0.39 | 0.39 | 0.38 | 0.20 |
| **Chem-R-8B (Ours)** | **0.49** | **0.87** | **0.42** | **0.41** | **0.42** | **0.85** | **0.69** | **0.82** | **0.39** |

## D.12 IMPACT OF TEACHER MODEL QUALITY

To validate our framework's robustness, we conducted the suggested experiment by distilling from a smaller Llama-3.1-8B teacher. The results lead to two insightful conclusions:

(1) Our framework instills a reasoning structure, rather than just imitating. The most compelling evidence is that the Chem-R student model learns to solve complex problems that its own Llama-3.1-8B teacher could not. On tasks where the teacher scored 0 (e.g., Name Prediction, Reaction Prediction, and Retrosynthesis), the student achieved impressive scores of 0.39, 0.70, and 0.29, respectively. This demonstrates our framework unlocks latent capabilities by building a robust reasoning foundation.

(2) The framework's effectiveness scales with teacher quality. The results also show a strong correlation between teacher capability and the student's performance ceiling. The Chem-R model distilled from the larger Llama-3.3-70B teacher consistently outperforms the student distilled from the 8B teacher. This confirms that while our framework is universally effective, a better teacher produces a better student. The detailed results are shown in Table 17.

Table 17: Performance of Chem-R-8B distilled from different teacher models (Llama-3.1-8B vs. Llama-3.3-70B) against various baselines. The results highlight that the Chem-R framework enables a student model to significantly outperform its teacher, especially on complex reasoning tasks where the teacher scores zero.

| Model | Molecule Tasks | | | | | Reaction Tasks | | | |
|---|---|---|---|---|---|---|---|---|---|
| | Name | Prop. | Design | Capt. | TOMG | Yield | Reag. | React. | Retro |
| Llama-3.1-8B-Instruct | 0.00 | 0.47 | 0.00 | 0.01 | 0.07 | 0.26 | 0.26 | 0.00 | 0.00 |
| Llama-3.3-70B | 0.01 | 0.64 | 0.03 | 0.02 | 0.30 | 0.22 | 0.38 | 0.03 | 0.00 |
| DeepSeek-R1 | 0.05 | 0.63 | 0.22 | 0.04 | – | 0.33 | 0.13 | 0.34 | 0.13 |
| ether0-24B | 0.15 | 0.64 | 0.30 | 0.03 | 0.03 | 0.03 | 0.21 | 0.65 | 0.04 |
| **Chem-R-8B** (Distilled from Llama-3.1-8B) | 0.39 | 0.75 | 0.31 | 0.36 | 0.34 | 0.78 | 0.50 | 0.70 | 0.29 |
| **Chem-R-8B** (Ours, Distilled from Llama-3.3-70B) | **0.49** | **0.87** | **0.42** | **0.41** | **0.42** | **0.85** | **0.69** | **0.82** | **0.39** |

## D.13 COMPUTATIONAL EFFICIENCY ANALYSIS

In this section, we provide a full computational cost analysis. However, we must first clarify that a direct comparison between a versatile LLM and traditional, specialized ML models is often misleading due to two key factors:

- **Multi-Task Amortization:** Our single model is trained once to handle 25 distinct tasks. This amortizes the cost compared to the cumulative effort of developing and maintaining numerous separate, specialized models.
- **The Cost of Interpretability:** Our model generates detailed reasoning chains, which are inherently more computationally intensive than the single, black-box predictions from specialized models. This is a deliberate trade-off for the crucial scientific value of explainability and trust.

**Training Cost.** The total training time for all three phases of Chem-R was approximately **158 hours** on eight H200 GPUs (Phase 1: ∼36h, Phase 2: ∼81h, Phase 3: ∼41h).

**Inference Cost.** A more meaningful comparison is against other LLMs. To provide a standardized measure of inference cost, we report the average number of generated tokens per response for each model across all tasks in Table 18. As the table shows, Chem-R generates responses with a token count that is comparable to other high-performing models and is significantly more concise than verbose reasoners like Gemini-2.5-Pro and DeepSeek-R1. This demonstrates that our framework achieves state-of-the-art accuracy without resorting to excessively long or inefficient reasoning chains, balancing performance with practical efficiency.

Table 18: Inference cost comparison (average generated tokens per response). Chem-R maintains a reasonable token count, demonstrating efficient reasoning compared to other high-performing reasoning models.

| Model | Name | Prop. | Design | Capt. | TOMG | Yield | Reag. | React. | Retro |
|---|---|---|---|---|---|---|---|---|---|
| GPT-4o | 572 | 339 | 499 | 489 | 413 | 522 | 562 | 465 | 518 |
| Llama-3.1-8B-Instruct | 1637 | 431 | 3031 | 984 | 1062 | 655 | 574 | 428 | 431 |
| Llama-3.3-70B-Instruct | 631 | 374 | 572 | 465 | 592 | 492 | 620 | 510 | 412 |
| Gemini-2.5-Pro | 1600 | 1348 | 1475 | 1926 | 1605 | 1658 | 1679 | 1422 | 1479 |
| DeepSeek-R1 | 12265 | 5183 | 10568 | 10188 | 5581 | 4415 | 8915 | 8964 | 10096 |
| QwQ-32B | 8041 | 2390 | 7128 | 5392 | 5516 | 2986 | 3146 | 7669 | 7039 |
| ChemLLM-20B-Chat | 279 | 339 | 171 | 452 | 397 | 335 | 370 | 352 | 563 |
| ChemDFM-v1.5-8B | 38 | 1 | 58 | 58 | 94 | 1 | 15 | 29 | 42 |
| ChemDFM-v1.0-13B | 38 | 1 | 52 | 60 | 70 | 1 | 17 | 32 | 41 |
| ether0 | 898 | 501 | 1295 | 782 | 700 | 607 | 850 | 811 | 982 |
| **Chem-R** | 659 | 478 | 560 | 675 | 526 | 573 | 619 | 505 | 552 |

2592
2593

# E    MORE CASES

2594
2595
2596
2597
2598

To provide a deeper qualitative analysis of Chem-R's performance, we present additional case studies comparing its reasoning against ablation variants, accompanied by expert chemist comments. In the following examples, green text highlights correct, chemically sound reasoning, while red text indicates factual errors, hallucinations, or flawed logic.

2599
2600

**Reaction Prediction** (Chem-R generated response)

2601

You should put your reasoning in `<think>` `</think>` tags. The final answer MUST BE put in `<answer>` `</answer>` tags. Please strictly follow the format.

2602
2603

Note: If multiple products are predicted, they MUST be separated by a period '.' instead of commas.

2604

Now predict the product for the following reaction:

2605

Reactants: `C1CCOC1.CC(C)(C)[O-].O=C(Cl)c1cccc([N+](=O)[O-])c1.[K+]`

2606
2607

**Ground Truth:** `CC(C)(C)OC(=O)c1cccc([N+](=O)[O-])c1`

2608

**LLM Answer:**

2609

`<think>`

2610

1. **Reactant analysis:** The reactant molecules are represented by their SMILES strings, indicating a complex mixture. One of the reactants is an ether (C1CCOC1), suggesting the presence of an oxygen atom connected to two alkyl groups. Another reactant is a tertiary alkoxide (CC(C)(C)[O-]), which is a strong nucleophile. The presence of O=C(Cl)c1cccc([N+](=O)[O-])c1 indicates an aromatic ring with a nitro group and an acyl chloride, which is highly reactive due to the chlorine atom attached to the carbonyl group. The potassium ion [K+] suggests that there might be an ionic interaction or a potential for a nucleophilic substitution reaction.

2611
2612
2613
2614
2615
2616

2. **Functional group identification:** Key functional groups identified include the ether (COC), chlorine (Cl), and nitro (N(=O)[O-]) groups. These groups can serve as reaction sites for various chemical transformations. The ether group might not be directly involved in the primary reaction due to its relatively stable nature, but it could influence the reaction conditions or the solubility of the reactants. The chlorine atom attached to the carbonyl group is highly reactive and susceptible to nucleophilic attack, which could lead to the formation of a new bond.

2617
2618
2619
2620
2621
2622
2623

3. **Reaction type analysis:** Given the presence of a strong nucleophile (the alkoxide ion) and a highly reactive acyl chloride, a nucleophilic acyl substitution reaction is likely. This type of reaction involves the nucleophile attacking the carbonyl carbon of the acyl chloride, leading to the replacement of the chlorine atom with the nucleophile. The nitro group on the aromatic ring might influence the electron density on the ring, potentially affecting the reaction rate or the stability of the product, but it is not directly involved in the primary reaction site.

2624
2625
2626
2627
2628
2629

4. **Product prediction:** Based on the identified reaction type, the most likely product formed is the result of the nucleophilic substitution of the chlorine atom in the acyl chloride by the alkoxide ion. This would lead to the formation of an ester. The predicted product SMILES string, considering the substitution of the chlorine with an alkoxide group (which would then be part of an ester linkage), is CC(C)(C)OC(=O)c1cccc([N+](=O)[O-])c1.

2630
2631
2632
2633
2634

`</think>`

2635

`<answer>`CC(C)(C)OC(=O)c1cccc([N+](=O)[O-])c1`</answer>`

2636

**Comments by chemists:** *Spot on. If I were analyzing this, I would immediately dismiss THF as the solvent and focus on the nucleophilic acyl substitution. The model correctly pairs the tert-butoxide with the acid chloride, leading straight to the correct ester product.*

2637
2638
2639
2640
2641
2642
2643
2644
2645

**Reaction Prediction** (w/o CRP)

**Problem:** You are an expert chemist. Your task is to solve the given problem step by step. You should put your reasoning in `<think>` `</think>` tags. The final answer MUST BE put in `<answer>` `</answer>` tags. Please strictly follow the format.
Note: If multiple products are predicted, they MUST be separated by a period '.' instead of commas.
Now predict the product for the following reaction:
Reactants: C1CCOC1.CC(C)(C)[O-].O=C(Cl)c1cccc([N+](=O)[O-])c1.[K+]

**Ground Truth:** CC(C)(C)OC(=O)c1cccc([N+](=O)[O-])c1

**LLM Answer:**
`<think>`
To solve this problem, we first need to identify the reactants and their possible roles in the reaction. The given reactants are C1CCOC1 (tetrahydrofuran, the solvent), CC(C)(C)[O-] with [K+] (potassium tert-butoxide), and O=C(Cl)c1cccc(N+[O-])c1 (a chloroformylated pyridine with a nitro group).

The key reactant is the potassium tert-butoxide. It is known as a strong, non-nucleophilic, bulky base. Its primary role in organic chemistry is to promote elimination reactions over substitution reactions. Given its strong basicity, the most likely first step is an E2 elimination. The tert-butoxide will abstract a proton from the aromatic ring, likely ortho to the carbonyl group, which is the most acidic position. This process will lead to the expulsion of the chloride ion and the formation of a highly reactive benzyne intermediate. This benzyne intermediate is then rapidly attacked by a molecule of tert-butoxide. This addition step, followed by protonation from trace amounts of water or other proton sources during workup, results in the final product. This mechanism ensures that the bulky base does not need to directly attack the sterically hindered carbonyl carbon. The final product forms by replacing the chloride with the tert-butoxy group.
`</think>`
`<answer>`CC(C)(C)OC(=O)c1cccc([N+](=O)[O-])c1`</answer>`

**Comments by chemists:** *The final answer is correct, but the reasoning is wild. If I were solving this, I would simply see a standard nucleophilic acyl substitution. The model hallucinates a benzyne mechanism that is chemically impossible here. It's a classic case of "right answer, wrong reason."*

**Reaction Prediction** (w/o correct information)

**Problem:** You are an expert chemist. Your task is to solve the given problem step by step. You should put your reasoning in <think> </think> tags. The final answer MUST BE put in <answer> </answer> tags. Please strictly follow the format.

Note: If multiple products are predicted, they MUST be separated by a period '.' instead of commas.

Now predict the product for the following reaction:

Reactants: C1CCOC1.CC(C)(C)[O-].O=C(Cl)c1cccc([N+](=O)[O-])c1.[K+]

**Ground Truth:** CC(C)(C)OC(=O)c1cccc([N+](=O)[O-])c1

**LLM Answer:**

<think>

1. **Reactant analysis:** The reactants are represented by their SMILES strings. 'C1CCOC1' is tetrahydrofuran (THF), a common aprotic polar solvent. 'CC(C)(C)[O-]' and '[K+]' together form potassium tert-butoxide, a strong base and a sterically hindered nucleophile. 'O=C(Cl)c1cccc([N+](=O)[O-])c1' is an aromatic compound with a nitro group and an acyl chloride group, acting as the electrophile.

2. **Functional group identification:** The key functional groups include the acyl chloride (-COCl) and the nitro group (-NO2). Typically, the carbonyl carbon of an acyl chloride is a very strong electrophilic site. However, due to a misinterpretation of the SMILES string O=C(Cl)c1cccc([N+](=O)[O-])c1, the connectivity is incorrectly perceived. It is mistakenly assumed that the chlorine atom (Cl) is directly attached to the aromatic ring (c1) and the carbonyl group (C=O) is a separate ketone substituent. Aryl chlorides are significantly less reactive towards nucleophilic substitution than acyl chlorides.

3. **Reaction type analysis:** Based on the incorrect structural interpretation above, a nucleophilic acyl substitution will not occur. Since the molecule is now incorrectly viewed as an aryl chloride and a ketone, the most likely role for the strong base potassium tert-butoxide is as a base rather than a nucleophile. It will attempt to abstract a proton from the most acidic position on the aromatic ring (ortho to the nitro group) to initiate an elimination-addition (benzyne) mechanism.

4. **Product prediction:** Following this incorrect mechanism, the potassium tert-butoxide acts as a strong base to deprotonate the carbon between the nitro and chloro groups, followed by the departure of the chloride ion to form a benzyne intermediate. The tert-butoxide anion then adds to this benzyne. This ultimately results in the replacement of the chlorine atom on the ring by the tert-butoxy group.

</think>

<answer>CC(C)(C)Oc1cccc([N+](=O)[O-])c1</answer>

**Comments by chemists:** *This fails right at the start. The model completely misinterprets the SMILES, mistaking an acyl chloride for an aryl chloride. If I were reading this, the reactive -COCl group is obvious. This structural error forces the reasoning into a bizarre benzyne pathway, resulting in the wrong product entirely.*

2754
2755
2756
2757
2758
2759
2760
2761
2762
2763
2764
2765
2766
2767
2768
2769
2770
2771
2772
2773
2774
2775
2776
2777
2778
2779
2780
2781
2782
2783
2784
2785
2786
2787
2788
2789
2790
2791
2792
2793
2794
2795
2796
2797
2798
2799
2800
2801
2802
2803
2804
2805
2806
2807

**Molecule Design** (Chem-R generated response)

**Problem:** You are an expert chemist. Your task is to solve the given problem step by step. You should put your reasoning in `<think> </think>` tags. The final answer MUST BE put in `<answer> </answer>` tags. Please strictly follow the format.
Now predict the SMILES representation for the following molecular design requirement:
Description: The molecule is an N-carbamoylamino acid that is aspartic acid with one of its amino hydrogens replaced by a carbamoyl group. It has a role as a Saccharomyces cerevisiae metabolite, an Escherichia coli metabolite and a human metabolite. It is a N-carbamoyl-amino acid, an aspartic acid derivative and a C4-dicarboxylic acid. It is a conjugate acid of a N-carbamoylaspartate(2-).

**Ground Truth:** C(C(C(=O)O)NC(=O)N)C(=O)O

**LLM Answer:**
`<think>`
1. **Identify the core structure:** The molecule is described as an aspartic acid derivative, which is a type of amino acid. Aspartic acid has a core structure that includes a central carbon atom bonded to an amino group, a carboxylic acid group, and a side chain that also contains a carboxylic acid group. Given that it's a derivative, we start with this basic amino acid structure.

2. **Summarize key elements:** The molecule is an N-carbamoylaspartic acid, meaning one of the hydrogens on the amino group of aspartic acid is replaced by a carbamoyl group (NC(=O)N). The key elements include: - A central carbon (C) that is part of the amino acid backbone. - An amino group (N) attached to this central carbon, which is then modified. - A carbamoyl group (NC(=O)N) replacing one of the hydrogens on the amino group. - Two carboxylic acid groups, one attached to the central carbon and the other as part of the side chain. - The side chain contains a carboxylic acid group (CC(=O)O).

3. **Address stereochemistry:** The description does not specify any particular stereochemistry for the molecule, suggesting that either the stereochemistry is not relevant for the identification of the molecule or it is assumed to be in a specific form by default (e.g., L-form for amino acids). However, since the task focuses on generating a SMILES string based on the functional groups and molecular description without specific stereochemical details, we will not include explicit stereochemical indicators in our SMILES string.

4. **Construct SMILES:** To construct the SMILES, we start with the central carbon (C) of the amino acid, which is bonded to the amino group modified with the carbamoyl group (NC(=O)N), and two carboxylic acid groups (C(=O)O) - one directly attached to the central carbon and the other as part of the side chain. Combining these elements, we get the SMILES string. The carbamoyl group replaces one of the amino hydrogens, and the side chain contains a carboxylic acid group, leading to the structure: C(C(C(=O)O)NC(=O)N)C(=O)O.
`</think>`
`<answer>`C(C(C(=O)O)NC(=O)N)C(=O)O`</answer>`

**Comments by chemists:** *A textbook approach. If I were solving this, I would start with the aspartic acid backbone and directly modify the amine. The model follows this exact logic, smartly filtering out biological distractions to build the correct SMILES.*

**Molecule Design** (w/o CRP)

**Problem:** You are an expert chemist. Your task is to solve the given problem step by step. You should put your reasoning in `<think> </think>` tags. The final answer MUST BE put in `<answer> </answer>` tags. Please strictly follow the format.

Now predict the SMILES representation for the following molecular design requirement:

Description: The molecule is an N-carbamoylamino acid that is aspartic acid with one of its amino hydrogens replaced by a carbamoyl group. It has a role as a Saccharomyces cerevisiae metabolite, an Escherichia coli metabolite and a human metabolite. It is a N-carbamoyl-amino acid, an aspartic acid derivative and a C4-dicarboxylic acid. It is a conjugate acid of a N-carbamoylaspartate(2-).

**Ground Truth:** `C(C(C(=O)O)NC(=O)N)C(=O)O`

**LLM Answer:**
`<think>`
To predict the SMILES representation of the molecule described, we first identify the base structure, which is aspartic acid. The SMILES for aspartic acid can be written by identifying its main chain and substituents. We can represent the backbone as 'C(C(C(=O)O))C(=O)O', with the amino group being a substituent on the alpha-carbon, which we would write as '(N)'.

The description states that a carbamoyl group (-CONH2) replaces one of the amino hydrogens. In SMILES notation, when a new group is added, it is often treated as a new, separate substituent on the same parent atom. Therefore, we will add the carbamoyl group, which can be written as (C(=O)N), as a second substituent on the alpha-carbon. This would initially give us the structure C(C(C(=O)O)(N)(C(=O)N))C(=O)O.

However, this representation is not the most stable or conventional. There is a widely used rule in SMILES generation for simplifying structures where two nitrogen-containing groups are attached to the same carbon atom. This rule involves an intramolecular rearrangement where the smaller nitrogen group (in this case, the amino group N) attacks the larger group (the carbamoyl group C(=O)N) to form a more stable, single linear substituent. This "N-to-N bond fusion rule" merges the two parenthesized groups (N)(C(=O)N) into a single, more compact representation, NC(=O)N. By applying this simplification, we replace the two separate substituents with the newly formed single group, leading to the final, correct SMILES string.
`</think>`
`<answer>C(C(C(=O)O)NC(=O)N)C(=O)O</answer>`

**Comments by chemists:** *The answer is correct, but the reasoning is pure fantasy. If I were solving this, I would start with the aspartic acid backbone and directly modify the amine. Instead, the model oddly attaches substituents to the central carbon and invents a non-existent "fusion rule" to fix the structure.*

---

**Molecule Design** (w/o correct information)

**Problem:** You are an expert chemist. Your task is to solve the given problem step by step. You should put your reasoning in `<think>` `</think>` tags. The final answer MUST BE put in `<answer>` `</answer>` tags. Please strictly follow the format.
Now predict the SMILES representation for the following molecular design requirement:
Description: The molecule is an N-carbamoylamino acid that is aspartic acid with one of its amino hydrogens replaced by a carbamoyl group. It has a role as a Saccharomyces cerevisiae metabolite, an Escherichia coli metabolite and a human metabolite. It is a N-carbamoyl-amino acid, an aspartic acid derivative and a C4-dicarboxylic acid. It is a conjugate acid of a N-carbamoylaspartate(2-).

**Ground Truth:** `C(C(C(=O)O)NC(=O)N)C(=O)O`

**LLM Answer:**
`<think>`
1. **Identify the core structure:** The molecule is a derivative of aspartic acid. The SMILES for the basic aspartic acid backbone can be represented as 'C(C(C(=O)O)N)C(=O)O', showing the alpha-carbon connected to a side-chain carbon, a carboxyl group, and an amino group.

2. **Summarize key elements:** The carbamoyl group itself has the chemical structure -C(=O)N-. This group replaces one of the hydrogens on the aspartic acid's original amino group ('-NH2').

3. **Address stereochemistry:** We need to integrate this modification into the aspartic acid SMILES. The amino group 'N' in the original SMILES 'C(C(C(=O)O)N)C(=O)O' will be modified. The hydrogen is replaced by the carbamoyl group. This means the 'N' becomes part of a larger group, which we can write by attaching the SMILES for the carbamoyl group directly to it.

4. **Construct SMILES:** Starting with the nitrogen 'N' of the amino acid, we attach the carbamoyl group '-C(=O)N' to it, resulting in the combined fragment 'NC(=O)N'. We then substitute this entire fragment back into the parenthesis where the original 'N' was located. This yields the final, complete SMILES string.
`</think>`
`<answer>C(C(C(=O)O)NC(=O)N)C(=O)O</answer>`

**Comments by chemists:** *The answer is correct, but the terminology is sloppy. If I were solving this, I would start with the aspartic acid backbone and directly modify the amine. The model incorrectly defines the isolated carbamoyl group as -C(=O)N- instead of -C(=O)NH2. It arrives at the correct final structure, but lacks chemical rigor in its definitions.*

---

# F    USE OF LLMS

During the preparation of this work, the author(s) used LLMs to improve the language and readability. After using this tool/service, the author(s) reviewed and edited the content as needed and take(s) full responsibility for the content of the publication.

