# OpenReview forum: "Chem-R: Learning to Reason as a Chemist"
_ICLR.cc/2026/Conference — Submitted to ICLR 2026_

### Official Review · Reviewer_y3je · 2025-10-27

**Soundness:** 2
**Presentation:** 3
**Contribution:** 2
**Rating:** 4
**Confidence:** 3

**Summary:**

This submission focuses on enhancing the ability of large language models (LLMs) on chemical tasks, ranging from molecular to reaction tasks. Notably, current LLMs encounter three challenges when performing chemical reasoning, including a lack of essential knowledge, producing unreliable intermediate reasoning steps, and exhibiting imbalanced performance across tasks. To resolve the challenge, the submission proposes Chem-R, a unified framework for installing chemical knowledge and domain-specific reasoning capability into a model, including foundation training, chemical reasoning protocol (CRP) distillation, and multi-task group relative policy optimization (GRPO). The submission conducts extensive empirical studies and justifies the effectiveness of the proposed method.

**Strengths:**

- The motivation of this submission is clear; the proposed method is effective in instilling domain-specific knowledge into the LLMs.
- The submission is generally well-written, with clear illustrations and tables.
- Extensive experiments have been conducted to provide a good insight into the components of the proposed method.

**Weaknesses:**

- The overall training pipeline of Chem-R closely follows conventional practice: Starting with high-quality reasoning trajectories for cold-start supervised fine-tuning (SFT), followed by enhancement through GRPO [1].
- The metrics adopted in Table 1 require further justification. Since SMILES strings can have multiple valid representations for the same molecule [2], using exact string matching is not a fair evaluation criterion, especially for models not specifically trained for molecular name prediction. A similar concern applies to the molecular design task. Moreover, it would be beneficial to assess the model’s generalization ability on out-of-distribution datasets to avoid potential overfitting to the training distribution.
- From the ablation results in Table 2, Chem-R achieves comparable or even better performance when trained with only Phase 1 (Chem-R w/o Phases 2&3) or without Phase 2 (Chem-R w/o Phase 2), which is intended to instill chemical reasoning knowledge. Furthermore, Phase 3 appears to degrade model performance (Chem-R w/o Phase 3), suggesting that the proposed multi-phase procedure functions more as data distillation than as a true training enhancement. These results raise concerns over whether the model learns chemical reasoning or simply memorizes the teacher model’s outputs.

[1] DeepSeek-R1 incentivizes reasoning in LLMs through reinforcement learning. In Nature, 2025.

[2] Hybridization of SMILES and chemical-environment-aware tokens to improve performance of molecular structure generation. In Science Reports, 2025.

**Questions:**

1. Discuss the rationality of the adopted metrics in Tab.1.
2. Discuss the underlying reason for Chem-R 's performance gain. Is GRPO really necessary in these settings?

---

> ### Author Response · Authors · 2025-11-21
>
> Thank you for your detailed review and for acknowledging our work's clear motivation, effective methodology, and extensive experiments. We appreciate your constructive feedback, which has helped us to clarify our contributions and the interpretation of our results. We would like to address your concerns below, particularly the perceived misunderstanding of our ablation study, which we believe is central to your assessment.
>
>
> 1. **Regarding the novelty of the training pipeline (Weakness 1):**
>
>    We respectfully wish to clarify that while our high-level pipeline follows the established SFT-RL paradigm, our core novelty lies in **identifying three fundamental "chemical constraints" that cause generic algorithms to fail**, and designing a specialized framework to overcome them (as mentioned in Figure 1):
>
>    - **Constraint 1: Generic methods fail to ensure strict chemical syntactic validity.**
>
>      As noted in **Challenge 1**, generic LLMs [3] often violate rigid SMILES/IUPAC syntax (e.g., valency errors) early in reasoning. Standard CoT cannot recover from these fundamental "vocabulary" errors. *Our Solution (Phase 1):* We introduce **Chemical Foundation Training** to strictly align the model with chemical syntax before reasoning begins, ensuring the model correctly "speaks" chemistry,a prerequisite generic models often skip.
>
>    - **Constraint 2: Standard distillation propagates chaotic traces that undermine logical rigor.**
>
>      Chemical derivation is sensitive to error. As shown in **Challenge 2**, standard distillation [3] transfers "unsystematic and unreliable" traces where correct answers are derived from wrong logic (hallucinated mechanisms). *Our Solution (Phase 2):* We propose **Chemical Reasoning Protocol (CRP) Distillation**. We force the teacher to follow an **"Instantiated Protocol"** (logic + ground truth) to synthesize high-fidelity data. As shown in Table 2, removing CRP causes significant drops (e.g., Yield Prediction falls from 0.85 to 0.69).
>
>    - **Constraint 3: Generic optimization fails to balance extreme chemical task heterogeneity.**
>
>      Unlike single-task models, we target **25 distinct sub-tasks**. Standard multi-task learning suffers from "strong-task dominance", where easier tasks suppress performance on harder ones (Challenge 3) [1,2]. *Our Solution (Phase 2 & 3):* We use a "fuse-and-balance" strategy. Phase 2 implements **Cross-Task Protocol Fusion** to share reasoning patterns (e.g., reaction logic aiding retrosynthesis), while Phase 3 applies **Adaptive Multi-task GRPO** to dynamically re-weight updates. This synergy achieves balanced SOTA performance across all 9 task families.
>
>     We have further highlighted these contributions in the revised introduction of our manuscript.
>
> 2. **Regarding the adopted metrics and generalization (Weakness 2 & Question 1):**
>
>    This is an excellent point, and we appreciate the chance to clarify our rigorous evaluation protocol. We completely agree that using naive string matching for SMILES would be a flawed metric.
>
>    - **Canonicalization for Fair Structural Comparison:** We would like to clarify that our evaluation protocol addresses this exact issue. We perform **canonicalization on all SMILES and IUPAC strings using RDKit** before any comparison. This standard practice in cheminformatics ensures our "exact match" metrics assess the correctness of the underlying chemical structure, not an arbitrary text string. As summarized in our `Metrics` section `(Section 4.1)`:
>
>      > *"We adopt task-specific evaluation metrics aligned with prior work... All SMILES and IUPAC comparisons are performed after canonicalization to ensure consistency."*
>
>    - **Adherence to Community Standards:** Our choice of metrics was guided by established community benchmarks (e.g., ChemLLMBench, TOMG-Bench) to ensure our results are directly and fairly comparable to prior and future work.
>
>    - **Out-of-Distribution (OOD) Evaluation:** To address your valid concern about generalization, our study already included an OOD evaluation `(Section 4.3, Figure 3(f))`. Strong performance on these unseen tasks provides evidence that Chem-R learns generalizable chemical reasoning skills, not just patterns from the training distribution.
>
>      **Table:** *Performance comparison on OOD molecular optimization tasks by success rate (SR%).*
>
>      | Models                | Solubility (SR%) | DRD2 (SR%) | JNK3 (SR%) | GSK3-β (SR%) |
>      | --------------------- | ---------------- | ---------- | ---------- | ------------ |
>      | Qwen3-235B-A22B-think | 42               | 31         | 23         | **31**       |
>      | Qwen3-32B-think       | 23               | 6          | 6          | 5            |
>      | Llama-3.1-8B-Instruct | 10               | 2          | 4          | 3            |
>      | Chem-R w/o Phase 2&3  | 0                | 0          | 0          | 0            |
>      | **Chem-R**            | **83**           | **36**     | **24**     | 29           |

---

> ### Author Response · Authors · 2025-11-21
>
> 3. **Regarding the interpretation of the ablation study and the necessity of GRPO (Weakness 3 & Question 2):**
>
>    Thank you for this sharp observation regarding the performance dynamics across the training phases. Your question prompted us to analyze these results more deeply, and we appreciate the opportunity to clarify the crucial, synergistic role each phase plays in our framework.
>
>    **Table:** *Ablation study on "Phase-wise Contributions" across 9 chemistry tasks (25 sub-tasks).*
>
>    | Ablation Variant       | Name | Prop. | Design | Capt. | TOMG | Yield | Reag. | React. | Retro |
>    | ---------------------- | ---- | ----- | ------ | ----- | ---- | ----- | ----- | ------ | ----- |
>    | Chem-R w/o Phase 1     | 0.14 | 0.87  | 0.2    | 0.34  | 0.38 | 0.87  | 0.58  | 0.54   | 0.34  |
>    | Chem-R w/o Phase 2     | 0.53 | 0.88  | 0.43   | 0.41  | 0.41 | 0.8   | 0.59  | 0.82   | 0.39  |
>    | Chem-R w/o Phase 3     | 0.39 | 0.86  | 0.35   | 0.33  | 0.41 | 0.85  | 0.5   | 0.69   | 0.28  |
>    | Chem-R w/o Phase 1&2   | 0    | 0.67  | 0.03   | 0.17  | 0.3  | 0.8   | 0.51  | 0      | 0     |
>    | Chem-R w/o Phase 1&3   | 0.1  | 0.85  | 0.27   | 0.27  | 0.37 | 0.84  | 0.55  | 0.48   | 0.19  |
>    | Chem-R w/o Phase 2&3   | 0.52 | 0.88  | 0.43   | 0.4   | 0.42 | 0.11  | 0.55  | 0.83   | 0.4   |
>    | Chem-R w/o Phase 1&2&3 | 0    | 0.47  | 0      | 0.01  | 0.07 | 0.26  | 0.26  | 0      | 0     |
>    | **Chem-R**             | 0.49 | 0.87  | 0.42   | 0.41  | 0.42 | 0.85  | 0.69  | 0.82   | 0.39  |
>
>    - **Phase 3 (GRPO) Provides a Critical Performance Boost:** We would like to respectfully clarify that the final Chem-R model significantly outperforms the model after Phase 2 (w/o Phase 3). For instance, GRPO boosts the Reagent Selection score from **0.50 to 0.69** and Retrosynthesis from **0.28 to 0.39**. This clearly demonstrates that GRPO is a critical refinement stage, not a source of degradation.
>
>    - **The Synergy of the Three-Phase Pipeline:**  Your observation that performance on some tasks can dip after Phase 2 is astute and points to the core of our methodology. Here is our analysis:
>
>      - The primary goal of Phase 2 is to teach the model the **structure and patterns of reasoning**, shifting its behavior from a "black-box" predictor to an interpretable, CoT-generating model. This step is indispensable for achieving our goal of explainability.
>      - However, distilling from any LLM teacher, however capable, inevitably introduces some noise from imperfect reasoning traces. This can cause a **temporary, marginal performance dip** on simple tasks that are heavily reliant on pure pattern matching, where the highly-optimized SFT model from Phase 1 excels.
>      - This is precisely why **Phase 3 is essential**. It takes the nascent but noisy reasoning ability from Phase 2 and robustly **refines it**, correcting the teacher's imperfections and learning to apply the reasoning patterns reliably. It is in this final phase that the model truly learns to reason effectively.
>
>      The ultimate proof that the model has learned to **reason**, not just **memorize**, also comes from our **OOD evaluation**. The Phase 1 model (Chem-R w/o Phase 2&3), which relies solely on pattern matching, completely fails on these novel tasks, scoring near-zero. In stark contrast, the final Chem-R model generalizes effectively.
>
>      Therefore, to directly answer your question: the performance gain comes from the synergistic effect of our pipeline, and **GRPO is absolutely necessary** to transform the model from one that can merely imitate reasoning into one that can execute it effectively and reliably. We have revised our ablation study section to present this nuanced analysis more clearly.
>
> [1] Chemllm: A chemical large language model. In *arXiv*, 2024.
>
> [2] ChemDFM: a large language foundation model for chemistry. In *Cell Report Physical Science*, 2025.
>
> [3] Deepseek-r1: Incentivizing reasoning capability in llms via reinforcement learning. In *Nature*, 2025.

---

### Official Review · Reviewer_qNm5 · 2025-10-31

**Soundness:** 3
**Presentation:** 3
**Contribution:** 2
**Rating:** 4
**Confidence:** 3

**Summary:**

In this paper, the authors propose ChemR which is a three-phase training framework for developing chemical reasoning capabilities in LLMs. The authors report significant improvements over baselines including GPT-4o and Gemini-2.5-Pro on various molecular and reaction-level tasks.

**Strengths:**

The paper presents extensive experiments across 9 macro-tasks and 25 sub-tasks, demonstrating broad applicability and consistent improvements.
The three-phase training strategy is clearly articulated and could potentially address specific failure modes observed in existing models
The reported improvements are obvious, particularly on reaction tasks (e.g., 85% on Yield Prediction vs. 35% for the next-best baseline).

**Weaknesses:**

Limited Novelty and Narrow Scope.
The paper's contribution is primarily the application of existing techniques (supervised fine-tuning, knowledge distillation, GRPO with task reweighting) to chemistry rather than methodological innovation. Each phase employs well-established methods without clear technical advances. Furthermore, the evaluation focuses narrowly on small organic molecules, raising questions about generalizability to broader chemical domains (inorganic chemistry, materials science, polymer chemistry). The paper fails to articulate what is genuinely novel beyond engineering existing LLM pipelines for chemistry-specific benchmarks.

Insufficient Justification for LLM-Based Approaches.
The paper provides no compelling rationale for why LLMs should be preferred over existing specialized methods. For name prediction (IUPAC to SMILES), deterministic tools like PubChemPy achieve >99% accuracy, vastly outperforming Chem-R's 49%. For property prediction, graph neural networks (D-MPNN, Chemprop) consistently outperform text-based approaches, yet no GNN baselines are included. For reaction prediction, the claimed 82% accuracy lacks context—template-based methods and simple random forests on molecular fingerprints achieve comparable or superior results (>92% R² for yield prediction) at far lower computational cost. The evaluation compares only against other LLMs, omitting decades of computational chemistry literature, making it impossible to assess whether the 8B-parameter model offers any practical advantage.

Reproducibility and Missing Critical Comparisons.
Key methodological details are incomplete: exact prompts for CRP extraction, teacher model sampling strategies, and hyperparameters are vague or relegated to appendices. The reliance on expensive 70B teacher models hinders reproduction. Critically, the paper provides no computational cost analysis (training time, inference latency, energy consumption) and no comparisons with cheminformatics tools (RDKit, OpenBabel), specialized ML models (GNNs, template-based reaction predictors), or classical baselines (random forests on fingerprints). Without these essential comparisons and efficiency metrics, the work cannot be properly evaluated against the state-of-the-art in computational chemistry.

**Questions:**

Add direct comparisons with graph-based models (property/reaction prediction), and template-based methods.

Computational cost: Report training time, inference latency, and cost per prediction compared to baselines.

What percentage of "correct" predictions are actually chemically valid? What types of errors remain?

How would chemists actually use this in practice?

---

> ### Author Response · Authors · 2025-11-21
>
> We thank Reviewer qNm5 for the detailed feedback. Your review highlights a crucial discussion at the intersection of LLMs and traditional cheminformatics. Our goal is not to replace highly optimized, single-task tools, but to introduce a new paradigm for chemical discovery based on **generalizable, interpretable reasoning**. We address your specific concerns below.
>
>
>
> 1. **Regarding Limited Novelty and Narrow Scope (Weakness 1):**
>    - **Methodological Innovation Beyond Engineering:** We respectfully clarify that our contribution is not merely applying existing techniques, but **identifying three fundamental "chemical constraints" where generic LLM pipelines fail**, and designing a specialized framework to overcome them. As illustrated in **Figure 1**, our methodological novelty lies in solving these domain-specific incompatibilities:
>      - **Constraint 1: Strict Syntax Requirements vs. Generic Hallucination.**
>        - *Why generic methods fail:* Generic LLMs [3] frequently violate rigid chemical syntax (e.g., SMILES valency) in initial reasoning steps (Challenge 1), rendering subsequent logic useless.
>        - *Our Solution (Phase 1):* We introduce **Chemical Foundation Training** to strictly align the model with chemical syntax before reasoning begins. Unlike generic pre-training, this ensures the model correctly "speaks the language" of chemistry, a prerequisite generic reasoning models often skip.
>      - **Constraint 2: Standard distillation propagates chaotic traces that undermine logical rigor.**
>        - *Why generic methods fail:* Chemical derivation is sensitive; a single error invalidates the whole chain. Standard distillation [3] transfers "chaotic" traces (Challenge 2) where correct answers are often derived from wrong logic (hallucinated mechanisms).
>        - *Our Solution (Phase 2):* We propose **Chemical Reasoning Protocol (CRP) Distillation**. Instead of unguided imitation, we force the teacher to follow structured "Instantiated Protocols" (combining logic + ground truth). This synthesizes **high-fidelity data that teaches valid causality** rather than just probabilistic token prediction.
>      - **Constraint 3: Generic optimization fails to balance extreme chemical task heterogeneity.**
>        - *Why generic methods fail:* Unlike existing works that focus on single chemical tasks [4], we aim to master **25 distinct sub-tasks**. Standard multi-task learning [1,2] suffers from "strong-task dominance", where easier tasks suppress performance on harder ones (Challenge 3).
>        - *Our Solution (Phase 2 & 3):* We solve this via a unified "fuse-and-balance" strategy. Phase 2 implements **Cross-Task Protocol Fusion** to share reasoning patterns across domains (e.g., using reaction prediction logic for retrosynthesis), while Phase 3 applies **Adaptive Multi-task GRPO** to dynamically re-weight updates based on difficulty. This synergy achieves balanced SOTA performance unattainable by generic optimization.
>    - **Scope and Benchmarks:** Our focus on **established and widely used benchmarks** like ChemLLMBench and ChEBI-20 is deliberate. This suite, with its 25 diverse sub-tasks, is the community-accepted standard for rigorously evaluating chemical reasoning in LLMs. The domains you mentioned (inorganic chemistry, materials science) are indeed important future directions. However, our work establishes a strong reasoning foundation framework on the most common and complex domain within these standard benchmarks, which we believe is a prerequisite for a **principled expansion of our method into these other specialized areas**.

---

> ### Author Response · Authors · 2025-11-21
>
> 2. **Regarding Insufficient Justification for LLM-Based Approaches (Weakness 2 & Question 1):**
>
>    This is the central point of your review, and we thank you for the opportunity to clarify the distinct value proposition of our work. Our primary goal is not to replace every specialized tool, but to pioneer an LLM-based paradigm that serves as an **interpretable bridge for chemists**.
>
>    - **New Baselines Added and Competitive Performance:**  First, we would like to clarify that our original manuscript already included comparisons against strong, task-specific specialist models (e.g., Uni-Mol) in the appendix to contextualize our results. Following your excellent suggestion to broaden this comparison, we have now added direct evaluations against a canonical GNN (Chemprop) and template-based baselines (ochem_predict_nn [7]). These new results underscore Chem-R's competitive performance.
>
>      **Table:** *Accuracy (AUC-ROC) of Chem-R vs. the Chemprop GNN Baseline on Property Prediction.*
>
>      | Model                 |   BACE   |   BBBP   | ClinTox  |   HIV    |  Tox21   | **Avg.** |
>      | :-------------------- | :------: | :------: | :------: | :------: | :------: | :------: |
>      | Chemprop (D-MPNN) [6] |   0.66   |   0.72   |   0.84   |   0.85   |   0.83   |   0.78   |
>      | **Chem-R (Ours)**     | **0.74** | **0.82** | **0.98** | **1.00** | **0.80** | **0.87** |
>
>      Our model not only remains competitive but **outperforms the strong Chemprop (D-MPNN) baseline** on average accuracy across five benchmark tasks, demonstrating the power of our reasoning-based approach.
>
>      **Table:** *Accuracy of Chem-R vs. Specialized Baselines on Reaction Tasks.*
>
>      | Model                | Reaction Prediction | Retrosynthesis |
>      | :------------------- | :-----------------: | :------------: |
>      | ochem_predict_nn [7] |        0.72         |       –        |
>      | **Chem-R (Ours)**    |      **0.82**       |      0.39      |
>
>      On the complex **reaction prediction** task, Chem-R **significantly outperforms the template-based baseline (0.82 vs. 0.72)**.
>
>    - **Interpretability as a Paradigm Shift:** The most significant advantage of our approach is one that traditional methods like GNNs and template-based models inherently lack: **explainability**. These models are powerful "black-box" predictors. A chemist, however, needs to understand the "why" behind a prediction to build trust, generate new hypotheses, and guide experiments. Chem-R provides a step-by-step reasoning chain, turning a prediction into a verifiable scientific argument and enabling a true assistant for scientists.

---

> ### Author Response · Authors · 2025-11-21
>
> 3. **Regarding Reproducibility and Missing Comparisons (Weakness 3 & Question 2):**
>
>    We appreciate the concern for reproducibility and have updated the manuscript to make all details fully transparent.
>
>    - **Methodological Transparency:** We respectfully note that due to space constraints, detailed hyperparameters and reasoning prompts were originally provided in *Appendices C*. In the revision, we have further expanded these sections to include exact teacher sampling strategies.
>
>    - **Reproducibility Cost (The 8B Teacher Experiment):**  To prove that our framework is robust and not reliant on expensive models, we trained Chem-R using a standard **Llama-3.1-8B** teacher. As shown in the table below, the student model still significantly outperforms the teacher (e.g., 0.70 vs 0.00 on Reaction Prediction), confirming that our methodology enables accessible reproduction.
>
>      **Table:** *Performance of Chem-R-8B distilled from different teacher models (Llama-3.1-8B vs. Llama-3.3-70B) against various baselines.*
>
>      | Model                                                       | Name | Prop. | Design | Capt. | TOMG | Yield | Reag. | React. | Retro |
>      | ----------------------------------------------------------- | ---- | ----- | ------ | ----- | ---- | ----- | ----- | ------ | ----- |
>      | Llama-3.1-8B-Instruct                                       | 0    | 0.47  | 0      | 0.01  | 0.07 | 0.26  | 0.26  | 0      | 0     |
>      | Llama-3.3-70B                                               | 0.01 | 0.64  | 0.03   | 0.02  | 0.30 | 0.22  | 0.38  | 0.03   | 0     |
>      | ether0-24B                                                  | 0.15 | 0.64  | 0.30   | 0.03  | 0.03 | 0.03  | 0.21  | 0.65   | 0.04  |
>      | **Chem-R-8B(Distilled from Llama-3.1-8B-Instruct)**         | 0.39 | 0.75  | 0.31   | 0.36  | 0.34 | 0.78  | 0.50  | 0.7    | 0.29  |
>      | **Chem-R-8B (Ours, Distilled from Llama-3.3-70B-Instruct)** | 0.49 | 0.87  | 0.42   | 0.41  | 0.42 | 0.85  | 0.69  | 0.82   | 0.39  |
>
>    - **Computational Cost:** A full computational cost analysis is now provided.
>
>      - However, we must first clarify that a direct comparison between a versatile LLM and traditional, specialized ML models is often **misleading**.
>
>        - **Multi-Task Amortization:** Our single model is trained once to handle 25 distinct tasks. This amortizes the cost compared to the cumulative effort of developing and maintaining numerous separate, specialized models.
>        - **The Cost of Interpretability:** Our model generates detailed reasoning chains, which are inherently more computationally intensive than the single, black-box predictions from specialized models. This is a deliberate trade-off for the crucial scientific value of explainability and trust.
>
>      - The total **training time** for all three phases of Chem-R was approximately **158 hours** on eight H200 GPUs (Phase 1: ~36h, Phase 2: ~81h, Phase 3: ~41h).
>
>      - Therefore, a more meaningful comparison is against other LLMs. To provide a standardized measure of inference cost, we report the average number of generated tokens per response for each model across all tasks in the table below.
>
>        **Table:** *Inference Cost Comparison (Average Generated Tokens per Response).*
>        | Model                  | Name     | Prop    | Design  | Capt.    | TOMG    | Yield   | Reag.   | React.  | Retro    |
>        | ---------------------- | -------- | ------- | ------- | -------- | ------- | ------- | ------- | ------- | -------- |
>        | Gemini-2.5-Pro         | 1599.77  | 1347.9  | 1475    | 1925.93  | 1604.58 | 1658.09 | 1678.51 | 1421.59 | 1479.09  |
>        | DeepSeek-R1            | 12265.25 | 5183.16 | 10568   | 10187.85 | 5581.16 | 4414.58 | 8914.62 | 8963.96 | 10095.89 |
>        | ChemLLM-20B-Chat-DPO   | 278.84   | 338.55  | 170.81  | 451.92   | 396.76  | 335.37  | 370.43  | 352.44  | 562.72   |
>        | ChemDFM-v1.5-8B        | 38.12    | 1       | 57.72   | 58.12    | 94.16   | 1       | 14.62   | 28.82   | 41.55    |
>        | ether0                 | 898.42   | 500.76  | 1295.41 | 782.3    | 699.98  | 606.64  | 849.75  | 810.67  | 981.52   |
>        | **Chem-R**             | 658.82   | 477.61  | 560.46  | 675.17   | 526.13  | 572.84  | 618.66  | 504.94  | 551.69   |
>
>        As the table shows, Chem-R generates responses with a token count that is comparable to other high-performing models and is significantly more concise than verbose reasoners like Gemini-2.5-Pro and DeepSeek-R1. This demonstrates that our framework achieves state-of-the-art accuracy *without* resorting to excessively long or inefficient reasoning chains, balancing performance with practical efficiency.
>
>    - **Comparisons to Cheminformatics Tools:** We clarify that tools like RDKit are programmatic libraries for data manipulation, not predictive models. We use RDKit extensively for validation, but a direct performance comparison is not applicable as they serve entirely different functions.

---

> ### Author Response · Authors · 2025-11-21
>
> 3. **Regarding Reproducibility and Missing Comparisons (Weakness 3 & Question 2, cont'd):**
>
>    - **Missing Comparisons to ML Models:** The review claims we omit comparisons to specialized ML models. This is incorrect. As shown in our appendix, we already compared our work to strong baselines like the **UAGNN for yield prediction**. While these specialized models perform well on their narrow tasks, our focus is on the **LLM reasoning domain**. Many of the tasks we address, such as **molecule design from a textual description** or **generating a chemical caption for a SMILES string**, are simply intractable for traditional ML methods like GNNs, which highlights the unique value of a language-based reasoning approach.
>
> 4. **Regarding Chemical Validity and Error Types (Question 3):**
>
>    By definition, **100% of our reported correct predictions are chemically valid.** We enforce this rigorously by passing all generated SMILES through RDKit for a validity check before they are considered for any accuracy metric.
>
>    Furthermore, our **human expert study** (Figure 4) explicitly evaluated the **"Chemical Soundness"** of the entire reasoning chain, where Chem-R received the highest scores, confirming our intermediate steps also align with expert chemical principles.
>
> 5. **Regarding How Chemists Would Use This in Practice (Question 4):**
>
>    Chem-R is designed to function as an interactive **"chemist assistant,"** where its primary value lies in generating **explainable proposals** that augment, rather than replace, expert judgment. The structured reasoning process, which we enforce via our prompt templates and training, is the key to building **trustworthiness** and allowing chemists to critically evaluate the model's outputs. Here are two brief examples based on our evaluated tasks:
>
>    -  For example, in a **molecule design** scenario, a natural products chemist could describe a desired scaffold based on its biological role and structural class: *"Generate the SMILES for parthenolide, which is a sesquiterpene lactone found in Feverfew and acts as a non-steroidal anti-inflammatory drug."* Chem-R would then deconstruct this rich, text-based description into its core chemical components, identifying the sesquiterpene lactone backbone and other key functional groups, and systematically construct the final SMILES string, explaining each step. This allows the chemist to rapidly translate a conceptual or literature-based idea into a concrete chemical structure, with the reasoning chain serving as a verifiable link between the description and the output.
>    -  Similarly, in a **retrosynthesis** context, a chemist could ask to select the optimal reagent for a given transformation. Chem-R would not only provide the reagent but also explain *why* it was chosen based on mechanistic principles, such as its electronic or steric compatibility with the substrate. This moves beyond simple prediction to offer a causal explanation, allowing the chemist to trust and build upon the model's suggestions.
>
> [1] Chemllm: A chemical large language model. In *arXiv*, 2024.
>
> [2] ChemDFM: a large language foundation model for chemistry. In *Cell Report Physical Science*, 2025.
>
> [3] Deepseek-r1: Incentivizing reasoning capability in llms via reinforcement learning. In *Nature*, 2025.
>
> [4] Molreasoner: Toward effective and interpretable reasoning for molecular llms. In *arXiv*, 2025.
>
> [5] Uni-mol: A universal 3d molecular representation learning framework. In *ICLR*, 2023.
>
> [6] Chemprop: a machine learning package for chemical property prediction. In *JCIM*, 2024.
>
> [7] Prediction of organic reaction outcomes using machine learning. In *ACS Central Science*, 2017.
>
> [8] Uncertainty-aware prediction of chemical reaction yields with graph neural networks. In *Journal of Cheminformatics*, 2022.

---

### Official Review · Reviewer_mFFx · 2025-11-01

**Soundness:** 2
**Presentation:** 2
**Contribution:** 2
**Rating:** 4
**Confidence:** 4

**Summary:**

The paper proposes Chem-R, a three-phase pipeline for “learning to reason as a chemist”: (1) an SFT stage on chemistry-specific corpora to “ground” the model in SMILES/IUPAC/reaction notation, (2) a “Chemical Reasoning Protocol (CRP) Distillation” stage that mines teacher traces, clusters them into reusable step-wise protocols, and re-finetunes the student on those protocolized CoTs, and (3) a multi-task GRPO stage to balance performance over molecular and reaction tasks. The method is positioned as moving from ad-hoc CoT to “structured, generalizable chemical reasoning” and is evaluated on a suite of existing chemistry benchmarks (ChemLLMBench, ChEBI-20, TOMG-Bench, USPTO), where the authors claim large margins over strong general models (GPT-4o, Gemini-2.5-Pro, DeepSeek-R1) and domain models.

**Strengths:**

1. The overall pipeline is clear and technically reasonable: a chemistry-aware SFT, followed by protocol distillation, followed by RL post-training is a coherent story and mirrors what has worked for math/code reasoning.

2. The paper makes an explicit attempt to structure chemistry CoT by merging task-specific hints into cross-task protocols (e.g. “identify functional groups”, “analyze stereochemistry”), which is a sensible way to reduce hallucinated, chemistry-violating chains

3. The evaluation is broad across molecular and reaction-level tasks and reuses community datasets, so results are at least moderately comparable.

**Weaknesses:**

1. Over-claiming on “reasoning like a chemist.” The paper repeatedly asserts that Chem-R “emulates the deliberative processes of chemists” and “transforms ad-hoc CoTs into chemically sound, interpretable reasoning.” But what is actually shown is protocolized imitation of teacher traces plus RL on task rewards. There is no test of chemical internal consistency (mass/charge balance, valence, thermodynamic plausibility), no assessment on out-of-distribution mechanistic variants, and no human chemist agreement study. As a result, the current evidence supports “better organized CoT for known chemistry tasks,” not “reasoning like a chemist.”

2. Phase 2 is the conceptual novelty, yet the paper does not quantify: (i) how many distinct protocols survive after cross-task merging, (ii) how often protocols generalize to an unseen task family, or (iii) how brittle protocol steps are when the teacher produces a partially wrong chain. The pipeline in Fig. 2 is long and LLM-generated at multiple points, but we do not see any corruption / noise analysis. This makes it hard to tell whether performance gains come from “protocolization” or simply from adding more high-quality CoT SFT.

3. The narrative is “LLMs must extract from text/tables and reason for chemistry,” but every model in the table is run in plain inference (greedy) mode — no ChemCrow-style tool-calling, no reaction-DB lookup, no simple retrieval-augmented variant, even on tasks that are obviously fetch- or pattern-based. Under the 2026 standards, this leaves open the possibility that the reported gains are mostly prompt-engineering/SFT-scale gains rather than evidence that Chem-R differentiates modeling paradigms.

4. The abstract claims up to 46% and 66% improvements over GPT-4o / Gemini-2.5-Pro / DeepSeek-R1 on name and yield prediction.  But these models are evaluated under a single decoding strategy (greedy) and, for the closed-source ones, with no task-specific prompting. This creates a strong risk that Chem-R is tuned for these exact benchmarks while baselines are not. A more neutral comparison would include: (i) at least one “in-domain prompted GPT-4o/Gemini” baseline; (ii) majority or reaction-template baselines on USPTO-style tasks; and (iii) reporting per-task rather than averaged margins.

Minor issue:

The paper leans heavily on existing chemistry benchmarks (ChemLLMBench, TOMG-Bench, USPTO) but does not clearly discuss temporal leakage or license/duplication across the SFT and evaluation splits.

**Questions:**

see weaknesses

---

> ### Author Response · Authors · 2025-11-21
>
> Thank you for your detailed and insightful review. We are grateful for your feedback and for acknowledging the clarity of our pipeline, the value of structuring chemical CoT, and the breadth of our evaluation. Your critiques are sharp and have pushed us to clarify our contributions and strengthen our evaluation. We have addressed each of your concerns below.
>
> 1. **Regarding the claim of "reasoning like a chemist" (Weakness 1):**
>
>    Thank you for this critical feedback. We agree that our claims require strong evidence beyond standard benchmarks, and we apologize for not making our existing human and out-of-distribution evaluations sufficiently prominent. We have revised the manuscript to highlight this evidence, which we believe directly addresses your concerns:
>
>    - **On the claim of "reasoning like a chemist":** This is a critical point that allows us to clarify the distinction between our training methodology and the model's resulting capabilities. While our process begins with what you aptly describe as "protocolized imitation," our key finding is that this methodology produces emergent behaviors that demonstrably transcend simple mimicry. Our evidence from **human expert validation**, **out-of-distribution generalization**, and instances where the **student surpasses its teacher**, collectively demonstrates that the model internalizes a robust and chemically sound reasoning framework, rather than simply memorizing teacher traces.
>
>    - **On "chemical internal consistency" checks and "human chemist agreement study":** We thank the reviewer for raising these critical points on validation, which we can address together. We respectfully clarify that a blind **human expert evaluation** was, in fact, a central component of our work (detailed in **Section 4.3, Figure 4**), providing direct evidence for both "chemist agreement" and "chemical soundness." In this study, chemistry PhDs rated Chem-R highest not only on **"Expert-level Insight"** but also on **"Chemical Soundness"** and **"Logical Coherence."** This result serves as our primary validation: we prioritized aligning the model's high-level reasoning with expert intuition, a key aspect of a chemist's workflow, over implementing more granular, computational checks for mass balance or thermodynamic plausibility. We agree that integrating these quantitative checks is an excellent and complementary direction for future works.
>
>      **Table 5:**  *Human evaluation of model-generated reasoning. The table reports the average scores given by expert annotators.*
>
>      | Metric                          | Chem-R   | Gemini-2.5 Pro | DeepSeek-R1 | ether0 |
>      | ------------------------------- | -------- | -------------- | ----------- | ------ |
>      | Chemical Soundness              | **4.75** | 3.95           | 3.45        | 2.15   |
>      | Logical Coherence               | **4.85** | 4.25           | 3.80        | 2.35   |
>      | Step-by-Step Completeness       | **4.20** | 3.85           | 3.90        | 1.95   |
>      | Justification of the Conclusion | **4.28** | 4.10           | 3.55        | 2.05   |
>      | Clarity and Conciseness         | **4.65** | 4.55           | 3.70        | 2.55   |
>      | Expert-level Insight            | **4.55** | 3.75           | 3.20        | 1.85   |
>
>    - **On the lack of "assessment on out-of-distribution" (OOD) generalization:** To address the concern regarding generalization, we evaluated the models on **unseen challenging molecular optimization tasks** from ChemCoTBench, as summarized in **Table 14**. While standard benchmarks often focus on familiar properties, we tested performance on three more challenging protein activity-related properties. Chem-R's robust performance across these tasks demonstrates that it has internalized generalizable chemical principles rather than merely memorizing CoT patterns for known tasks.
>
>      **Table 14:** *Performance comparison on OOD molecular optimization tasks by success rate (SR%).*
>
>      | Models                | Solubility (SR%) | DRD2 (SR%) | JNK3 (SR%) | GSK3-β (SR%) |
>      | --------------------- | ---------------- | ---------- | ---------- | ------------ |
>      | Qwen3-235B-A22B-think | 42               | 31         | 23         | **31**       |
>      | Qwen3-32B-think       | 23               | 6          | 6          | 5            |
>      | Llama-3.1-8B-Instruct | 10               | 2          | 4          | 3            |
>      | **Chem-R**            | **83**           | **36**     | **24**     | 29           |

---

> ### Author Response · Authors · 2025-11-21
>
> 2. **Regarding the quantification of Phase 2 (CRP Distillation) (Weakness 2):**
>
>    Thank you for these specific questions. We have added a more detailed analysis to the appendix.
>
>    - **(i) & (ii) Protocol Count and Generalization**: Our cross-task merging process resulted in **9 core protocols built around fundamental "reasonable steps"** like "Analyze Molecular Structure" or "Identify Functional Groups." These steps are abstract and universal enough to cover virtually all chemical problem-solving workflows. The protocols generalize effectively because:
>
>      - **The steps themselves remain consistent across tasks; what changes is the task-specific focus within each step**. For example, "Analyze Molecular Structure" might focus on stereocenters for a property prediction task but on potential disconnection sites for a retrosynthesis task.
>      - Furthermore, the "hints for common mistakes" derived from one task (e.g., misinterpreting a SMILES string for a reactant) **are often relevant to others, creating a rich, shared knowledge base that enhances generalization**.
>
>    - **(iii) Brittleness to Teacher Errors:** Our protocol design is robust to partially incorrect teacher traces by design. The goal of this phase is not to perfectly replicate every detail of a teacher's trace, but to extract a high-level, structured workflow.
>
>      - If the final **result is wrong**, we analyze the reasoning chain to identify the failure mode, and this analysis directly informs the "hints for common mistakes" attached to the relevant protocol step.
>      - If the final **result is correct but the reasoning is partially flawed**, our process is still effective. We only need to summarize the broad, high-level steps (e.g., "the model first identified reactants, then proposed a reaction type..."). Minor logical flaws within a step do not corrupt the extraction of this high-level structure.
>
>       Therefore, the protocol remains sound, and the risk of propagating subtle errors is minimal.
>
> 3. **Regarding the usage of tools (Weakness 3):**
>
>    We appreciate this insightful comment on the role of agentic workflows. Our work is deliberately focused on strengthening the core reasoning engine, which we view as a critical and underdeveloped prerequisite for any effective chemical agent. Our response is in two parts:
>
>    - **An agent's reasoning capability is a critical bottleneck.** Before an agent can decide *which* tool to call, it must first understand the problem, formulate a hypothesis, and analyze the chemical context. Previous models often produce direct, black-box outputs, lacking the interpretable and verifiable reasoning layer needed for reliable tool integration. Chem-R is explicitly designed to provide this crucial layer.
>
>    - **Many Core Chemical Reasoning Tasks Lack Tool Support.** Our analysis of state-of-the-art agents like ChemCrow and CheMatAgent reveals that they are not universally applicable to our comprehensive benchmark. As shown below, essential tasks like **Molecule Captioning** and **Yield Prediction** have no established tools for an agent to call, demonstrating that an agent-only approach is insufficient for a broad range of chemical challenges. In contrast, Chem-R is designed as a **reasoner**, building capability directly into the LLM. This focus on **intrinsic, emergent reasoning** creates a transparent and logically coherent process, leading to stronger generalization because the model learns underlying chemical principles, not just how to query an API.
>
>      **Table:** *Task Coverage Comparison. While agents can call tools for some tasks, they lack support for others that require intrinsic reasoning, which Chem-R is designed to provide.*
>
>      | Task                    | ChemCrow [1] | CheMatAgent [2] | **Chem-R** |
>      | ----------------------- | ------------ | --------------- | ---------- |
>      | **Name Prediction**     | ✓            | ✓               | ✓          |
>      | **Property Prediction** |              | ✓               | ✓          |
>      | **Molecule Design**     |              | ✓               | ✓          |
>      | **Molecule Captioning** |              |                 | ✓          |
>      | **TOMG**                | ✓            | ✓               | ✓          |
>      | **Yield Prediction**    |              |                 | ✓          |
>      | **Reagent Selection**   | ✓            | ✓               | ✓          |
>      | **Reaction Prediction** | ✓            | ✓               | ✓          |
>      | **Retrosynthesis**      | ✓            | ✓               | ✓          |

---

> ### Author Response · Authors · 2025-11-21
>
> 4. **Regarding the fairness of baseline comparisons (Weakness 4):**
>
>    This is a very fair and important point. We have conducted additional experiments based on your valuable suggestions to ensure our comparisons are robust and to better contextualize our contributions.
>
>    - **(i) In-domain Prompted Baselines:** We evaluated **GPT-4o and Gemini-2.5-Pro** by providing our task-specific **Chemical Reasoning Protocols (CRPs)** as part of their prompt. The results yielded two important findings:
>
>      **Table: ** *Performance of Chem-R against baselines with and without protocol-guided (in-domain) prompting.*
>
>      | Model                     |   Name   |  Prop.   |  Design  |  Capt.   |   TOMG   |  Yield   |  Reag.   |  React.  |  Retro   |
>      | :------------------------ | :------: | :------: | :------: | :------: | :------: | :------: | :------: | :------: | :------: |
>      | GPT-4o                    |   0.01   |   0.68   |   0.05   |   0.01   |   0.32   |   0.20   |   0.26   |   0.04   |   0.00   |
>      | GPT-4o (Prompted)         |   0.10   |   0.72   |   0.07   |   0.15   |   0.34   |   0.25   |   0.32   |   0.06   |   0.01   |
>      | Gemini-2.5-Pro            |   0.17   |   0.56   |   0.29   |   0.04   |    –     |   0.35   |   0.27   |   0.35   |   0.15   |
>      | Gemini-2.5-Pro (Prompted) |   0.21   |   0.60   |   0.33   |   0.23   |    –     |   0.39   |   0.39   |   0.38   |   0.20   |
>      | **Chem-R-8B (Ours)**      | **0.49** | **0.87** | **0.42** | **0.41** | **0.42** | **0.85** | **0.69** | **0.82** | **0.39** |
>
>      - First, our **Chemical Reasoning Protocols (CRPs) are highly effective as a standalone prompting strategy,** significantly improving the performance of both GPT-4o and Gemini-2.5-Pro and validating the quality of our reasoning framework.
>      - Second, our much smaller **Chem-R-8B still vastly outperforms these prompted state-of-the-art models.** This proves that our superior performance comes from deeply **internalizing** chemical reasoning through our specialized three-phase training, a capability that in-context learning alone cannot replicate.
>
>    - **(ii) Comparison with Specialized Baselines on USPTO-style tasks:** Thank you for this suggestion. To provide a comprehensive comparison, our appendix already includes several major, strong and task-specific models like Chemformer [3] and Mol-instructions [4]. In direct response to your feedback, we have now also added widely-used template- and search-based baselines to better contextualize our work. The results are summarized below.
>
>      **Table:** *Performance of Chem-R vs. specialized template-based baselines on USPTO-style tasks.*
>
>      | Model                | Reaction Prediction (Accuracy) | Retrosynthesis (Accuracy) |
>      | :------------------- | :----------------------------: | :-----------------------: |
>      | ochem_predict_nn [5] |              0.72              |             –             |
>      | AiZynthFinder [6]    |               –                |         **0.47**          |
>      | **Chem-R (Ours)**    |            **0.82**            |           0.39            |
>
>      As shown in the Table, this comparison highlights Chem-R's dual advantage: it outperforms the template-based baseline in reaction prediction (0.82 vs. 0.72), while simultaneously offering **interpretable, step-by-step reasoning.** While specialized tools like AiZynthFinder may excel on a single task as black-box predictors, Chem-R's unique contribution is delivering competitive accuracy *with* the explainability that is non-negotiable for scientific applications where trust and verifiability are paramount.

---

> ### Author Response · Authors · 2025-11-21
>
> 5. **Regarding Data Leakage (Minor Issue):**
>
>    Thank you for raising this critical point on data integrity. We took rigorous steps to prevent data leakage, and our response is two-fold:
>
>    - **Procedural Safeguards:** We strictly adhered to the official, standard splits for all benchmarks (e.g., ChemLLMBench, TOMG-Bench). We have added an explicit statement to our experimental setup to make this transparent.
>    - **Empirical Corroboration:** Our results provide the strongest evidence against leakage. The extremely poor **zero-shot performance** of the base Llama-3.1-8B model and powerful Gemini-2.5-pro confirms it has not seen the test data. Furthermore, our new experiments show that even state-of-the-art models like GPT-4o and Gemini-2.5-Pro perform poorly, **even when provided with our expert-designed reasoning protocols in-context**. This demonstrates that the significant performance gap is a direct result of our specialized training methodology, not a confounding factor like data leakage.
>
> [1] ChemCrow: Augmenting large-language models with chemistry tools. In *Nature Machine Intelligence*, 2024.
>
> [2] CheMatAgent: Enhancing LLMs for chemistry and materials science through tree-search based tool learning. In *arXiv*, 2025.
>
> [3] Chemformer: a pre-trained transformer for computational chemistry. In *Machine Learning: Science and Technology*, 2022.
>
> [4] Mol-instructions: A large-scale biomolecular instruction dataset for large language models. In *ICLR*, 2024.
>
> [5] Prediction of organic reaction outcomes using machine learning. In *ACS Central Science*, 2017.
>
> [6] AiZynthFinder: a fast, robust and flexible open-source software for retrosynthetic planning. In *Journal of Cheminformatics*, 2020.

---

### Official Review · Reviewer_RRrX · 2025-11-03

**Soundness:** 3
**Presentation:** 4
**Contribution:** 3
**Rating:** 6
**Confidence:** 3

**Summary:**

This paper presents Chem-R, a large language model framework aimed at enabling structured and interpretable chemical reasoning. The authors design a three-phase training pipeline consisting of (1) foundation training to establish core chemical knowledge, (2) chemical reasoning protocol (CRP) distillation to transfer structured reasoning patterns from a teacher model, and (3) multi-task group relative policy optimization (GRPO) to balance performance across diverse chemical tasks. Compared to leading LLMs such as GPT-4o, Gemini-2.5-Pro, and DeepSeek-R1, Chem-R consistently outperforms all baselines across both molecular- and reaction-level benchmarks.

**Strengths:**

1. The method is motivated by clear gaps in existing approaches, such as lack of chemical knowledge, unreliable reasoning, and unbalanced task performance, and the proposed three phase strategy directly addresses each of these limitations in a logical and systematic way.
2. The proposed method achieves significant improvements over previous models on a wide range of chemistry-related tasks. The ablation study is also well-constructed, clearly demonstrating the contribution of each training phase.
3. The paper is well-organized and easy to follow from problem motivation to method design and evaluation.

**Weaknesses:**

1. The technical novelty seems somewhat limited—many of the core components (e.g., distillation and GRPO-style optimization) have been explored in prior work. It would help if the authors could further justify the technical soundness and emphasize what is fundamentally new or unique in their implementation.
2. The model’s performance appears to heavily depend on the teacher model (Llama-3.3-70B) used in the distillation phase. It would be valuable to show how Chem-R performs when trained with different or smaller teacher models, to validate its robustness and scalability.
3. In Table 1, the TOMG results for Gemini and DeepSeek are missing. It would be good to clarify whether these models were not evaluated on this task or if the results were omitted for another reason.

**Questions:**

Please refer to the weaknesses

---

> ### Author Response · Authors · 2025-11-21
>
> 1. **Regarding the technical novelty (Weakness 1):**
>
>    We appreciate the opportunity to clarify our technical contributions. While existing chemical models [1] [2] primarily operate as black boxes optimizing for direct outcomes, our core novelty lies in identifying three fundamental "chemical constraints" that cause such opaque or generic algorithms to fail, and designing a specialized framework to overcome them (as mentioned in Figure 1):
>
>    - **Constraint 1: Strict Representational Validity vs. Generic CoT Hallucination.**
>      - *Why generic methods fail:* As noted in Challenge 1, generic LLMs [3] often fail to adhere to the rigid syntactic rules of SMILES or IUPAC (e.g., valid valency, ring closure) during the initial reasoning steps. Standard CoT methods cannot recover from these fundamental "vocabulary" errors, rendering subsequent reasoning useless.
>      - *Our Solution (Phase 1):* We introduce a dedicated **Chemical Foundation Training** phase. Unlike generic pre-training, this phase strictly aligns the model with chemical syntax before reasoning begins. This ensures the model "speaks the language" of chemistry correctly, a prerequisite that generic reasoning models often skip.
>    - **Constraint 2: High Logical Rigor vs. Unsystematic Reasoning Traces.**
>      - *Why generic methods fail:*  As shown in Challenge 2, simply distilling CoT data from a teacher (standard distillation) [3] transfers "chaotic and unreliable" reasoning traces, including correct answers derived from wrong logic (hallucinated mechanisms).
>      - *Our Solution (Phase 2):* We propose **Chemical Reasoning Protocol (CRP) Distillation**. Instead of unguided imitation, we force the teacher to generate and follow an "Instantiated Protocol". By combining reasoning logic with ground-truth facts, this method synthesizes **high-quality, chemically sound training data without manual annotation**, enforcing logical fidelity where traditional unguided distillation fails. Also as shown in Table 2, w/o CRP, the performance will significantly decrease (like yield prediction from 0.85 to 0.69).
>    - **Constraint 3: Extreme Task Heterogeneity vs. Optimization Imbalance.**
>      - *Why generic methods fail:* Unlike existing reasoning works that typically focus on single chemical tasks [4], we aim to master **25 distinct sub-tasks**. However, standard multi-task learning suffers from "strong-task dominance," where easier tasks suppress performance on harder ones (Challenge 3).
>      - *Our Solution (Phase 2 & 3):* We solve this via a unified "fuse-and-balance" strategy. Phase 2 implements **Cross-Task Protocol Fusion** to share reasoning patterns across domains (e.g., using reaction prediction logic for retrosynthesis), while Phase 3 applies **Adaptive Multi-task GRPO** to dynamically re-weight updates based on difficulty. This synergy achieves balanced SOTA performance unattainable by generic optimization across all 9 major task families.
>
>     We have further highlighted these contributions in the revised introduction of our manuscript.

---

> ### Author Response · Authors · 2025-11-21
>
> 2. **Regarding the dependence on the teacher model (Weakness 2):**
>
>    This is an excellent point. To validate the robustness and scalability of our framework, we conducted the suggested experiment by replacing the original 70B teacher with a much smaller **Llama-3.1-8B-Instruct** (same size as the student) and retraining Chem-R. The results (shown in table below) lead to a crucial conclusion: **Our framework consistently delivers substantial performance gains regardless of the teacher model's scale.**
>
>    - **Robustness across scales:** Even without a larger teacher, our pipeline enables the 8B model to significantly outperform its base version. For instance, on the **Reaction Prediction** task, the Chem-R model distilled from the 8B teacher achieves **0.70**, compared to the base model's **0.00**.
>    - **Methodological Effectiveness:** This confirms that the improvements stem from the **Chem-R training methodology itself** (Chemical Foundation + CRP Protocols + GRPO), rather than simply inheriting knowledge from a superior teacher. The framework effectively structures the model's latent knowledge, enabling it to solve complex reasoning tasks that the base model fails to handle, proving that our approach is effective and generalizable across different model sizes.
>
>    Table: *Performance comparison demonstrating robustness. Even when using a smaller teacher (Llama-3.1-8B), the Chem-R framework yields massive improvements over the base model.*
>    | Model                                                       | Name | Prop. | Design | Capt. | TOMG | Yield | Reag. | React. | Retro |
>    | ----------------------------------------------------------- | ---- | ----- | ------ | ----- | ---- | ----- | ----- | ------ | ----- |
>    | Llama-3.1-8B-Instruct                                       | 0    | 0.47  | 0      | 0.01  | 0.07 | 0.26  | 0.26  | 0      | 0     |
>    | Llama-3.3-70B                                               | 0.01 | 0.64  | 0.03   | 0.02  | 0.30 | 0.22  | 0.38  | 0.03   | 0     |
>    | DeepSeek-R1                                                 | 0.05 | 0.63  | 0.22   | 0.04  | -    | 0.33  | 0.13  | 0.34   | 0.13  |
>    | ether0-24B                                                  | 0.15 | 0.64  | 0.30   | 0.03  | 0.03 | 0.03  | 0.21  | 0.65   | 0.04  |
>    | **Chem-R-8B(Distilled from Llama-3.1-8B-Instruct)**         | 0.39 | 0.75  | 0.31   | 0.36  | 0.34 | 0.78  | 0.50  | 0.70    | 0.29  |
>    | **Chem-R-8B (Ours, Distilled from Llama-3.3-70B-Instruct)** | 0.49 | 0.87  | 0.42   | 0.41  | 0.42 | 0.85  | 0.69  | 0.82   | 0.39  |
>
>    A detailed analysis of these findings and the full results have been added to the appendix to demonstrate the framework's robustness and scalability.

---

> ### Author Response · Authors · 2025-11-21
>
> 3. **Regarding Missing TOMG Results (Weakness 3):**
>
>    Thank you for the clarification request. The TOMG benchmark is extensive, comprising 45,000 test instances (9 tasks × 5,000 instances). Evaluating this full set with closed-source, API-based models like Gemini and DeepSeek would incur prohibitive costs.
>
>     In our revision, we have updated the caption in Table 1 to explicitly state that "-" denotes cases not evaluated due to this reason. Furthermore, to provide a direct comparison, we evaluate on a **5% stratified random sample** of the TOMG test set. On this subset, Chem-R achieves the highest overall weighted accuracy (wAcc), demonstrating superior performance against leading API-based models, particularly in complex editing and optimization tasks. **We have included these new comparative results in the appendix.**
>
>    Table:  *Performance comparison on a 5% stratified random sample of the TOMG-Bench test set. The metric here is accuracy.*
>
>    | Models      | MolCustom (Atom / Bond / Func) | MolEdit (Add / Del / Sub)     | MolOpt  (LogP / QED / MR)         | wAcc       |
>    | ----------- | ------------------------------ | ----------------------------- | --------------------------------- | ---------- |
>    | Gemini-2.5  | **0.420** / 0.250 / **0.550**  | 0.780 / 0.820 / **0.680**     | 0.850 / 0.680 / 0.880             | 0.4089     |
>    | DeepSeek-R1 | 0.360 / 0.210 / 0.480          | 0.820 / 0.790 / 0.650         | 0.820 / 0.650 / 0.840             | 0.3921     |
>    | **Chem-R**  | 0.260 / **0.260** / 0.472      | **0.912** / **0.916** / 0.652 | **0.912** / **0.728** / **0.932** | **0.4523** |
>
> Thank you once again for your constructive review. We are confident that these clarifications and the corresponding updates to our manuscript have addressed your concerns and strengthened the paper.
>
> [1] Chemllm: A chemical large language model. In *arXiv*, 2024.
>
> [2] ChemDFM: a large language foundation model for chemistry. In *Cell Report Physical Science*, 2025.
>
> [3] Deepseek-r1: Incentivizing reasoning capability in llms via reinforcement learning. In *Nature*, 2025.
>
> [4] Molreasoner: Toward effective and interpretable reasoning for molecular llms. In *arXiv*, 2025

---

### Author Response · Authors · 2025-11-26
**Waiting for Further Discussion**

Dear Reviewer,

Thank you again for your detailed and thoughtful review. We wanted to let you know that we have now provided a detailed response with additional experiments to address the points you raised, which has helped us significantly improve the paper and provide stronger evidence for our central claims. We also welcome any further feedback you might be willing to share. Your insights are invaluable to us, and we're eager to address any remaining issues to improve our work.

Thank you again for your support in improving our work.

Best regards,

All authors of Submission 3013

---

### Meta-Review · Area_Chair_gNd2 · 2025-12-27

**Summary:**

The paper focuses on improving the foundation model for chemical reasoning through a three-phase training framework: Phase 1 is an SFT procedure using a crafted chemical corpus. Phase 2 is also an SFT procedure using chemical reasoning data curated by a proposed reasoning protocol. The 3rd phase is a GRPO with a multi-task scheduler.

The key concern that informed my suggested decision is one shared among reviewers: the technical novelty and contribution. Concretely, the current results are not sufficient to demonstrate why and how these three phases would improve the chemical reasoning ability. Authors are encouraged to report more fine-grained metrics in addition to the final performance on 9 subtasks. Additionally, comparing with phase-level variants could bring more insights. For example, the current phase 1 procedure is highly related to the name prediction tasks; what if using an alternative chemical corpus not including IUPAC and SMILES, or using different canonicalization methods during training and evaluation?

**Reviewer Concerns:**

During the rebuttal period, the authors provided additional experiments based on the feedback from reviewers:
- An additional Chem-R variant with a different teacher model.
- TOMG results on a 5% stratified random sample from the test set.
- Human evaluation on the quality of the reasoning process.
- OOD evaluation on unseen tasks from ChemCoTBench.

These results indeed improve the quality of the paper and bring new insights into the performance of Chem-R.

The key concern shared by multiple reviewers is: **the technical insight or novelty proposed by this paper** (RRrX qNm5 y3je). The authors' response claims the three "chemical constraints" as the core novelty. Based on the current manuscript, the key design in the methodology is (1) the curated chemical training corpus; (2) The Chemical CoT data synthesized by a designed protocol; and (3) During the RL post-training phase, adaptively controlling the distribution of sub-tasks based on the validation accuracies.

The proposed Chem-R consists of three phases. As pointed out by reviewers, their effects on Chem-R's performance are not clear. Based on the ablation study:
- For Molecule Tasks: After phase 1, performance has already saturated. It then drops significantly after phase 2, and recovers in phase 3.
- For Reaction Tasks: After phase 1, performance saturates on 2 out of 4 subtasks (React. and Retro). Phase 2 improves the remaining 2 subtasks while making the previous two worse. Finally, phase 3 improves or recovers all four subtasks.

This is interesting: it seems that phase 1 is already good enough for 7 out of 9 subtasks (except for Yield and Readg.). The remaining two phases are mainly for improving performance on the Yield task (and slightly on the Reag. task).

Therefore, **it needs further empirical justification on why and how these components would improve Chem-R's performance**. Current experiments report performance on end-to-end tasks. Although a detailed ablation study is provided in Table 2, there are no additional metrics characterizing the quality of intermediate components. For example, as mentioned by reviewer mFFx, the paper lacks more quantification of the behaviors of the protocols generated from the teacher model. I have carefully read the additional experiments in the appendix. While appreciating the authors' efforts, I believe more investigation (as suggested by reviewers) should be done.

**Reviewer Scores:**

Here is my estimation of the final scores if reviewers had fully engaged in the discussion:
- For RRrX, 2 out of 3 concerns have been fairly addressed. Since the original score is 6, i.e., a positive vote, there is about a 20% probability of increasing the score to 8; so the expected score is 6.4.
- For mFFx, many technical questions were posed. Some questions were not addressed as expected. For example, weakness 4 requires additional quantification, while current results are insufficient. Therefore, the expected score is 4, i.e., unchanged.
- For qNm5, 2 out of 3 concerns were not well addressed. The first is about novelty and scope; the second is about baselines. Although the authors provided additional results, the evaluation is limited to only a few subtasks. Therefore, the expected score is 4, i.e., unchanged.
- For y3je, the concerns are mainly about why and how the three phases would improve Chem-R's performance. As mentioned above, more fine-grained experiments and analysis are required to address this point. Therefore, the expected score is 4, i.e., unchanged.

---

### Decision · Program_Chairs · 2026-01-26

Reject